# From Optimization to Generalization under Heavy-Tailed Data: The Role of Gradient Clipping

**Aleksandr Shestakov** [1 2 3 †]   **Martin Takač** [3]   **Eduard Gorbunov** [3]

## Abstract

Gradient clipping is widely used to stabilize stochastic gradient methods and is often theoretically motivated by heavy-tailed gradient noise, where even second moments may be infinite, seemingly contradicting the finite-sum ERM setting, where all empirical moments are finite once the dataset is fixed. We resolve this paradox by explicitly separating data sampling from optimization randomness: although moments are finite conditional on the dataset, heavy-tailed data induce dataset-dependent noise whose second moment typically grows with the dataset size $N$. In particular, when $\|\nabla f(x_\star, \xi)\|$ has tail index $\alpha \in (1, 2)$, the quantity $\frac{1}{N}\sum_{i=1}^{N}\|\nabla f(x_\star, \xi_i)\|^2$ scales as $N^{\frac{2}{\alpha}-1}$, leading to deteriorating convergence guarantees for standard SGD as $N$ increases. In contrast, we show that SGD with clipping avoids this growth and admits finite-sum convergence guarantees under heavy-tailed data for broad step-size and clipping schedules. We further derive generalization bounds for strongly convex smooth objectives and show that the tail behavior of gradients at the population minimizer is the key quantity linking optimization and generalization under heavy-tailed data.

## 1. Introduction

Gradient-based training is the workhorse of modern machine learning, yet in many large-scale applications the raw stochastic gradients can be highly erratic. A simple and widely used remedy is *gradient clipping*, which rescales a stochastic gradient whenever its norm exceeds a prescribed

Part of this work was done when Aleksandr Shestakov was a visiting student at MBZUAI, UAE. [1]Basic Research of Artificial Intelligence Laboratory [2]Moscow Independent Research Institute of Artificial Intelligence [3]Mohamed bin Zayed University of Artificial Intelligence. Correspondence to: Aleksandr Shestakov <aleksandr.shestakov.opt@gmail.com>.

*Proceedings of the 43$^{rd}$ International Conference on Machine Learning*, Seoul, South Korea. PMLR 306, 2026. Copyright 2026 by the author(s).

threshold. Clipping was originally introduced in deep learning to mitigate exploding gradients (Pascanu et al., 2013; Mikolov et al., 2012), and it has since become a standard component in training large attention-based models (Zhang et al., 2020). Empirically, stochastic gradients in these settings often exhibit heavy tails (Gurbuzbalaban et al., 2021; Barsbey et al., 2021; Jiao & Keller-Ressel, 2024), motivating a growing body of theory on stochastic optimization under heavy-tailed noise and the benefits of robust updates such as clipping (Zhang et al., 2020; Gorbunov et al., 2020; Sadiev et al., 2023; Nguyen et al., 2023a).

In this paper, we study the population risk minimization problem

$$\min_{x \in \mathcal{X}} F(x) := \mathbb{E}_{\xi \sim \mathcal{D}}\big[f(x, \xi)\big], \tag{1}$$

where $\xi$ is a random data point and $f(x, \xi)$ is the loss of a model with parameters $x \in \mathcal{X} \subseteq \mathbb{R}^d$ on this data point. This population formulation is standard in statistics and machine learning (Snyman et al., 2005; Shalev-Shwartz & Ben-David, 2014; Goodfellow et al., 2016).

In practice we observe a dataset $S = \{\xi_i\}_{i=1}^{N}$ drawn i.i.d. from $\mathcal{D}$ and solve the empirical risk minimization (ERM) problem

$$\min_{x \in \mathcal{X}} F_N(x) := \frac{1}{N}\sum_{i=1}^{N} f(x, \xi_i). \tag{2}$$

A central question is how the ERM minimizer $\widehat{x} \in \arg\min F_N$ approximates the population minimizer $x_\star \in \arg\min F$. It is well known that the estimation error decreases with the sample size (Shalev-Shwartz et al., 2010; Valle-Pérez & Louis, 2020), which favors large datasets that better reflect the underlying distribution. At the same time, increasing $N$ also changes the *optimization regime*: practical training is typically compute-limited (a fixed number of iterations or epochs), and for a fixed budget the optimization error can become dominated by dataset-dependent noise effects.

Among the most popular algorithms for solving equation 1 and its empirical counterpart equation 2 are gradient-based methods (Robbins & Monro, 1951; Moulines & Bach, 2011; Ruder, 2016). For $F_N$, stochastic gradient descent (SGD) samples an index $i_t \sim \mathrm{Unif}(\{1, \ldots, N\})$ and performs

$$x^{t+1} = x^t - \gamma_t \nabla f(x^t, \xi_{i_t}). \tag{3}$$

While simple and effective, SGD is not always stable; for instance, it is sensitive to exploding gradients (Mikolov et al., 2012). In this work, we focus on stochastic gradient descent with clipping (ClipSGD),

$$x^{t+1} = x^t - \gamma_t \, \mathrm{clip}_{\lambda_t}\big(\nabla f(x^t, \xi_{i_t})\big),$$

$$\text{where} \quad \mathrm{clip}_c(v) := \min\left(1, \frac{c}{\|v\|}\right) v.$$

A common formalization of heavy-tailed stochastic gradients is that the centered gradient has only a finite $\alpha$-th moment for some $\alpha \in (1, 2]$:

$$\mathbb{E}\big\|\nabla f(x, \xi) - \nabla F(x)\big\|^{\alpha} < \infty, \qquad \forall x \in \mathcal{X}. \quad (4)$$

Under such assumptions, classical variance-based analyses of SGD no longer apply, whereas clipped (or truncated) methods admit non-asymptotic guarantees and can attain optimal or near-optimal rates for broad problem classes (Zhang et al., 2020; Gorbunov et al., 2020; Sadiev et al., 2023; Nguyen et al., 2023a).

At first glance, this "heavy-tailed noise" perspective seems at odds with finite-sum optimization: conditioned on a fixed dataset $S$, all moments of $\nabla f(x, \xi_i)$ are finite, and standard finite-sum analyses quantify stochasticity via dataset-dependent quantities such as

$$
\begin{aligned}
\sigma_{2,N} &:= \frac{1}{N} \sum_{i=1}^{N} \|\nabla f(\widehat{x}, \xi_i)\|^2 \\
&= \mathbb{E}_{i \sim \mathrm{Unif}([N])} \|\nabla f(\widehat{x}, \xi_i)\|^2, \quad (5)
\end{aligned}
$$

which is finite for every $N$. Therefore, the heavy-tailed noise seems not to affect the optimization complexity for finite-sum minimization, which is at odds with empirical observations. We resolve this apparent tension by explicitly separating *data sampling* (drawing $S$) from *optimization randomness* (sampling indices during training). Under heavy-tailed data, the dataset-dependent proxy in equation 5 typically *grows with $N$*: larger datasets contain more extreme samples, and these extremes can dominate the second-moment quantities that enter SGD convergence bounds when the compute budget is fixed.

To capture this effect sharply, we go beyond moment (non)existence and use tail asymptotics. We assume a regularly varying tail of the form

$$\mathbb{P}\left(\|\nabla f(x_\star, \xi)\| \geq r\right) \sim c \, r^{-\alpha}, \qquad r \to \infty,$$

for some $\alpha \in (1, 2]$. Such distributions lie in the domain of attraction of an $\alpha$-stable law (Gnedenko & Kolmogorov, 1968; Uchaikin & Zolotarev, 2011), which yields sharp scaling for dataset averages of heavy-tailed quantities – precisely the objects that appear in finite-sum optimization guarantees. This perspective explains when and why clipping removes the harmful growth with $N$ and enables clean links between optimization and generalization under heavy-tailed data.

## 1.1. Contributions

Our main contributions are summarized below.

- **Finite-sum convergence with clipping under heavy-tailed data.** We prove convergence guarantees for ClipSGD on $F_N$ in a finite-sum setting where the dataset is sampled from a heavy-tailed distribution. The bounds hold for broad non-increasing step-size schedules and clipping thresholds above an explicit stability level.

- **Dataset-size dependence of noise terms.** Using stable-law scaling, we characterize how heavy tails induce a dataset-dependent noise proxy that grows with $N$ for unclipped SGD (typically as $N^{\frac{2}{\alpha}-1}$), while clipping removes this growth.

- **Generalization under heavy tails and the role of the minimizer.** For strongly convex smooth objectives we derive first ERM generalization bounds under heavy-tailed data and connect them to optimization guarantees, highlighting that the tail behavior near the population minimizer governs the relevant constants.

## 2. Related work

The literature on gradient clipping, heavy-tailed stochastic optimization, and generalization is vast. We therefore focus on three closely related directions: (i) convergence guarantees for clipped/truncated first-order methods under heavy-tailed noise, (ii) generalization bounds under weak tail assumptions, and (iii) finite-sum analyses of SGD and related variance-reduced methods, with particular emphasis on how the relevant "noise-at-the-optimum" quantities enter the bounds.

**Gradient clipping under heavy-tailed noise.** Gradient clipping was introduced in deep learning as a practical safeguard against exploding gradients (Pascanu et al., 2013) and has since become a standard component of training pipelines, including large attention-based models (Zhang et al., 2020). Its use is often motivated by empirical and theoretical evidence that stochastic gradients can exhibit heavy-tailed behavior (Şimşekli et al., 2019; Gurbuzbalaban et al., 2021; Barsbey et al., 2021; Jiao & Keller-Ressel, 2024). On the theory side, robust stochastic optimization under weak moment assumptions predates modern deep learning: Nemirovskij & Yudin (1983) analyzed Mirror Descent with only an $\alpha$-th moment, and Vural et al. (2022) sharpened these guarantees and established matching lower bounds for uniformly convex objectives. More directly related to clipping/truncation, Nazin et al. (2019) studied truncated stochastic mirror descent and obtained early high-probability guarantees under mild noise assumptions.

A line of work then developed convergence guarantees specifically for clipped gradient methods under heavy tails.

Zhang et al. (2020) provided one of the first analyses of ClipSGD under bounded central $\alpha$-th moments, covering smooth non-convex and strongly convex objectives and linking clipping to the empirical success of attention models. Subsequent papers established high-probability bounds with polylogarithmic dependence on the confidence level, starting with accelerated clipping schemes for smooth convex and strongly convex problems (Gorbunov et al., 2020) and extending to broader settings, including non-smooth and composite/distributed formulations (Gorbunov et al., 2024a; Parletta et al., 2024; Sadiev et al., 2023; Nguyen et al., 2023b;a; Gorbunov et al., 2024b; Das et al., 2024; Chezhegov et al., 2025a;b). Beyond clipping, closely related robust updates based on normalization or sign operators have also been analyzed in the heavy-tailed regime (Cutkosky & Mehta, 2021; Hübler et al., 2024; Kornilov et al., 2025). Complementary results study when *plain* SGD can still converge under infinite-variance noise (typically under additional structural assumptions) and characterize stable-law limits (Wang et al., 2021; Fatkhullin et al., 2025). In addition, convergence of different analogs of ClipSGD under symmetric (and close to symmetric) heavy-tailed noise are studied by Armacki et al. (2023; 2024); Puchkin et al. (2024). Finally, orthogonal to clipping-based techniques, Davis et al. (2021) shows how to boost low-probability guarantees to high confidence in stochastic convex optimization via an inexact proximal-point framework combined with robust distance estimation.

Most of the above analyses are stated for the streaming setting and quantify stochasticity through moment/tail conditions that hold uniformly along the trajectory. In contrast, we explicitly separate data sampling and optimization randomness in the *finite-sum* regime: the dataset itself is drawn from a heavy-tailed distribution, which makes the classical "variance-at-the-optimum" terms random and typically growing with $N$. Our results characterize this dataset-size dependence via stable-law scaling and show how clipping removes the harmful growth.

**Generalization and stability under heavy-tailed data.** Stability-based generalization theory originates with Bousquet & Elisseeff (2002), with extensions to randomized algorithms by Elisseeff et al. (2005); Shalev-Shwartz et al. (2010) further shows that stability is essentially equivalent to learnability. For SGD, Hardt et al. (2016) derived uniform stability bounds under smoothness and bounded gradients, and Bottou & Bousquet (2007) popularized the decomposition of excess error into approximation, estimation, and optimization terms. In the heavy-tailed regime, concentration may fail and ERM analysis requires different tools: Mendelson (2015) derived sharp estimation error bounds without relying on classical concentration. More recent work studies generalization of heavy-tailed stochastic dynamics and sample-average approximation: Raj et al.

(2023) analyze algorithmic stability via heavy-tailed SDEs (for Lipschitz losses), and Oliveira & Thompson (2023) provide non-asymptotic bounds for heavy-tailed sample-average approximation under weak assumptions. Closest to our setting, Liu & Tong (2024) derive in-expectation and high-probability excess-risk bounds for convex and strongly convex *smooth* losses without requiring Lipschitzness, under bounded gradient variance. Our generalization analysis builds on this stability/SAA perspective and extends it to settings with potentially unbounded variance, highlighting that the tail behavior of $\nabla f(x_\star, \xi)$ at the population minimizer is the key quantity that controls the constants.

**Finite-sum minimization.** Classical finite-sum analyses of SGD in the strongly convex regime (Moulines & Bach, 2011) and their extension to arbitrary sampling via stochastic reformulation (Gower et al., 2019) yield rates that depend on variance bounds of stochastic gradients at the empirical minimizer. Similar dependence appears in incremental methods such as SAG (Schmidt et al., 2017). In contrast, other methods such as SDCA, SVRG, and SAGA and their accelerated variants avoid explicit dependence on the variance at the optimum, at the cost of convergence bounds that depend explicitly on $N$ (Shalev-Shwartz & Zhang, 2013; Johnson & Zhang, 2013; Defazio et al., 2014; Allen-Zhu, 2018). Our work connects these finite-sum viewpoints to heavy-tailed data: under heavy-tailed sampling, the usual noise proxy at the optimum can itself scale with $N$ even though it is finite conditional on a fixed dataset, and we show that gradient clipping removes precisely this dataset-size deterioration.

## 3. Main results

**Notation.** We use the standard Euclidean norm for vectors: $\|x\| := \langle x, x \rangle^{1/2}, x \in \mathbb{R}^d$. We consider stochastic realizations to be differentiable functions, furthermore, we assume that differentiation and expectation can be interchanged, so that $\nabla F(x) = \mathbb{E}_{\xi \sim \mathcal{D}}[\nabla f(x, \xi)]$. By $x_\star$ we denote the optimum of the initial problem (1), and by $\widehat{x}$ the minimizer of the finite-sum problem (2). By $\nabla f_i(x)$ we denote $\nabla f(x, \xi_i)$. For positive sequences $\{X_N\}$ and $\{a_N\}$, we write $X_N = \Theta_\mathbb{P}(a_N)$ if for every $\delta \in (0, 1)$ there exist constants $0 < c_\delta < C_\delta < \infty$ and $N_0$ such that

$$\mathbb{P}(c_\delta a_N \leq X_N \leq C_\delta a_N) \geq 1 - \delta, \qquad \forall N \geq N_0.$$

**Assumptions.** Below, we provide all the assumptions used in the paper. Our results are derived either for convex and strongly convex problems.

**Assumption 3.1** (Convexity). *For almost every $\xi$, the function $f(x, \xi)$ is convex on $\mathbb{R}^d$, i.e.,*

$$f(y, \xi) \geq f(x, \xi) + \langle \nabla f(x, \xi), y - x \rangle, \quad \forall x, y \in \mathbb{R}^d.$$

**Assumption 3.2** (Strong convexity). *For almost every $\xi$, the function $f(x, \xi)$ is $\mu$-strongly convex with $\mu > 0$, i.e.,*

$$f(y, \xi) \geq f(x, \xi) + \langle \nabla f(x, \xi), y - x \rangle + \frac{\mu}{2}\|y - x\|^2, \forall x, y \in \mathbb{R}^d.$$

Convexity and strong convexity are not only standard assumptions in optimization (Nesterov et al., 2018), but also common in the analysis of generalization error (Shalev-Shwartz et al., 2010; Liu & Tong, 2024), as they rule out non-global local optima.

Next, we rely on the following assumptions about smoothness, the domain, and tail index.

**Assumption 3.3** (Smoothness)**.** *For almost every $\xi$, the function $f(x,\xi)$ is L-smooth, i.e.,*

$$\|\nabla f(x,\xi) - \nabla f(y,\xi)\| \leq L\|y - x\|, \quad \forall x, y \in \mathbb{R}^d.$$

**Assumption 3.4** (Compact set)**.** *The set $\mathcal{X} \subset \mathbb{R}^d$ is closed, convex, and has finite diameter, i.e.,*

$$\Delta := \max\{\|x - y\| : x, y \in \mathcal{X}\} < \infty.$$

**Assumption 3.5** (Heavy-tailed noise at the minimizer)**.** *There exist $c > 0$ and a tail index $\alpha \in (1, 2]$ such that*

$$\mathbb{P}\big(\|\nabla f(x_\star, \xi) - \nabla F(x_\star)\| \geq r\big) \sim c\, r^{-\alpha}.$$

*When $\nabla F(x_\star) = 0$, this condition reduces to the tail assumption on $\|\nabla f(x_\star, \xi)\|$.*

The last assumption encompasses the standard conditions on the existence of moments. The importance of stochasticity at the optimum has been known for a long time from the statistical point of view, since weak convergence of estimators is characterized by the variance at the minimizer (Kushner & Clark, 2012). Focusing on the minimizer avoids imposing uniform tail or moment bounds along the entire optimization trajectory, which is particularly important when the tail index changes during training. For heavy-tailed noise this is especially relevant, as its tail index may change significantly along the optimization trajectory, and the noise at the optimum may have lighter tails than the noise encountered along the optimization path (Şimşekli et al., 2019).

The structure of the stochastic noise during training has also been the subject of extensive recent investigation (Şimşekli et al., 2019; Wang et al., 2021; Gurbuzbalaban et al., 2021; Lim et al., 2022; Barsbey et al., 2021). One explanation for its behavior is that it belongs to the class of stable distributions (Uchaikin & Zolotarev, 2011). While this assumption is quite expressive, a subtle issue is that stability of the vector

$$\varepsilon := \nabla f(x, \xi) - \nabla F(x)$$

does not imply stability of, for instance, $\|\varepsilon\|$, and vice versa. However, stability of either of these variables still leads to heavy-tailed noise. To relax such strong assumptions on exact stability, we assume only the asymptotic tail behavior.

### 3.1. Generalization bounds

We begin with algorithm-agnostic bounds on the excess population risk of ERM under heavy-tailed data.

**Theorem 3.1.** *Let the Assumptions 3.2, 3.3, 3.4, 3.5 be satisfied with tail index $\alpha \in (1, 2]$. Then for any $q \in (1, \alpha)$*

*and for $N = \Omega\left(\left(\frac{L\Delta^{2-q}}{\mu^{q-1}(\mathbb{E}\|\nabla f(x_\star,\xi)\|^q)^{\frac{2-q}{q}}}\right)^{\frac{1}{q-1}}\right)$, we have*

$$\mathbb{E}\big[F(\widehat{x}) - F(x_\star)\big] = \mathcal{O}\left(\frac{\Delta^{2-q}\, \mathbb{E}\|\nabla f(x_\star,\xi)\|^q}{\mu^{q-1}\, N^{q-1}}\right). \quad (6)$$

*Moreover, with probability at least $1 - \beta$,*

$$F(\widehat{x}) - F(x_\star) = \mathcal{O}\left(\frac{\Delta^{2-q}\, \mathbb{E}\|\nabla f(x_\star,\xi)\|^q}{\mu^{q-1}\, N^{q-1}\, \beta}\right). \quad (7)$$

Theorem 3.1 characterizes the influence of the heavy-tailed noise on the generalization bounds. The bounds are similar in spirit to the classical $\mathcal{O}(1/N)$ rates under the finite variance (Bousquet & Elisseeff, 2002; Shalev-Shwartz et al., 2010; Liu & Tong, 2024).

We next show that this degradation is unavoidable under heavy tails by exhibiting a one-dimensional construction where the ERM excess risk matches the rate in Theorem 3.1 up to any infinitesimal difference in the exponent.

**Lemma 3.1.** *Consider the minimization problem $F(x) = \mathbb{E}f(x,\xi)$ on the domain $\left[-\frac{\Delta}{2}, \frac{\Delta}{2}\right]$ with $f(x,\xi) = \frac{\mu}{2}x^2 + \xi x$, where $\xi$ is drawn from two-sided Pareto distribution with index $\alpha$. Then, we have*

$$\mathbb{E}\left[F(\widehat{x}) - F(x_\star)\right] = \Omega\left(\frac{\Delta^{2-\alpha}}{\mu^{\alpha-1}N^{\alpha-1}}\right). \quad (8)$$

In addition to the strongly convex setup, generalization bounds for convex setup with Tikhonov regularization can also be derived. The corresponding results and all the necessary proofs are given in Appendix A.

### 3.2. Convergence guarantees for convex functions

We now analyze ClipSGD (Algorithm 1) for minimizing convex smooth finite-sum objectives under heavy-tailed data. The goal is to obtain bounds that depend on noise properties at (or near) the minimizer rather than uniform bounds along the whole trajectory.

---

**Algorithm 1** ClipSGD

---

1: **Input:** step sizes $\{\gamma_t\}_{t \geq 1}$, clipping radii $\{\lambda_t\}_{t \geq 1}$, initial point $x^1 \in \mathcal{X}$, constraint set $\mathcal{X}$
2: **for** $t = 1, 2, \ldots, T$ **do**
3:     Sample $i_t \sim \text{Unif}(\{1, \ldots, N\})$
4:     $x^{t+1} = \text{proj}_{\mathcal{X}}\big(x^t - \gamma_t \text{clip}_{\lambda_t}(\nabla f(x^t, \xi_{i_t}))\big)$
5: **end for**

---

**Theorem 3.2.** *Let the Assumptions 3.1, 3.3 and 3.4 be satisfied. Additionally, demand the step size sequence to be non-increasing $\gamma_{t+1} \leq \gamma_t$, $\gamma_t \leq \frac{1}{4L}$, and $\lambda_t \geq 2^5 3^2 L\Delta + 2\|\nabla F_N(x^1)\|$. Then, for $\forall \alpha \in (1, 2]$ the iterates of the ClipSGD satisfy*

$$\mathbb{E}_{\text{alg}}\left[F_N(\overline{x}^T) - F_N(\widehat{x})\right]$$

$$= \mathcal{O}\left(\frac{\Delta^2}{\gamma_T T} + \frac{\mathbb{E}_i\|\nabla f_i(x_\star)\|^\alpha + L^\alpha \Delta^\alpha}{T}\sum_{t=1}^T \gamma_t \lambda_t^{2-\alpha}\right.$$

$$+ \frac{\Delta\mathbb{E}_i\|\nabla f_i(x_\star)\|^\alpha + L^\alpha \Delta^{\alpha+1}}{T}\sum_{t=1}^T \lambda_t^{1-\alpha}$$

$$\left.+ I(\alpha \neq 2)\frac{\Delta^{\frac{2}{2-\alpha}} L^{\frac{\alpha}{2-\alpha}}}{T}\sum_{t=1}^T \lambda_t^{\frac{2-2\alpha}{2-\alpha}}\right),$$

*where $\overline{x}^T = \frac{1}{T}\sum_{t=1}^T x^t$, $\mathbb{E}_{\text{alg}}$ is the expectation with respect to the random indices sampled during the algorithm, conditionally on the dataset, and $\mathbb{E}_i$ denotes the uniform average over the fixed dataset.*

**Corollary 3.3.** *There exist horizon-fixed $\gamma_t \sim \frac{1}{T^{1/\alpha}}, \lambda_t \sim T^{1/\alpha}$ and horizon-free $\gamma_t \sim \frac{1}{t^{1/\alpha}}, \lambda_t \sim t^{1/\alpha}$ schedules, that give $\mathcal{O}\left(T^{\frac{1-\alpha}{\alpha}}\right)$ convergence rates.*

The stability condition involves $\|\nabla F_N(x^1)\|$. This quantity can be replaced by $\|\nabla F_N(x^0)\|$ for any fixed reference point $x^0 \in \mathcal{X}$; we choose $x^1$ because the initial point is known before the algorithm starts. In practice, this threshold can be treated as a conservative clipping parameter rather than computed exactly.

A notable feature of Theorem 3.2 is that the bound depends on gradient statistics at the population minimizer $x_\star$, rather than requiring uniform moment bounds along the entire trajectory. This aligns with empirical observations that the tail index can vary substantially during training and supports using larger step sizes early on (Şimşekli et al., 2019).

While the optimization target is $F_N$, our bounds are written in terms of quantities at $x_\star$ to enable a clean link to generalization: controlling gradients at the population minimizer provides a dataset-level proxy that governs both the ERM excess risk and the optimization noise terms.

An important implication of Theorem 3.2 is that clipping can be beneficial even when one does not know the exact tail index in advance: the method converges under broad schedules, and the main role of $\alpha$ is to determine how to tune $\{\gamma_t, \lambda_t\}$ to optimize the rate, as we discuss below.

**Comparison to unclipped SGD.** To highlight the role of clipping in the finite-sum regime, we compare our bounds to standard SGD bounds (Bach, 2024), which depend on a second-moment such as $\sigma_{2,N}$ defined in (5):

$$\mathbb{E}_{\text{alg}}\left[F_N(\overline{x}^T) - F_N(\widehat{x})\right] = \mathcal{O}\left(\frac{\sigma_{2,N}}{\sqrt{T}}\right)$$

$$= \mathcal{O}\left(\frac{\Delta^2}{\sqrt{T}} + \frac{\mathbb{E}_i\|\nabla f_i(x_\star)\|^2}{\sqrt{T}}\right),$$

where we use $\sigma_{2,N} = \mathbb{E}_i\|\nabla f_i(\widehat{x}) \pm \nabla f_i(x_\star)\|^2 \leq$

$2\mathbb{E}_i\|\nabla f_i(\widehat{x}) - \nabla f_i(x_\star)\|^2 + 2\mathbb{E}_i\|\nabla f_i(x_\star)\|^2 \overset{\text{As. 3.3}}{\leq}$ $L^2\mathbb{E}_i\|\widehat{x} - x_\star\|^2 + 2\mathbb{E}_i\|\nabla f_i(x_\star)\|^2 \overset{\text{As. 3.4}}{\leq} L^2\Delta^2 + 2\mathbb{E}_i\|\nabla f_i(x_\star)\|^2$. At first glance, classical SGD bounds may suggest a better dependence on $T$. The key distinction in our setting is that the noise proxy entering SGD bounds is *dataset-dependent*. Under heavy-tailed data, this proxy can grow with $N$ as the dataset contains more extreme samples, and this growth can dominate the overall error when $N$ is large. The classical result (Gnedenko & Kolmogorov, 1968; Feller, 1991) states that for heavy-tailed random variables, after appropriate normalization, converge to an $\alpha$-stable law. In our setting, this yields sharp scaling for dataset averages such as $\frac{1}{N}\sum_{i=1}^N \|\nabla f(x_\star, \xi_i)\|^2$:

**Theorem 3.4.** *Let the Assumption 3.5 be satisfied. Then, for $\alpha < 2$ we have*

$$\frac{1}{N}\sum_{i=1}^N \|\nabla f_i(x_\star)\|^2 = \Theta_{\mathbb{P}}\left(N^{\frac{2}{\alpha}-1}\right), \tag{9}$$

*and for $\alpha = 2$ we have*

$$\frac{1}{N}\sum_{i=1}^N \|\nabla f_i(x_\star)\|^2 = \Theta_{\mathbb{P}}(\ln N). \tag{10}$$

We note that $\nabla F(x_\star)$ is a deterministic vector, meaning that $\|\nabla f(x_\star, \xi) - \nabla F(x_\star)\|$ and $\|\nabla f(x_\star, \xi)\|$ have the same tail index. A precise high-probability version and its proof are given in Appendix B. It is also worth noticing that the asymptotics of $\frac{1}{N}\sum_{i=1}^N \|\nabla f(x, \xi_i)\|^p$ for $p \in (1, 2)$ can also be established, as shown below.

**Theorem 3.5.** *Let the Assumption 3.5 be satisfied with $\alpha < 2$. Then, for $p < \alpha$ we have*

$$\frac{1}{N}\sum_{i=1}^N \|\nabla f_i(x_\star)\|^p = \Theta_{\mathbb{P}}(1), \tag{11}$$

*for $p = \alpha$*

$$\frac{1}{N}\sum_{i=1}^N \|\nabla f_i(x_\star)\|^p = \Theta_{\mathbb{P}}(\ln N), \tag{12}$$

*and for $p > \alpha$*

$$\frac{1}{N}\sum_{i=1}^N \|\nabla f_i(x_\star)\|^p = \Theta_{\mathbb{P}}\left(N^{\frac{p}{\alpha}-1}\right). \tag{13}$$

As a consequence, convergence of unclipped SGD deteriorates as $N$ grows because its noise term scales with the dataset-dependent quantity in Theorem 3.4:

$$\mathbb{E}_{\text{alg}}\left[F_N(\overline{x}^T) - F_N(\widehat{x})\right] = \mathcal{O}\left(\frac{\Delta^2}{\sqrt{T}} + \frac{N^{\frac{2}{\alpha}-1}}{\sqrt{T}}\right), \tag{14}$$

In contrast, aggressive clipping removes the polynomial growth in $N$ caused by the empirical second moment:

$$\mathbb{E}_{\text{alg}}\left[F_N(\overline{x}^T) - F_N(\widehat{x})\right] = \mathcal{O}\left(\frac{\Delta^2}{T^{\frac{\alpha-1}{\alpha}}} + \frac{\mathbb{E}_i\|\nabla f_i(x_\star)\|^\alpha}{T^{\frac{\alpha-1}{\alpha}}}\right),$$
(15)

where the last term does not increase with the growth of $N$. If the clipping is chosen at the critical tail index, the growth is only logarithmical, which is significantly better than polynomial one. This provides a finite-sum explanation for why clipping is effective in large-scale regimes.

**Step-size-dependent and tail-index-agnostic bounds.**

While Corollary 3.3 suggests choosing $\gamma_t \sim t^{-1/\alpha}$ and $\lambda_t \sim t^{1/\alpha}$, the convergence holds regardless of the problem's heavy-tailed index. Hence, it might be beneficial to choose $\gamma_t$ and $\lambda_t$ to be not related to the true index. It is therefore important to understand how performance changes when the tuning parameter differs from the true tail index.

Here and further we denote by $\alpha_{\text{alg}}$ the index used to tune the schedules $\gamma_t \sim t^{-1/\alpha_{\text{alg}}}$ and $\lambda_t \sim t^{1/\alpha_{\text{alg}}}$, and by $\alpha_{\text{true}}$ the tail index in the Assumption 3.5.

**Corollary 3.6.** *Let all the Assumptions for the Theorem 3.2 be satisfied, as well as Assumption 3.5. Let the step sizes be chosen as $\gamma_t = \gamma_0 t^{-1/\alpha_{\text{alg}}}$, where $\gamma_0 \leq \frac{1}{4L}$, and clipping radii be chosen as $\lambda_t = \lambda_0 t^{1/\alpha_{\text{alg}}}$, where $\lambda_0 \geq 2L\Delta + 2\|\nabla F_N(x^1)\|$. Then, depending on the relation between $\alpha_{\text{alg}}$ and $\alpha_{\text{true}}$ we obtain the following convergence:*

*1. If $\alpha_{\text{alg}} < \alpha_{\text{true}}$, then,*

$$\mathbb{E}_{\text{alg}}\left[F_N(\overline{x}^T) - F_N(\widehat{x})\right]$$
$$= \mathcal{O}\left(\frac{\Delta^2}{T^{\frac{\alpha_{\text{alg}}-1}{\alpha_{\text{alg}}}}} + \frac{\mathbb{E}\|\nabla f(x_\star, \xi)\|^{\alpha_{\text{alg}}}}{T^{\frac{\alpha_{\text{alg}}-1}{\alpha_{\text{alg}}}}}\right);$$

*2. If $\alpha_{\text{alg}} = \alpha_{\text{true}}$, then,*

$$\mathbb{E}_{\text{alg}}\left[F_N(\overline{x}^T) - F_N(\widehat{x})\right]$$
$$= \mathcal{O}\left(\frac{\Delta^2}{T^{\frac{\alpha_{\text{alg}}-1}{\alpha_{\text{alg}}}}} + \frac{\ln N}{T^{\frac{\alpha_{\text{alg}}-1}{\alpha_{\text{alg}}}}}\right);$$

*3. If $\alpha_{\text{alg}} > \alpha_{\text{true}}$, then,*

$$\mathbb{E}_{\text{alg}}\left[F_N(\overline{x}^T) - F_N(\widehat{x})\right]$$
$$= \mathcal{O}\left(\frac{\Delta^2}{T^{\frac{\alpha_{\text{alg}}-1}{\alpha_{\text{alg}}}}} + \frac{N^{\frac{\alpha_{\text{alg}}}{\alpha_{\text{true}}}-1}}{T^{\frac{\alpha_{\text{alg}}-1}{\alpha_{\text{alg}}}}}\right).$$

Note, that expectation $\mathbb{E}$ in the RHS of the first bound stands for the expectation across all the distribution, not only across the samples drawn from the dataset.

One benefit of the analysis is that it applies to arbitrary scheduling parameter $\alpha_{\text{alg}}$. The trade-off, caused by the mismatch between the tail indices is dependent on the size of the dataset. Previously, several works analyzed tail-index agnostic clipping (Koloskova et al., 2023; Hübler et al.,

2024), however, they have provided bounds in the non-convex setup and do not ERM setting, while we provide bounds in the convex and strongly convex cases and characterize the effect of the heavy-tailed data in the finite-sum setting.

It can be seen that with different regimes, depending on the number of samples $N$ and number of iterations $T$ it might be preferable to choose $\alpha_{\text{alg}}$ not matching the $\alpha_{\text{true}}$. Basically, there are three distinct regimes, based on relation between $N$ and $T$, where the convergence bounds differ significantly: $N \gg T$, $N \approx T$, and $N \ll T$.

When $N \gg T$, the number of samples significantly exceeds the number of iterations, and the problem setting is similar to online optimization (Orabona, 2019), where a new data sample is drawn at each iteration. In this setup, the effect of heavy-tailed noise is most pronounced, since the algorithm does not capture the ERM structure. This is consistent with our previous results, in which we drew new samples from the dataset independently at each step. In contrast, when $N \ll T$, we perform multiple passes over the dataset. This regime is typical, for instance, in computer vision problems (He et al., 2016), where the presence of heavy-tailed noise is less influential. The last setup, but no less important, is when the number of samples and the number of iterations are comparable: $N \approx T$. In this scenario, both ERM and stochastic effects influence the optimization process equally and cannot be ignored. Typical examples of this regime include both pre-training and fine-tuning of LLMs (Brown et al., 2020; Touvron et al., 2023), where the number of epochs rarely exceeds 10.

Therefore, depending on the relation between $N$ and $T$ it might be preferable to choose different $\alpha_{\text{alg}}$ for scheduling the step sizes and clipping radii for ClipSGD.

**Lemma 3.2.** *Let all the Assumptions for the Theorem 3.2 and Assumption 3.5 be satisfied. Then, for 3 different regimes, based on relation between $N$ and $T$ we have following optimal choice of $\alpha_{\text{alg}}$:*

*1. For $N \gg T$ the optimal choice is $\alpha_{\text{alg}} = \alpha_{\text{true}}$, which results in overall bound $\mathcal{O}\left(\ln N/T^{\frac{\alpha_{\text{true}}-1}{\alpha_{\text{true}}}}\right)$.*

*2. For $N \approx T$ the optimal choice is $\alpha_{\text{alg}} = \alpha_{\text{true}}$ which results in overall bound $\mathcal{O}\left(\ln T/T^{\frac{\alpha_{\text{true}}-1}{\alpha_{\text{true}}}}\right)$.*

*3. For $N \ll T$ the optimal choice is $\alpha_{\text{alg}} = 2$ which results in overall bound $\mathcal{O}\left(1/\sqrt{T}\right)$.*

When $\alpha_{\text{alg}}$ approaches $\alpha_{\text{true}}$ from below, the corresponding population moment becomes increasingly large, which can make constants unstable in finite samples. For this reason we focus on choices $\alpha_{\text{alg}} \geq \alpha_{\text{true}}$ when discussing fixed finite budgets.

If we additionally tune the hyperparameters $\gamma_0$ and $\lambda_0$ on dataset statistics as follows:

$$\gamma_0 = \Delta \left( \frac{N}{\sum_{i=1}^{N} \|\nabla f_i(x_\star)\|^{\alpha_{\text{alg}}}} \right)^{1/\alpha_{\text{alg}}}, \quad (16)$$

$$\lambda_0 = \left( \frac{1}{N} \sum_{i=1}^{N} \|\nabla f_i(x_\star)\|^{\alpha_{\text{alg}}} \right)^{1/\alpha_{\text{alg}}}, \quad (17)$$

it can further improve constants, analogously to variance-dependent tuning in classical SGD.

**Lemma 3.3.** *Let all the Assumptions for the Theorem 3.2 and Assumption 3.5 be satisfied and the tuned $\gamma_0, \lambda_0$ are used as defined above. Then, for 3 different regimes, based on relation between $N$ and $T$ we have the following optimal choice of $\alpha_{\text{alg}}$:*

1. *For $N \gg T$ the optimal choice is $\alpha_{\text{alg}} = \alpha_{\text{true}}$, which results in overall bound $\mathcal{O}\left( (\ln N)^{1/\alpha_{\text{true}}} / T^{\frac{\alpha_{\text{true}}-1}{\alpha_{\text{true}}}} \right)$.*

2. *For $N \approx T$ the optimal choice is $\alpha_{\text{alg}} > \alpha_{\text{true}}$ which result in overall bound $\mathcal{O}\left( 1/T^{\frac{\alpha_{\text{true}}-1}{\alpha_{\text{true}}}} \right)$.*

3. *For $N \ll T$ the optimal choice is $\alpha_{\text{alg}} = 2$ which results in overall bound $\mathcal{O}\left( 1/\sqrt{T} \right)$.*

Overall, even coarse information about the heaviness of the tails can be useful for selecting conservative clipping and step-size schedules, since asymptotics of $\frac{1}{N} \sum_{i=1}^{N} \|\nabla f_i(x_\star)\|^{\alpha_{\text{alg}}} \sim N^{\frac{\alpha_{\text{alg}}}{\alpha_{\text{true}}}-1}$ can be similarly derived as in Theorem 3.4. This is especially useful when $N$ is large relative to the number of iterations.

### 3.3. Convergence guarantees for strongly convex functions

Analogously to the convex case, we obtain the convergence bounds for the strongly convex setup.

**Theorem 3.7.** *Let the Assumptions 3.2, 3.3, 3.4 be satisfied. Then, for any $\alpha \in (1,2]$ choosing $\gamma_t = \frac{4}{\mu(t+16L/\mu)}, \lambda_t = \lambda_0 t^{1/\alpha}$ with $\lambda_0 \geq \max\{2^5 3^2 L\Delta + 2\|\nabla F_N(x^1)\|, (2^{3\alpha+5} 3^{2\alpha} L^\alpha(\Delta\|\nabla F_N(x^1)\| + L\Delta^2)^{\alpha-1} \cdot \mu^{-1})^{1/(2(\alpha-1))}\}$ results in the following convergence bounds for ClipSGD:*

$$\mathbb{E}_{\text{alg}}\left[ F_N(\widetilde{x}^T) - F_N(\widehat{x}) \right] = \mathcal{O}\left( \frac{L^2\Delta^2}{\mu T^2} + \right.$$

$$+ \frac{(\mathbb{E}_i\|\nabla f_i(x_\star)\|^\alpha)^2 \lambda_0^{2(1-\alpha)}}{\mu T^{\frac{2(\alpha-1)}{\alpha}}} + \frac{\mathbb{E}_i\|\nabla f_i(x_\star)\|^\alpha \lambda_0^{2-\alpha}}{\mu T^{\frac{2(\alpha-1)}{\alpha}}}$$

$$\left. + \textit{faster decaying terms} \right),$$

*and if $\lambda_t$ is chosen as $\lambda_t = \lambda_0 t$ with $\lambda_0 \geq 2^5 3^2 L\Delta + 2\|\nabla F_N(x^1)\|$, then, it results in the following convergence of ClipSGD:*

$$\mathbb{E}_{\text{alg}}\left[ F_N(\widetilde{x}^T) - F_N(\widehat{x}) \right] = \mathcal{O}\left( \frac{L^2\Delta^2}{\mu T^2} + \right.$$

$$+ \frac{\mathbb{E}_i\|\nabla f_i(x_\star)\|^\alpha \lambda_0^{2-\alpha}}{\mu T^{\alpha-1}} + \frac{\Delta\mathbb{E}_i\|\nabla f_i(x_\star)\|^\alpha \lambda_0^{1-\alpha}}{\mu T^{\alpha-1}}$$

$$\left. + \textit{faster decaying terms} \right),$$

*where $\widetilde{x}^T = \frac{1}{W_T} \sum_{t=1}^{T} w_t x^t$, $W_T = \sum_{t=1}^{T} w_t$, $w_t = t + \frac{16L}{\mu}$, $\mathbb{E}_{\text{alg}}$ is the expectation with respect to the random indices sampled during the algorithm, conditionally on the dataset, and $\mathbb{E}_i$ denotes the uniform average over the fixed dataset.*

The main difference from the convex setup is the possibility of two distinct bounds, that arise from different analyses. When previously, the latter bound was was not taken into consideration, due to the worse dependence on $T$, the first one suffers from $(\mathbb{E}_i\|\nabla f_i(x_\star)\|^\alpha)^2$, which influences the convergence more significantly, in case $N \gg T$ or $N \approx T$. Similar to the convex case we analyze the different regimes of ClipSGD:

**Lemma 3.4.** *Let all the Assumptions for Theorem 3.7 and Assumption 3.5 be satisfied. Depending on the relation between the $\alpha_{\text{alg}}$ and $\alpha_{\text{true}}$ the noise terms admit three regimes*

1. *If $\alpha_{\text{alg}} < \alpha_{\text{true}}$ the term scales as $\mathcal{O}\left( \min\left\{ (\mathbb{E}\|\nabla f(x_\star,\xi)\|^{\alpha_{\text{alg}}})^2 / T^{\frac{2(\alpha_{\text{alg}}-1)}{\alpha_{\text{alg}}}}; \right.\right.$*

   $$\left.\left. \mathbb{E}\|\nabla f(x_\star,\xi)\|^{\alpha_{\text{alg}}} / T^{\alpha_{\text{alg}}-1} \right\} \right)$$

2. *If $\alpha_{\text{alg}} = \alpha_{\text{true}}$ it scales as $\mathcal{O}\left( \min\left\{ (\ln N)^2 / T^{\frac{2(\alpha_{\text{alg}}-1)}{\alpha_{\text{alg}}}}; \ln N / T^{\alpha_{\text{alg}}-1} \right\} \right)$*

3. *If $\alpha_{\text{alg}} > \alpha_{\text{true}}$ it scales as*

   $$\mathcal{O}\left( \min\left\{ N^{\frac{2\alpha_{\text{alg}}}{\alpha_{\text{true}}}-2} / T^{\frac{2(\alpha_{\text{alg}}-1)}{\alpha_{\text{alg}}}}; N^{\frac{\alpha_{\text{alg}}}{\alpha_{\text{true}}}-1} / T^{\alpha_{\text{alg}}-1} \right\} \right)$$

**Lemma 3.5.** *Let all the Assumptions for Theorem 3.7 and Assumption 3.5 be satisfied. Then, for 3 different regimes, based on relation between $N$ and $T$, we have the following optimal choice of $\alpha_{\text{alg}}$:*

1. *For $N \gg T$ the optimal choice is $\alpha_{\text{alg}} = \alpha_{\text{true}}$ which results in overall bound $\mathcal{O}\left( \ln N / T^{\alpha_{\text{true}}-1} \right)$. The second bound is optimal.*

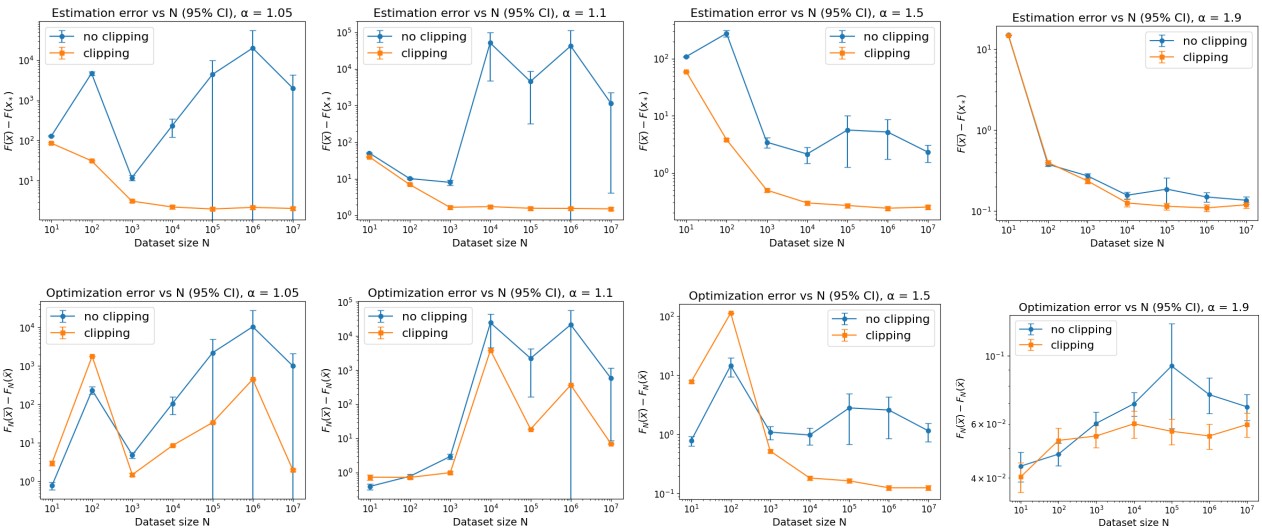

*Figure 1.* First row: population excess risk $F(\overline{x}^T) - F(x_\star)$. Second row: optimization error $F_N(\overline{x}^T) - F_N(\widehat{x})$.

2. *For $N \approx T$ the optimal choice is $\alpha_{\text{alg}} = 2$ which results in overall bound $\mathcal{O}\left(1/T^{\frac{2(\alpha_{\text{true}}-1)}{\alpha_{\text{true}}}}\right)$. The second bound is optimal.*

3. *For $N \ll T$ the optimal choice is $\alpha_{\text{alg}} = 2$ which results in overall bound $\mathcal{O}(1/T)$. Both bounds are optimal.*

It can be seen, that bound with better dependence on $T$ turns out to be worse, if we take into consideration the dataset size. To improve the bound we can tune $\lambda_0$ trying to equilibrate the tradeoff in the first bound:

$$\lambda_0 = \left(\frac{1}{N}\sum_{i=1}^{N}\|\nabla f_i(x_\star)\|^{\alpha_{\text{alg}}}\right)^{1/\alpha_{\text{alg}}}. \qquad (18)$$

Tuned $\lambda_0$ for strongly convex setup turns out to be the same as for the convex one, which allows to use it without examining the problem's convexity properties.

**Lemma 3.6.** *Let all the Assumptions for Theorem 3.7 be satisfied and the tuned $\lambda_0$ is used as defined above. Then, for 3 different regimes, based on relation between $N$ and $T$ we have the following choice of $\alpha_{\text{alg}}$:*

1. *For $N \gg T$ the optimal choice is $\alpha_{\text{alg}} = \alpha_{\text{true}}$ which results in overall bound $\mathcal{O}\left(\ln N/T^{\alpha_{\text{true}}-1}\right)$. The second bound is optimal.*

2. *For $N \approx T$ the optimal choice is $\alpha_{\text{alg}} > \alpha_{\text{true}}$ which results in overall bound $\mathcal{O}\left(1/T^{\frac{2(\alpha_{\text{true}}-1)}{\alpha_{\text{true}}}}\right)$. The first bound is optimal.*

3. *For $N \ll T$ the optimal choice is $\alpha_{\text{alg}} = 2$ which results in overall bound $\mathcal{O}(1/T)$. Both bounds are optimal.*

With tuned $\lambda_0$ we can use more aggressive clipping without worsening convergence guarantees.

We have analyzed both generalization and optimization errors under heavy-tailed noise and have shown, how the heavy-tailed stochasticity influences the ClipSGD. It turns out, that if clipping is utilized, it can be set to a wide range of options, not particularly depending on the true heavy-tail index. We have also derived, that impact of heavy-tailed noise increases with the rising dataset sizes.

## 4. Experimental validation

Since the performance of the ClipSGD has been thoroughly studied, in this section, we focus on the effect of increasing dataset sizes on the optimization process, as well as validating the assumptions on the tail indices of the square of the stochastic gradient's norm.

We validate the predicetd dependence on the number of samples. We consider function $F(x) = \frac{1}{2}\|x\|^2$ with $f(x,\xi) = \frac{1}{2}\|x\|^2 + \langle \xi, x\rangle$, where $\|\xi\|^2$ is $\mathcal{S}\beta\mathcal{S}$ with $\beta = \frac{\alpha}{2}$ and $\mathbb{E}\,\xi = 0$. Proper $\xi$ is sampled as follows: according to (Kanter, 1975) we construct a positive strictly $\frac{\alpha}{2}$ stable $\eta$ with scale $= 1$; then we sample a vector uniformly from the sphere $u \in \mathcal{U}\left(\mathcal{S}^{d-1}\right)$. We define $\xi = \sqrt{\eta} \cdot u \cdot c$, where $c$ is deterministic and stands for the scale parameter. We take $c = 10$. We consider $\alpha \in \{1.05, 1.1, 1.5, 1.9\}$, and for each $\alpha$ draw $N \in \left\{10, 10^2, 10^3, 10^4, 10^5, 10^6, 10^7\right\}$ samples from the corresponding distribution. We run SGD and ClipSGD with $\lambda_t = t^{1/\alpha}, \gamma_t = t^{-1/\alpha}/4$ for $T = 10^3$ iterations over 100 runs, with $\{\xi_i\}_{i=1}^{N}$ being fixed – the stochasticity is only in the choice of $\xi_t$ at every iteration. The obtained results are plotted in Figure 1.

It can be seen, that with the increasing dataset sizes and fixed number of iterations, optimization error $F_N(\overline{x}^T) - F_N(\widehat{x})$

is sometimes comparable between ClipSGD and SGD. However, population excess risk $F(\overline{x}^T) - F(x_\star)$ decreases for ClipSGD and increases or stabilizes for SGD, which coincides with the derived theory. With $\alpha = 1.9$ the noise is close to the finite-variance regime, therefore, SGD and ClipSGD demonstrate similar results.

## 5. Conclusion

This work analyzes ClipSGD for finite-sum minimization. We demonstrate the convergence bounds for arbitrary non-increasing step sizes and clipping radii. Our analysis shows that if the sample functions are drawn from a distribution with heavy-tailed noise, clipping is not necessary for convergence of the finite-sum method, but it can significantly improve the rates, when the number of samples is comparable to or greatly exceeds the number of iterations. Next, we demonstrate problem-aware and problem-agnostic bounds. Another theoretical contribution is the derivation of generalization bounds, which also demonstrate the impact of the heavy-tailed noise on the empirical minimization problem.

Our work develops the understanding of clipping and closes the gap between the finite-sum minimization and heavy-tailed noise. Several open questions arise from this work for future research: deriving convergence rates with high probability, connecting generalization and optimization bounds in the convex setup, considering more general smoothness assumptions.

## Impact Statement

This paper presents work whose goal is to advance the field of Machine Learning. There are many potential societal consequences of our work, none of which we feel must be specifically highlighted here.

## Acknowledgements

The research of Aleksandr Shestakov was supported by the Ministry of Economic Development of the Russian Federation (agreement No. 139-15-2025-013, dated June 20, 2025, IGK 000000C313925P4B0002).

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

# A. Generalization bounds

First, we formulate the auxiliary lemmas.

**Lemma A.1.** *Let $x \geq 0$, $u, v \geq 0$, $p \in (0, 1)$ and suppose that the following inequality holds:*

$$x \leq u + vx^p.$$

*Then, if*

$$v(2u)^{p-1} \leq \frac{1}{2},$$

*we have*

$$x \leq 2u.$$

*Proof.* If $x = 0$, then the desired inequality is satisfied. Otherwise, suppose the opposite, that $x > 2u$. Then, we obtain

$$x \leq u + vx^p < \frac{x}{2} + vx^p.$$

After rearranging the terms we have

$$\frac{x}{2} \leq vx^p.$$

Divide both sides by $x$ and use the monotonicity of $x^r$ with $r < 0$:

$$\frac{1}{2} < vx^{p-1} < v(2u)^{p-1} \leq \frac{1}{2}.$$

This is a contradiction. □

**Lemma A.2.** *Let $x \geq 0$, $u, v \geq 0$, $p \in (0, 1)$ and the following inequality is satisfied:*

$$x \leq u + vx^p.$$

*Then, we have*

$$x \leq 2u + (2v)^{\frac{1}{1-p}}.$$

*Proof.* We have either $x \leq (2v)^{\frac{1}{1-p}}$ or $vx^p \leq x/2 \Rightarrow x \leq 2u$. □

Then, we proceed with the proof of the main theorems.

**Theorem A.1.** *Let the Assumptions 3.2 to 3.5 be satisfied with tail index $\alpha \in (1, 2]$. Then for any $q \in (1, \alpha)$ and for*

$$N = \Omega\left(\left(\frac{L\Delta^{2-q}}{\mu^{q-1}(\mathbb{E}\|\nabla f(x_\star, \xi)\|^q)^{\frac{2-q}{q}}}\right)^{\frac{1}{q-1}}\right), \tag{19}$$

*we have*

$$\mathbb{E}\big[F(\widehat{x}) - F(x_\star)\big] = \mathcal{O}\left(\frac{\Delta^{2-q}\,\mathbb{E}\|\nabla f(x_\star, \xi)\|^q}{\mu^{q-1}\,N^{q-1}}\right). \tag{20}$$

*Proof.* By $\mathbb{E}_\xi$ we define the expectation across randomness $\xi \sim \mathcal{D}$. Then,

$$\mathbb{E}_\xi\left[F(\widehat{x}) - F(x_\star)\right] = \mathbb{E}_\xi\left[F(\widehat{x}) - F_N(x_\star)\right] \leq \mathbb{E}_\xi\left[F(\widehat{x}) - F_N(\widehat{x})\right].$$

As in previous analyses (Shalev-Shwartz et al., 2009; Liu & Tong, 2024) we introduce $F_N^{(j)} = \frac{1}{N}\left(f(x, \xi_j') + \sum_{i \neq j} f(x, \xi_i)\right)$, where $\xi_j'$ is an i.i.d. copy of $\xi_j$. Introduce $\widehat{x}^{(j)} = \underset{x \in \mathcal{X}}{\operatorname{argmin}}\, F_N^{(j)}(x)$. Then,

$$
\begin{aligned}
F_N(\widehat{x}^{(j)}) - F_N(\widehat{x}) &= \frac{f(\widehat{x}^{(j)}, \xi_j) - f(\widehat{x}, \xi_j)}{N} + \frac{1}{N}\sum_{i \neq j} f(\widehat{x}^{(j)}, \xi_i) - f(\widehat{x}, \xi_i) \\
&= \frac{f(\widehat{x}^{(j)}, \xi_j) - f(\widehat{x}, \xi_j)}{N} - \frac{f(\widehat{x}^{(j)}, \xi_j') - f(\widehat{x}, \xi_j')}{N} + F_N^{(j)}(\widehat{x}^{(j)}) - F_N^{(j)}(\widehat{x}) \\
&\leq \frac{f(\widehat{x}^{(j)}, \xi_j) - f(\widehat{x}, \xi_j)}{N} - \frac{f(\widehat{x}^{(j)}, \xi_j') - f(\widehat{x}, \xi_j')}{N} \\
&\leq \frac{1}{N}\left\langle \nabla f(\widehat{x}^{(j)}, \xi_j), \widehat{x}^{(j)} - \widehat{x}\right\rangle + \frac{1}{N}\left\langle \nabla f(\widehat{x}, \xi_j'), \widehat{x} - \widehat{x}^{(j)}\right\rangle.
\end{aligned}
$$

Invoking Young's inequality C.1 leads to

$$F_N(\widehat{x}^{(j)}) - F_N(\widehat{x}) \leq \frac{1}{c^q q N}\|\nabla f(\widehat{x}^{(j)}, \xi_j)\|^q + \frac{1}{c^q q N}\|\nabla f(\widehat{x}, \xi_j')\|^q + \frac{2c^{\frac{q}{q-1}}(q-1)}{qN}\|\widehat{x}^{(j)} - \widehat{x}\|^{\frac{q}{q-1}}.$$

By strong convexity of $F_N$ and the fact that $\widehat{x}$ minimizes $F_N$ we have

$$F_N(\widehat{x}^{(j)}) - F_N(\widehat{x}) \geq \left\langle \nabla F_N(\widehat{x}), \widehat{x}^{(j)} - \widehat{x} \right\rangle + \frac{\mu}{2}\|\widehat{x}^{(j)} - \widehat{x}\|^2 \geq \frac{\mu}{2}\|\widehat{x}^{(j)} - \widehat{x}\|^2,$$

where the second inequality holds according to Lemma C.3. Combining previous inequalities, we get

$$\|\widehat{x}^{(j)} - \widehat{x}\|^2 \leq \frac{2}{\mu c^q q N}\|\nabla f(\widehat{x}^{(j)}, \xi_j)\|^q + \frac{2}{\mu c^q q N}\|\nabla f(\widehat{x}, \xi_j')\|^q + \frac{4c^{\frac{q}{q-1}}(q-1)}{\mu q N}\|\widehat{x}^{(j)} - \widehat{x}\|^{\frac{q}{q-1}}.$$

From the boundedness of the set, we get

$$\|\widehat{x}^{(j)} - \widehat{x}\|^{\frac{q}{q-1}} \leq \Delta^{\frac{2-q}{q-1}}\|\widehat{x}^{(j)} - \widehat{x}\|^2 \leq \frac{2\Delta^{\frac{2-q}{q-1}}}{\mu c^q q N}\|\nabla f(\widehat{x}^{(j)}, \xi_j)\|^q + \frac{2\Delta^{\frac{2-q}{q-1}}}{\mu c^q q N}\|\nabla f(\widehat{x}, \xi_j')\|^q + \frac{4c^{\frac{q}{q-1}}\Delta^{\frac{2-q}{q-1}}(q-1)}{\mu q N}\|\widehat{x}^{(j)} - \widehat{x}\|^{\frac{q}{q-1}}.$$

$$\left(1 - \frac{4c^{\frac{q}{q-1}}\Delta^{\frac{2-q}{q-1}}(q-1)}{\mu q N}\right)\|\widehat{x}^{(j)} - \widehat{x}\|^{\frac{q}{q-1}} \leq \frac{2\Delta^{\frac{2-q}{q-1}}}{\mu c^q q N}\|\nabla f(\widehat{x}^{(j)}, \xi_j)\|^q + \frac{2\Delta^{\frac{2-q}{q-1}}}{\mu c^q q N}\|\nabla f(\widehat{x}, \xi_j')\|^q$$

After taking expectation, since $(\widehat{x}, \xi_j) \overset{d}{=} (\widehat{x}^{(j)}, \xi_j')$, we get

$$\left(1 - \frac{4c^{\frac{q}{q-1}}\Delta^{\frac{2-q}{q-1}}(q-1)}{\mu q N}\right)\mathbb{E}_{\xi,\xi'}\|\widehat{x}^{(j)} - \widehat{x}\|^{\frac{q}{q-1}} \leq \frac{4\Delta^{\frac{2-q}{q-1}}}{\mu c^q q N}\mathbb{E}_{\xi,\xi'}\|\nabla f(\widehat{x}, \xi_j')\|^q.$$

Take $c = \left(\frac{q}{8(q-1)}\right)^{\frac{q-1}{q}}\frac{(\mu N)^{\frac{q-1}{q}}}{\Delta^{\frac{2-q}{q}}}$, so the left-hand side equals to $\frac{1}{2}\mathbb{E}\|\widehat{x}^{(j)} - \widehat{x}\|^{\frac{q}{q-1}}$. Then, we have

$$\mathbb{E}_{\xi,\xi'}\|\widehat{x}^{(j)} - \widehat{x}\|^{\frac{q}{q-1}} \leq \frac{8}{q}\left(\frac{8(q-1)}{q}\right)^{q-1}\frac{\Delta^{\frac{2q-q^2}{q-1}}}{(\mu N)^q}\mathbb{E}_{\xi,\xi'}\|\nabla f(\widehat{x}, \xi_j')\|^q.$$

Since $(\widehat{x}, \xi_j') \overset{d}{=} (\widehat{x}^{(j)}, \xi_j)$, we have

$$\mathbb{E}_\xi\left[F(\widehat{x}) - F_N(\widehat{x})\right] = \mathbb{E}_{\xi,\xi'}\left[F(\widehat{x}) - F_N(\widehat{x})\right] = \mathbb{E}_{\xi,\xi'}\left[\frac{1}{N}\sum_{j=1}^N F(\widehat{x}) - f(\widehat{x}, \xi_j)\right] = \mathbb{E}_{\xi,\xi'}\left[\frac{1}{N}\sum_{j=1}^N f(\widehat{x}, \xi_j') - f(\widehat{x}, \xi_j)\right]$$

$$= \frac{1}{N}\sum_{j=1}^N \mathbb{E}_{\xi,\xi'}\left[f(\widehat{x}, \xi_j') - f(\widehat{x}^{(j)}, \xi_j')\right] \leq \frac{1}{N}\sum_{j=1}^N \mathbb{E}_{\xi,\xi'}\left\langle \nabla f(\widehat{x}, \xi_j'), \widehat{x} - \widehat{x}^{(j)}\right\rangle$$

$$\leq \frac{1}{\rho^q q N}\sum_{j=1}^N \mathbb{E}_{\xi,\xi'}\|\nabla f(\widehat{x}, \xi_j')\|^q + \frac{\rho^{\frac{q}{q-1}}(q-1)}{qN}\sum_{j=1}^N \mathbb{E}_{\xi,\xi'}\|\widehat{x}^{(j)} - \widehat{x}\|^{\frac{q}{q-1}}$$

$$\leq \frac{1}{\rho^q q N}\sum_{j=1}^N \mathbb{E}_{\xi,\xi'}\|\nabla f(\widehat{x}, \xi_j')\|^q + \frac{\rho^{\frac{q}{q-1}}(q-1)}{qN}\frac{8}{q}\left(\frac{8(q-1)}{q}\right)^{q-1}\frac{\Delta^{\frac{2q-q^2}{q-1}}}{(\mu N)^q}\sum_{j=1}^N \mathbb{E}_{\xi,\xi'}\|\nabla f(\widehat{x}, \xi_j')\|^q.$$

To minimize the right-hand side, set $\rho = \left(\frac{q-1}{xy}\right)^{\frac{q-1}{q^2}}$, where $x = q$, $y = \frac{q-1}{q}\frac{8}{q}\left(\frac{8(q-1)}{q}\right)^{q-1}\frac{\Delta^{\frac{2q-q^2}{q-1}}}{(\mu N)^q}$. Then, we obtain

$$\mathbb{E}_\xi\left[F(\widehat{x}) - F_N(\widehat{x})\right] \leq \frac{q}{(q-1)^{\frac{q-1}{q}}q^{\frac{1}{q}}}\left(\frac{8(q-1)}{q^2}\left(\frac{8(q-1)}{q}\right)^{q-1}\right)^{\frac{q-1}{q}}\frac{\Delta^{2-q}}{\mu^{q-1}N^{q-1}}\frac{1}{N}\sum_{j=1}^N \mathbb{E}_{\xi,\xi'}\|\nabla f(\widehat{x}, \xi_j')\|^q.$$

Then, we focus on the second term $\mathbb{E}_{\xi,\xi'}\|\nabla f(\widehat{x}, \xi_j')\|^q$. After using the convexity of the $\|\cdot\|^q$ we have

$$\mathbb{E}_{\xi,\xi'}\|\nabla f(\widehat{x}, \xi_j')\|^q \leq 2^{q-1}\mathbb{E}_{\xi,\xi'}\|\nabla f(\widehat{x}, \xi_j') - \nabla f(x_\star, \xi_j')\|^q + 2^{q-1}\mathbb{E}_{\xi,\xi'}\|\nabla f(x_\star, \xi_j')\|^q.$$

We focus on the first term. From smoothness and convexity we have:

$$\mathbb{E}_{\xi,\xi'}\|\nabla f(\widehat{x}, \xi_j') - \nabla f(x_\star, \xi_j')\|^q \leq \left(\mathbb{E}_{\xi,\xi'}\|\nabla f(\widehat{x}, \xi_j') - \nabla f(x_\star, \xi_j')\|^2\right)^{\frac{q}{2}} \leq \left(2L\mathbb{E}_\xi\left[F(\widehat{x}) - F(x_\star)\right]\right)^{\frac{q}{2}}.$$

Therefore, we have

$$\mathbb{E}_\xi \left[ F(\widehat{x}) - F_N(\widehat{x}) \right] \le \frac{C\Delta^{2-q}}{\mu^{q-1}N^{q-1}} \left( \mathbb{E}\|\nabla f(x_\star, \xi)\|^q + (2L\mathbb{E}_\xi \left[ F(\widehat{x}) - F_N(\widehat{x}) \right])^{\frac{q}{2}} \right),$$

where $C = \frac{2^{q-1}(8(q-1))^{q-1}}{(q-1)^{\frac{q-1}{q}} q^q}$. Then, we apply Lemma A.1 with $x = \mathbb{E}_\xi \left[ F(\widehat{x}) - F_N(\widehat{x}) \right]$, $u = \frac{C\Delta^{2-q}\mathbb{E}\|\nabla f(x_\star, \xi)\|^q}{\mu^{q-1}N^{q-1}}$, $v = \frac{C\Delta^{2-q}(2L)^{q/2}}{\mu^{q-1}N^{q-1}}$ and $p = \frac{q}{2}$. Therefore, for

$$N = \Omega \left( \left( \frac{L\Delta^{2-q}}{\mu^{q-1}(\mathbb{E}\|\nabla f(x_\star, \xi)\|^q)^{\frac{2-q}{q}}} \right)^{\frac{1}{q-1}} \right),$$

we obtain

$$\mathbb{E}_\xi \left[ F(\widehat{x}) - F_N(\widehat{x}) \right] \le \frac{2C\Delta^{2-q}\mathbb{E}\|\nabla f(x_\star, \xi)\|^q}{\mu^{q-1}N^{q-1}}.$$

$\square$

**Corollary A.2.** *If one is interested in bounds regardless of $N$ being exceeding a certain threshold, we use the Lemma A.2 with $x = \mathbb{E}_\xi \left[ F(\widehat{x}) - F_N(\widehat{x}) \right]$, $u = \frac{C\Delta^{2-q}\mathbb{E}\|\nabla f(x_\star, \xi)\|^q}{\mu^{q-1}N^{q-1}}$, $v = \frac{C\Delta^{2-q}(2L)^{q/2}}{\mu^{q-1}N^{q-1}}$, $p = \frac{q}{2}$ and obtain*

$$\mathbb{E}_\xi \left[ F(\widehat{x}) - F(x_\star) \right] = \mathcal{O} \left( \frac{\Delta^{2-q}\mathbb{E}\|\nabla f(x_\star, \xi)\|^q}{\mu^{q-1}N^{q-1}} + \frac{\Delta^2 L^{\frac{q}{2-q}}}{\mu^{\frac{2(q-1)}{2-q}} N^{\frac{2(q-1)}{2-q}}} \right). \tag{21}$$

*The second term decays faster with the growth of $N$.*

Since random variable $F(\widehat{x}) - F(x_\star)$ is nonnegative, we can apply Markov's inequality and obtain the bounds with high probability

**Corollary A.3.** *With probability at least $1 - \delta$,*

$$F(\widehat{x}) - F(x_\star) = \mathcal{O} \left( \frac{\Delta^{2-q}\mathbb{E}\|\nabla f(x_\star, \xi)\|^q}{\mu^{q-1}N^{q-1}\delta} \right).$$

We also consider the convex problem with Tikhonov regularization:

$$\min_{x \in \mathcal{X}} \left\{ F_{N,\lambda}(x) = \frac{1}{N} \sum_{i=1}^N f(x, \xi_i) + \frac{\lambda}{2}\|x - x_0\|^2 \right\}. \tag{22}$$

**Theorem A.4** (Tikhonov-regularized ERM generalization). *Let the Assumptions 3.1, 3.3, 3.4, 3.5 be satisfied with tail index $\alpha \in (1, 2]$. Consider the stochastic average approximation problem with Tikhonov regularization as in Equation (22). Then, for any $q \in (1, \alpha)$ and for*

$$N = \Omega \left( \left( \frac{(L + \lambda)\Delta^{2-q}}{\lambda^{q-1}(\mathbb{E}\|\nabla f(x_{\star,\lambda}, \xi) + \lambda(x_{\star,\lambda} - x_0)\|^q)^{\frac{2-q}{q}}} \right)^{\frac{1}{q-1}} \right), \tag{23}$$

*we have*

$$\mathbb{E}\left[ F(\widehat{x}_\lambda) - F(x_\star) \right] = \mathcal{O} \left( \frac{\Delta^{2-q}\mathbb{E}\|\nabla f(x_{\star,\lambda}, \xi) + \lambda(x_{\star,\lambda} - x_0)\|^q}{\lambda^{q-1}N^{q-1}} + \frac{\lambda\Delta^2}{2} \right), \tag{24}$$

*where $x_{\star,\lambda} = \underset{x \in \mathcal{X}}{\arg\min} \, F(x) + \frac{\lambda}{2}\|x - x_0\|^2$.*

*Proof.* Define $F_\lambda(x) = F(x) + \frac{\lambda}{2}\|x - x_0\|^2$ and $\widehat{x}_\lambda = \underset{x \in \mathcal{X}}{\arg\min} \, F_{N,\lambda}$. Then, we have

$$F(\widehat{x}_\lambda) - F(x_\star) = F(\widehat{x}_\lambda) - F_\lambda(x_{\star,\lambda}) + F_\lambda(x_{\star,\lambda}) - F(x_\star) \le F(\widehat{x}_\lambda) - F_\lambda(x_{\star,\lambda}) + F_\lambda(x_\star) - F(x_\star)$$

$$= F(\widehat{x}_\lambda) - F_\lambda(x_{\star,\lambda}) + \frac{\lambda}{2}\|x_\star - x_0\|^2 \le F(\widehat{x}_\lambda) - F_\lambda(x_{\star,\lambda}) + \frac{\lambda\Delta^2}{2}.$$

Since $F_\lambda$ is $\lambda$-strongly convex, from previous theorem we might obtain the generalization bounds. For instance, analyzing in expectation for sufficiently large $N$ we obtain

$$\mathbb{E}\left[ F(\widehat{x}_\lambda) - F(x_\star) \right] = \mathcal{O} \left( \frac{\Delta^{2-q}\mathbb{E}\|\nabla f(x_{\star,\lambda}, \xi) + \lambda(x_{\star,\lambda} - x_0)\|^q}{\lambda^{q-1}N^{q-1}} + \frac{\lambda\Delta^2}{2} \right).$$

The high-probability bounds are analogous: with probability at least $1 - \beta$ we obtain

$$F(\widehat{x}_\lambda) - F(x_\star) = \mathcal{O}\left(\frac{\Delta^{2-q}\mathbb{E}\|\nabla f(x_{\star,\lambda},\xi) + \lambda(x_{\star,\lambda} - x_0)\|^q}{\lambda^{q-1}N^{q-1}\beta} + \frac{\lambda\Delta^2}{2}\right).$$

$\square$

In order to derive the bounds for the tuned parameter $\lambda$, first, we apply the Young's inequality (Lemma C.1) to the first term:

$$\mathbb{E}\|\nabla f(x_{\star,\lambda},\xi) + \lambda(x_{\star,\lambda} - x_0)\|^q \leq 3^{q-1}\left(\mathbb{E}\|\nabla f(x_\star,\xi)\|^q + \mathbb{E}\|\nabla f(x_\star,\xi) - \nabla f(x_{\star,\lambda},\xi)\|^q + \lambda\|x_{\star,\lambda} - x_0\|^q\right) \quad (25)$$

Keeping the first term unchanged, we bound the second one using the smoothness of $f(x,\xi)$:

$$\mathbb{E}\|\nabla f(x_\star,\xi) - \nabla f(x_{\star,\lambda},\xi)\|^q \leq (2L(F(x_{\star,\lambda}) - F(x_\star)))^{q/2},$$

and then from the fact, that

$$F(x_{\star,\lambda}) + \frac{\lambda}{2}\|x_{\star,\lambda} - x_0\|^2 \leq F(x_\star) + \frac{\lambda}{2}\|x_\star - x_0\|^2$$

we obtain

$$\mathbb{E}\|\nabla f(x_\star,\xi) - \nabla f(x_{\star,\lambda},\xi)\|^q \leq \left(L\lambda\|x_\star - x_0\|^2\right)^{q/2} \leq L^{q/2}\lambda^{q/2}\Delta^q.$$

The third term in 25 is bounded as follows: $\lambda\|x_{\star,\lambda} - x_0\|^q \leq \lambda\Delta^q$. Altogether, combining these with 24 we get

$$\mathbb{E}[F(\widehat{x}_\lambda) - F(x_\star)] = \mathcal{O}\left(\frac{\Delta^{2-q}\mathbb{E}\|\nabla f(x_\star,\xi)\|}{\lambda^{q-1}N^{q-1}} + \frac{L^{q/2}\lambda^{1-q/2}\Delta^2}{N^{q-1}} + \frac{\lambda\Delta^2}{N^{q-1}} + \frac{\lambda\Delta^2}{2}\right),$$

where the leading terms are the first and the last ones. Optimizing with $\lambda$ we obtain the following:

**Corollary A.5.** *Minimizing the right hand side in the equation 24 by taking* $\lambda \sim \frac{1}{\Delta N^{\frac{q-1}{q}}}\left(\mathbb{E}\|\nabla f(x_\star,\xi)\|^q\right)^{1/q}$, *with*

$$N = \Omega\left(\left(\frac{(L+\lambda)\Delta}{(\mathbb{E}\|\nabla f(x_\star,\xi)\|^q)^{1/q}}\right)^{\frac{q}{q-1}}\right). \quad (26)$$

*we obtain*

$$\mathbb{E}[F(\widehat{x}_\lambda) - F(x_\star)] = \mathcal{O}\left(\frac{\Delta\left(\mathbb{E}\|\nabla f(x_\star,\xi)\|^q\right)^{1/q}}{N^{\frac{q-1}{q}}}\right), \quad (27)$$

*To obtain guarantees with probability at least* $1 - \beta$, *we choose* $\lambda \sim \frac{1}{\Delta\beta^{\frac{1}{q}}N^{\frac{q-1}{q}}}\left(\mathbb{E}\|\nabla f(x_\star,\xi)\|^q\right)^{1/q}$, *hence:*

$$F(\widehat{x}_\lambda) - F(x_\star) = \mathcal{O}\left(\frac{\Delta\left(\mathbb{E}\|\nabla f(x_\star,\xi)\|^q\right)^{1/q}}{\beta^{\frac{1}{q}}N^{\frac{q-1}{q}}}\right). \quad (28)$$

*Without bounds on $N$ we have*

$$\mathbb{E}[F(\widehat{x}_\lambda) - F(x_\star)] = \mathcal{O}\left(\frac{\Delta\left(\mathbb{E}\|\nabla f(x_\star,\xi)\|^q\right)^{1/q}}{N^{\frac{q-1}{q}}} + \frac{\Delta^{\frac{2}{2-q}}L^{\frac{q}{2-q}}}{\left(\mathbb{E}\|\nabla f(x_\star,\xi)\|^q\right)^{\frac{2(q-1)}{q(2-q)}}N^{\frac{2(q-1)}{q(2-q)}}}\right), \quad (29)$$

*and with probability at least* $1 - \beta$:

$$F(\widehat{x}_\lambda) - F(x_\star) = \mathcal{O}\left(\frac{\Delta\left(\mathbb{E}\|\nabla f(x_\star,\xi)\|^q\right)^{1/q}}{\beta^{\frac{1}{q}}N^{\frac{q-1}{q}}} + \frac{\Delta^{\frac{2}{2-q}}L^{\frac{q}{2-q}}\beta^{\frac{2(q-1)}{q(2-q)}}}{\left(\mathbb{E}\|\nabla f(x_\star,\xi)\|^q\right)^{\frac{2(q-1)}{q(2-q)}}N^{\frac{2(q-1)}{q(2-q)}}}\right). \quad (30)$$

To prove, that heavy-tail noise indeed influence the generalization bound we provide the lower bound:

**Lemma A.3.** *Let $\xi$ be distributed as a two-sided Pareto random variable, i.e.* $\mathbb{P}(\xi < x) \sim k|x|^{-\alpha}, x \to -\infty, \mathbb{P}(\xi > x) \sim kx^{-\alpha}, x \to \infty$. *Consider problem $F(x) = \frac{\mu}{2}x^2$ with $f(x,\xi_i) = \frac{\mu}{2}x^2 + x\xi_i$ on a bounded set $\mathcal{X} = \left[-\frac{\Delta}{2}; \frac{\Delta}{2}\right]$. Then,*

$$\mathbb{E}[F(\widehat{x}) - F(x_\star)] = \Omega\left(\frac{\Delta^{2-\alpha}}{\mu^{\alpha-1}N^{\alpha-1}}\right).$$

*Proof.* The minimum of function $F(x) = \frac{\mu}{2}x^2$ is $x_\star = 0$, whereas the minimum of $F_N(x) = \frac{\mu}{2}x^2 + \frac{1}{N}\sum\limits_{i=1}^{N}\xi_i x$ is

$$\widehat{x} = \text{proj}_{\mathcal{X}}(\widetilde{x}) = \text{proj}_{\mathcal{X}}\left(-\frac{1}{\mu N}\sum_{i=1}^{N}\xi_i\right).$$

Therefore, $F(\widehat{x}) - F(x_\star) = \frac{\mu}{2}\widehat{x}^2$. To derive the lower generalization error we consider only the case, where $\widehat{x}$ lies on the boundary of the set $\mathcal{X}$:

$$\mathbb{E}\left[F(\widehat{x}) - F(x_\star)\right] \geq \mathbb{P}(|\widetilde{x}| \geq \Delta/2) \cdot \frac{\mu\Delta^2}{8}.$$

According to Theorem 1 in (Nagaev, 1982), from the symmetry of the two-sided Pareto distribution for $c > 0$ we have

$$\mathbb{P}\left(\sum_{i=1}^{N}\xi_i \geq cN\right) \sim N\mathbb{P}(\xi \geq cN), \qquad \mathbb{P}\left(\sum_{i=1}^{N}\xi_i \leq -cN\right) \sim N\mathbb{P}(\xi \leq -cN).$$

Therefore,

$$\mathbb{P}\left(\left|\sum_{i=1}^{N}\xi_i\right| \geq cN\right) \sim N\mathbb{P}(|\xi| \geq cN). \tag{31}$$

Take $c = \frac{\mu\Delta}{2}$, then

$$\mathbb{P}\left(|\widetilde{x}| \geq \Delta/2\right) = \mathbb{P}\left(\left|\frac{1}{\mu N}\sum_{i=1}^{N}\xi_i\right| \geq \frac{\Delta}{2}\right) = \mathbb{P}\left(\left|\sum_{i=1}^{N}\xi_i\right| \geq \frac{\mu\Delta N}{2}\right) \sim N\mathbb{P}(|\xi| \geq \mu\Delta N/2) \sim \frac{1}{\mu^\alpha N^{\alpha-1}\Delta^\alpha}. \tag{32}$$

Putting this into the lower bound we obtain

$$\mathbb{E}\left[F(\widehat{x}) - F(x_\star)\right] \geq \mathbb{P}(|\widetilde{x}| \geq \Delta/2) \cdot \frac{\mu\Delta^2}{8} \sim \frac{\Delta^{2-\alpha}}{\mu^{\alpha-1}N^{\alpha-1}}. \tag{33}$$

$\square$

## B. Heavy-tailedness

In this section we analyze the effect of heavy-tailed noise for finite-sum minimization.

**Theorem B.1** (Chapter 7 Theorem 2 from (Gnedenko & Kolmogorov, 1968)). *Let the random variable $\eta$ satisfy the following conditions:*

$$\mathbb{P}(\eta < 0) = 0, \mathbb{P}(\eta > x) \sim cx^{-\alpha}, \quad \alpha \in (0, 2), \quad x > 0. \tag{34}$$

*Then, there exist sequences $\{A_i\}$ and $\{B_i\}$, such that*

$$\frac{1}{B_N} \sum_{i=1}^{N} (\eta_i - A_N) \xrightarrow{d} G_\alpha, \tag{35}$$

*where $G_\alpha$ is a strictly stable random variable with parameters $\alpha, \beta = 1$.*

It is important to notice, that the uniqueness is not claimed. The example of sequences $\{A_n\}$ and $\{B_n\}$ can be found, for example, in Table 2.1 in (Uchaikin & Zolotarev, 2011):

| $\alpha$ | $A_N$ | $B_N$ |
|---|---|---|
| $0 < \alpha < 1$ | $0$ | $(\pi c)^{1/\alpha} (2\Gamma(\alpha) \sin(\alpha\pi/2))^{-1/\alpha} N^{1/\alpha}$ |
| $\alpha = 1$ | $c \ln N$ | $\pi c N / 2$ |
| $1 < \alpha < 2$ | $\mathbb{E}\eta$ | $(\pi c)^{1/\alpha} (2\Gamma(\alpha) \sin(\alpha\pi/2))^{-1/\alpha} N^{1/\alpha}$ |

*Table 1.* Sequences $\{A_N\}$ and $\{B_N\}$, constant $c$ is from Theorem B.1, $\Gamma(\cdot)$ stands for a Gamma function.

First of all, we prove the asymptotic of the $\sum_{i=1}^{N} \|\nabla f(x_\star, \xi_i)\|^p$. If the objective function satisfied Assumption 3.5 with the tail index $\alpha \in (1, 2)$, then, $\|\nabla f(x_\star), \xi\|^p$ has the tail index $\frac{\alpha}{p}$. Using B.1, the fact, that strictly stable random variable $G_{\frac{\alpha}{p}}$ is positive and has everywhere continuous CDF, from the Portmanteau lemma and constants $A_N$ and $B_N$ we have the following:

**Theorem B.2.** *Let the Assumption 3.5 be satisfied with $\alpha < 2$. Then, for $p < \alpha$ we have*

$$\frac{1}{N} \sum_{i=1}^{N} \|\nabla f_i(x_\star)\|^p = \Theta_\mathbb{P}(1), \tag{36}$$

*for $p = \alpha$*

$$\frac{1}{N} \sum_{i=1}^{N} \|\nabla f_i(x_\star)\|^p = \Theta_\mathbb{P}(\ln N), \tag{37}$$

*and for $p > \alpha$*

$$\frac{1}{N} \sum_{i=1}^{N} \|\nabla f_i(x_\star)\|^p = \Theta_\mathbb{P}\left(N^{\frac{p}{\alpha} - 1}\right). \tag{38}$$

Then we analyze the effect of clipping on the optimization bounds. Suppose, step sizes and clipping radii are selected according to the index $\alpha_{\text{alg}}$, and objective function satisfies Assumption 3.5 with index $\alpha_{\text{true}}$. Then, the tail index of $\|\nabla f(x_\star, \xi)\|^{\alpha_{\text{alg}}}$ equals to $\frac{\alpha_{\text{true}}}{\alpha_{\text{alg}}}$. Denote by $\kappa = \frac{\alpha_{\text{true}}}{\alpha_{\text{alg}}} \in (1/2; 2)$ and by $q(N) = \sup_{x \in \mathbb{R}} |P_{S_N}(x) - P_{G_\kappa}(x)|$, where $P_\xi(\cdot)$ stands for the CDF of $\xi$ and $S_N$ for the normalized sum. Therefore, we can obtain

$$\mathbb{P}\left(\frac{1}{N} \sum_{i=1}^{N} \|\nabla f(x_\star, \xi_i)\|^{\alpha_{\text{alg}}} \leq C\right) = \mathbb{P}\left(\sum_{i=1}^{N} \|\nabla f(x_\star, \xi_i)\|^{\alpha_{\text{alg}}} \leq NC\right)$$

$$= \mathbb{P}\left(\sum_{i=1}^{N} (\|\nabla f(x_\star, \xi_i)\|^{\alpha_{\text{alg}}} - A_N(\kappa)) \leq N(C - A_N(\kappa))\right)$$

$$= \mathbb{P}\left(\frac{1}{B_N(\kappa)} \sum_{i=1}^{N} (\|\nabla f(x_\star, \xi_i)\|^{\alpha_{\text{alg}}} - A_N(\kappa)) \leq \frac{N}{B_N(\kappa)}(C - A_N(\kappa))\right)$$

$$\geq \mathbb{P}\left(G_\kappa \leq \frac{N}{B_N(\kappa)}(C - A_N(\kappa))\right) - q(N) \geq 1 - \beta,$$

where $\beta$ is from the definition. Therefore,

$$\frac{N}{B_N(\kappa)}(C - A_N(\kappa)) \geq P_{G_\kappa}^{-1}(1 - \beta + q(N)),$$

and

$$C \geq A_N(\kappa) + \frac{B_N(\kappa)}{N} P_{G_\kappa}^{-1}(1 - \beta + q(N)).$$

We want to focus on $P_{G_\kappa}^{-1}(1 - \beta + q(N))$. From Markov's inequality for any $w \in (0, \kappa)$ we can obtain $\mathbb{P}(|G_\kappa| < t) \geq 1 - \frac{\mathbb{E}|G_\kappa|^w}{t^w}$. Therefore for

$$t \geq \min_{w \in (0,\kappa)} \left(\frac{\mathbb{E}|G_\kappa|^w}{\beta - q(N)}\right)^{1/w}$$

we have $\mathbb{P}(G_\kappa < t) > \mathbb{P}(|G_\kappa| < t) > 1 - \beta + q(N)$.

Combining the above, we derive the following result:

**Theorem B.3.** *Let the Assumption 3.5 be satisfied and $N$ be great enough, so $\beta > q(N)$. Then with probability at least $1 - \beta$ the following bounds on $\frac{1}{N}\sum_{i=1}^{N}\|\nabla f(x_\star, \xi_i)\|^{\alpha_{\mathrm{alg}}}$ hold:*

*With $\alpha_{\mathrm{alg}} < \alpha_{\mathrm{true}}$*

$$\frac{1}{N}\sum_{i=1}^{N}\|\nabla f(x_\star, \xi_i)\|^{\alpha_{\mathrm{alg}}} = \mathcal{O}\left(\mathbb{E}\|\nabla f(x_\star, \xi)\|^{\alpha_{\mathrm{alg}}} + N^{\frac{\alpha_{\mathrm{alg}}}{\alpha_{\mathrm{true}}} - 1} \min_{w \in (0, \frac{\alpha_{\mathrm{true}}}{\alpha_{\mathrm{alg}}})} \left(\frac{\mathbb{E}|G_{\alpha_{\mathrm{true}}/\alpha_{\mathrm{alg}}}|^w}{\beta - q(N)}\right)^{1/w}\right). \tag{39}$$

*With $\alpha_{\mathrm{alg}} = \alpha_{\mathrm{true}}$*

$$\frac{1}{N}\sum_{i=1}^{N}\|\nabla f(x_\star, \xi_i)\|^{\alpha_{\mathrm{alg}}} = \mathcal{O}\left(\ln N + \min_{w \in (0,1)} \left(\frac{\mathbb{E}|G_1|^w}{\beta - q(N)}\right)^{1/w}\right). \tag{40}$$

*With $\alpha_{\mathrm{alg}} > \alpha_{\mathrm{true}}$*

$$\frac{1}{N}\sum_{i=1}^{N}\|\nabla f(x_\star, \xi_i)\|^{\alpha_{\mathrm{alg}}} = \mathcal{O}\left(N^{\frac{\alpha_{\mathrm{alg}}}{\alpha_{\mathrm{true}}} - 1} \min_{w \in (0, \frac{\alpha_{\mathrm{true}}}{\alpha_{\mathrm{alg}}})} \left(\frac{\mathbb{E}|G_{\alpha_{\mathrm{true}}/\alpha_{\mathrm{alg}}}|^w}{\beta - q(N)}\right)^{1/w}\right). \tag{41}$$

### B.1. Analysis of $q(N)$

In order to derive non-asymptotic bounds, we need to bound $q(N)$. The literature shows, that the tail behavior influences $q(N)$ significantly. Under the mildest conditions, when the heavy-tailed Assumption 3.5 is satisfied, we can use the results from (Juozulynas & Paulauskas, 1998; Christoph & Wolf, 1992), that demonstrate

$$q(N) \leq CN^{-\theta}(\ln N)^\tau,$$

where $\theta = I\{\alpha \neq 1/2, 0 < \alpha \leq 1\} + 2I\{\alpha = 1/2\} + \frac{2-\alpha}{\alpha}I\{1 < \alpha < 2\}, \tau = 2I(\alpha = 1)$.

In this setting, one should take $B_N = N^{1/\alpha}$ and $A_N$ as follows:

$$A_N = \begin{cases} 0, & \alpha < 1/2 \\ m, & 1/2 \leq \alpha < 2, \alpha \neq 1 \\ (m + 2\pi^{-1}\ln N), & \alpha = 1 \end{cases}$$

$$m = \begin{cases} (\pi - 2)(2\pi)^{-1}, & \alpha = 1/2 \\ 2\pi^{-1}(\gamma + 1 + \ln(\pi/2)), & \alpha = 1 \\ -\alpha((1-\alpha)\pi)^{-1}\Gamma(\alpha)|\sin(\alpha\pi)|^{1/\alpha}, & 1/2 \leq \alpha < 2, \alpha \neq 1 \end{cases}$$

This choice of $\{A_i\}$ and $\{B_i\}$ does not influence the asymptotics in Theorem B.3.

**Regularly varying tail** To derive more precise bounds on $q(N)$ we use the results from (de Haan & Peng, 1999), where they considered the regularly varying tail: $\mathbb{P}(\xi > x) \sim cx^{-\alpha}(1 + l(x))$. Therefore, if we know, that

$$\mathbb{P}(\|\nabla f(x_\star, \xi)\| > x) \sim cx^{-\alpha_{\mathrm{true}}}(1 + l(x)),$$

then,

$$\mathbb{P}(\|\nabla f(x_\star, \xi)\|^{\alpha_{\mathrm{alg}}} > x) \sim cx^{\frac{-\alpha_{\mathrm{true}}}{\alpha_{\mathrm{alg}}}}\left(1 + l(x^{\frac{1}{\alpha_{\mathrm{alg}}}})\right) = cx^{\frac{-\alpha_{\mathrm{true}}}{\alpha_{\mathrm{alg}}}}(1 + m(x)).$$

**Lemma B.1.** *If $l(x)$ is regularly varying with index $s$ and $\alpha_{\mathrm{alg}} > 0$, then, $m(x)$ will be regularly varying with index $\frac{s}{\alpha_{\mathrm{alg}}}$.*

*Proof.* From regular variation of $l$ by definition we have

$$\lim_{t\to\infty} \frac{l(tx)}{l(t)} = x^s.$$

Therefore,

$$\lim_{t\to\infty} \frac{m(tx)}{m(t)} = \lim_{t\to\infty} \frac{l((tx)^{\frac{1}{\alpha_{\mathrm{alg}}}})}{l(t^{\frac{1}{\alpha_{\mathrm{alg}}}})} = \lim_{r\to\infty} \frac{l(rx^{\frac{1}{\alpha_{\mathrm{alg}}}})}{l(r)} = x^{\frac{s}{\alpha_{\mathrm{alg}}}}.$$

$\square$

Therefore, from Theorems 1,2 and 3 from (de Haan & Peng, 1999) we can derive following asymptotics on $q(N)$:

$$q(N) \sim \begin{cases} \frac{1}{N}, & 0 < \frac{\alpha_{\mathrm{true}}}{\alpha_{\mathrm{alg}}} < 1, \ 1 < \frac{\alpha_{\mathrm{true}}}{\alpha_{\mathrm{alg}}} - \frac{s}{\alpha_{\mathrm{alg}}} < 2, \ \frac{s}{\alpha_{\mathrm{alg}}} < -\frac{\alpha_{\mathrm{true}}}{\alpha_{\mathrm{alg}}}, \\ l(N^{\frac{\alpha_{\mathrm{alg}}}{\alpha_{\mathrm{true}}}}), & 0 < \frac{\alpha_{\mathrm{true}}}{\alpha_{\mathrm{alg}}} < 1, 1 < \frac{\alpha_{\mathrm{true}}}{\alpha_{\mathrm{alg}}} - \frac{s}{\alpha_{\mathrm{alg}}} < 2, \ -\frac{\alpha_{\mathrm{true}}}{\alpha_{\mathrm{alg}}} < \frac{s}{\alpha_{\mathrm{alg}}} \\ l(N^{\frac{\alpha_{\mathrm{alg}}}{\alpha_{\mathrm{true}}}}), & \frac{\alpha_{\mathrm{true}}}{\alpha_{\mathrm{alg}}} = 1, \ -1 < \frac{s}{\alpha_{\mathrm{alg}}} < 0 \\ l(N^{\frac{\alpha_{\mathrm{alg}}}{\alpha_{\mathrm{true}}}}), & 1 < \frac{\alpha_{\mathrm{true}}}{\alpha_{\mathrm{alg}}} < 2, \frac{\alpha_{\mathrm{true}}}{\alpha_{\mathrm{alg}}} < \frac{s}{\alpha_{\mathrm{alg}}} + 2, \ \frac{s}{\alpha_{\mathrm{alg}}} \le 0. \end{cases}$$

# C. Optimization bounds

**Convex case.**

First of all, we provide the useful lemmas, that we will use throughout the proof.

**Lemma C.1** (Young's inequalities).

$$xy \leq \frac{x^p}{p} + \frac{y^q}{q}, \quad \forall x, y > 0; p, q > 1, s.t. \frac{1}{p} + \frac{1}{q} = 1,$$

$$\langle x, y \rangle \leq \frac{\rho}{2}\|x\|^2 + \frac{1}{2\rho}\|y\|^2, \quad \forall x, y \in \mathbb{R}^d, \rho > 0.$$

**Lemma C.2.** *Let $\mathcal{X}$ be a convex closed set, and $\mathrm{proj}_{\mathcal{X}}(\cdot) : \mathbb{R}^d \to \mathbb{R}^d$ be a projection operator onto this set. Then,*

$$\|\mathrm{proj}_{\mathcal{X}}(x) - \mathrm{proj}_{\mathcal{X}}(y)\| \leq \|x - y\|, \quad \forall x, y \in \mathbb{R}^d. \tag{42}$$

**Lemma C.3.** *Let $\mathcal{X}$ be a convex closed set, and $y \in \mathbb{R}^d$ be the minimum of convex differentiable function $g$ on this set. Then,*

$$\langle \nabla g(y), x - y \rangle \geq 0, \quad \forall x \in \mathcal{X}. \tag{43}$$

**Lemma C.4.** *Let $X$ be a random vector in $\mathbb{R}^d$ and $\widetilde{X} = \mathrm{clip}_{\lambda}(X)$. Then, for $\forall \alpha \in [0, 2]$*

$$\mathbb{E}\|\widetilde{X}\|^2 \leq \lambda^{2-\alpha}\mathbb{E}\|X\|^{\alpha}. \tag{44}$$

*Proof.* For $\forall t \geq 0$ we have

$$t \leq \lambda \Rightarrow t^{2-\alpha} \leq \lambda^{2-\alpha},$$

Therefore,

$$t^2 \leq \lambda^{2-\alpha}t^{\alpha}.$$

For $t \geq \lambda$ we can derive similar result, by replacing $t \leftrightarrow \lambda, \alpha \leftrightarrow 2 - \alpha$. Hence,

$$\min\{t^2, \lambda^2\} \leq \lambda^{2-\alpha}t^{\alpha}.$$

Choosing $t = \|X\|$ and taking expectation of both sides we obtain the needed. $\square$

**Lemma C.5** (Lemma C.1 from (Sadiev et al., 2023)). *Let $X$ be a random vector in $\mathbb{R}^d$ and $\widetilde{X} = \mathrm{clip}_{\lambda}(X)$. If*

$$\mathbb{E}X = x, \quad \mathbb{E}\|X - x\|^{\alpha} < \infty,$$

*for $\alpha > 1$ and $\|x\| \leq \frac{\lambda}{2}$, then,*

$$\|\mathbb{E}\widetilde{X} - x\| \leq \frac{2^{\alpha}}{\lambda^{\alpha-1}}\mathbb{E}\|X - x\|^{\alpha}. \tag{45}$$

**Theorem C.1** (3.2). *Let the Assumptions 3.1, 3.3 and 3.4 be satisfied. Additionally, assume the step size sequence to be non-increasing $\gamma_{t+1} \leq \gamma_t$, $\gamma_t \leq \frac{1}{4L}$, and $\lambda_t \geq 2^5 3^2 L\Delta + 2\|\nabla F_N(x^1)\|$. Then, for $\forall \alpha \in (1, 2]$ the iterates of the ClipSGD satisfy*

$$\mathbb{E}_{\mathrm{alg}}\left[F_N(\overline{x}^T) - F_N(\widehat{x})\right] = \mathcal{O}\left(\frac{\Delta^2}{\gamma_T T} + \frac{\mathbb{E}_i\|\nabla f_i(x_\star)\|^{\alpha} + L^{\alpha}\Delta^{\alpha}}{T}\sum_{t=1}^{T}\gamma_t\lambda_t^{2-\alpha} + \frac{\Delta\mathbb{E}_i\|\nabla f_i(x_\star)\|^{\alpha} + L^{\alpha}\Delta^{\alpha+1}}{T}\sum_{t=1}^{T}\lambda_t^{1-\alpha}\right.$$

$$\left. + I(\alpha \neq 2)\frac{\Delta^{\frac{2}{2-\alpha}}L^{\frac{\alpha}{2-\alpha}}}{T}\sum_{t=1}^{T}\lambda_t^{\frac{2-2\alpha}{2-\alpha}}\right), \tag{46}$$

*where $\overline{x}^T = \frac{1}{T}\sum_{t=1}^{T}x^t$, $\mathbb{E}_{\mathrm{alg}}$ is the expectation with respect to the random indices sampled during the algorithm, conditionally on the dataset, and $\mathbb{E}_i$ denotes the uniform average over the fixed dataset.*

*Proof.* First of all, we show, that the projection does not affect the convergence:

$$\|x^{t+1} - \widehat{x}\| = \|\mathrm{proj}_{\mathcal{X}}(x^t - \gamma_t\mathrm{clip}_{\lambda_t}(\nabla f_i(x^t))) - \widehat{x}\| = \|\mathrm{proj}_{\mathcal{X}}(x^t - \gamma_t\mathrm{clip}_{\lambda_t}(\nabla f_i(x^t))) - \mathrm{proj}_{\mathcal{X}}(\widehat{x})\|$$

$$\leq \|x^t - \gamma_t\mathrm{clip}_{\lambda_t}(\nabla f_i(x^t)) - \widehat{x}\|.$$

Then,

$$\|x^{t+1} - \widehat{x}\|^2 \le \|x^t - \widehat{x}\|^2 - 2\gamma_t \left\langle \text{clip}_{\lambda_t} \left( \nabla f_i(x^t) \right), x^t - \widehat{x} \right\rangle + \gamma_t^2 \|\text{clip}_{\lambda_t} \left( \nabla f_i(x^t) \right)\|^2$$

$$= \|x^t - \widehat{x}\|^2 - 2\gamma_t \left\langle \text{clip}_{\lambda_t} \left( \nabla f_i(x^t) \right), x^t - \widehat{x} \right\rangle + \gamma_t^2 \|\text{clip}_{\lambda_t} \left( \nabla f_i(x^t) \right) \pm \text{clip}_{\lambda_t} \left( \nabla f_i(\widehat{x}) \right)\|^2$$

$$\le \|x^t - \widehat{x}\|^2 - 2\gamma_t \left\langle \text{clip}_{\lambda_t} \left( \nabla f_i(x^t) \right), x^t - \widehat{x} \right\rangle + 2\gamma_t^2 \|\text{clip}_{\lambda_t} \left( \nabla f_i(x^t) \right) - \text{clip}_{\lambda_t} \left( \nabla f_i(\widehat{x}) \right)\|^2 + 2\gamma_t^2 \|\text{clip}_{\lambda_t} \left( \nabla f_i(\widehat{x}) \right)\|^2$$

$$\le \|x^t - \widehat{x}\|^2 - 2\gamma_t \left\langle \text{clip}_{\lambda_t} \left( \nabla f_i(x^t) \right), x^t - \widehat{x} \right\rangle + 2\gamma_t^2 \|\nabla f_i(x^t) - \nabla f_i(\widehat{x})\|^2 + 2\gamma_t^2 \|\text{clip}_{\lambda_t} \left( \nabla f_i(\widehat{x}) \right)\|^2$$

$$\le \|x^t - \widehat{x}\|^2 - 2\gamma_t \left( F_N(x^t) - F_N(\widehat{x}) \right) - 2\gamma_t \left\langle \text{clip}_{\lambda_t} \left( \nabla f_i(x^t) \right) - \nabla F_N(x^t), x^t - \widehat{x} \right\rangle + 2\gamma_t^2 \|\nabla f_i(x^t) - \nabla f_i(\widehat{x})\|^2$$

$$+ 2\gamma_t^2 \|\text{clip}_{\lambda_t} \left( \nabla f_i(\widehat{x}) \right)\|^2, \tag{47}$$

where in the last step we used the convexity of $F_N : F_N(x^t) - F_N(\widehat{x}) \le \langle \nabla F_N(x^t), x^t - \widehat{x} \rangle$. After taking conditional expectation over index $i$ we need to analyze three terms: $-2\gamma_t \mathbb{E}_i \left\langle \text{clip}_{\lambda_t} \left( \nabla f_i(x^t) \right) - \nabla F_N(x^t), x^t - \widehat{x} \right\rangle, 2\gamma_t^2 \mathbb{E}_i \|\nabla f_i(x^t) - \nabla f_i(\widehat{x})\|^2$ and $2\gamma_t^2 \mathbb{E}_i \|\text{clip}_{\lambda_t} \left( \nabla f_i(\widehat{x}) \right)\|^2$.

From smoothness we have

$$\|\nabla F_N(x^t)\| \le \|\nabla F_N(x^t) - \nabla F_N(x^1)\| + \|\nabla F_N(x^1)\| \le L\Delta + \|\nabla F_N(x^1)\|,$$

therefore for $\lambda_t \ge 2L\Delta + 2\|\nabla F_N(x^1)\|$ we bound the first term as follows:

$$-2\gamma_t \mathbb{E}_i \left\langle \text{clip}_{\lambda_t} \left( \nabla f_i(x^t) \right) - \nabla F_N(x^t), x^t - \widehat{x} \right\rangle = -2\gamma_t \left\langle \mathbb{E}_i \text{clip}_{\lambda_t} \left( \nabla f_i(x^t) \right) - \nabla F_N(x^t), x^t - \widehat{x} \right\rangle$$

$$\le 2\gamma_t \|\mathbb{E}_i \text{clip}_{\lambda_t} \left( \nabla f_i(x^t) \right) - \nabla F_N(x^t)\| \cdot \|x^t - \widehat{x}\| \le 2\gamma_t \|\mathbb{E}_i \text{clip}_{\lambda_t} \left( \nabla f_i(x^t) \right) - \nabla F_N(x^t)\| \cdot \Delta$$

$$\le \frac{2^{\alpha+1} \gamma_t \Delta}{\lambda_t^{\alpha-1}} \mathbb{E}_i \|\nabla f_i(x^t) - \nabla F_N(x^t)\|^\alpha \le \frac{2^{\alpha+1} 3^{\alpha-1} \gamma_t \Delta}{\lambda_t^{\alpha-1}} \left( \mathbb{E}_i \|\nabla f_i(x^t) - \nabla f_i(\widehat{x})\|^\alpha + \mathbb{E}_i \|\nabla f_i(\widehat{x})\|^\alpha + \|\nabla F_N(x^t)\|^\alpha \right)$$

$$\le \frac{2^{\alpha+1} 3^{\alpha-1} \gamma_t \Delta}{\lambda_t^{\alpha-1}} \left( \mathbb{E}_i \|\nabla f_i(x^t) - \nabla f_i(\widehat{x})\|^\alpha + \mathbb{E}_i \|\nabla f_i(\widehat{x})\|^\alpha + 2^{\alpha-1} \|\nabla F_N(x^t) - \nabla F_N(\widehat{x})\|^\alpha + 2^{\alpha-1} \|\nabla F_N(\widehat{x})\|^\alpha \right), \tag{48}$$

where the first inequality holds due to the Cauchy-Schwarz inequality, the second is due to the Lemma C.5, and the last two due to the Jensen's inequality, as $\| \cdot \|^\alpha$ is convex function.

From Hölder's inequality, for random variable $\eta$ we have

$$\mathbb{E}\|\eta\|^\alpha \le \left( \mathbb{E}\|\eta\|^2 \right)^{\frac{\alpha}{2}},$$

Therefore, first and third terms in (48) can be bounded as follows:

$$\mathbb{E}_i \|\nabla f_i(x^t) - \nabla f_i(\widehat{x})\|^\alpha \quad \le \quad \left( \mathbb{E}_i \|\nabla f_i(x^t) - \nabla f_i(\widehat{x})\|^2 \right)^{\frac{\alpha}{2}} \le \left( 2L \left( F_N(x^t) - F_N(\widehat{x}) \right) \right)^{\frac{\alpha}{2}}.$$

$$\|\nabla F_N(x^t) - \nabla F_N(\widehat{x})\|^\alpha \quad = \quad \left( \|\nabla F_N(x^t) - \nabla F_N(\widehat{x})\|^2 \right)^{\frac{\alpha}{2}} \le \left( 2L \left( F_N(x^t) - F_N(\widehat{x}) \right) \right)^{\frac{\alpha}{2}}.$$

The last term in (48) is bounded via the convexity of $\| \cdot \|^\alpha$:

$$\|\nabla F_N(\widehat{x})\|^\alpha = \left\| \frac{1}{N} \sum_{i=1}^N \nabla f(\widehat{x}, \xi_i) \right\|^\alpha \le \frac{1}{N} \sum_{i=1}^N \|\nabla f(\widehat{x}, \xi_i)\|^\alpha.$$

Combining inequalities above we obtain

$$-2\gamma_t \mathbb{E}_i \left\langle \text{clip}_{\lambda_t} \left( \nabla f_i(x^t) \right) - \nabla F_N(x^t), x^t - \widehat{x} \right\rangle \le \frac{2^{\alpha+1} 3^{\alpha-1} \gamma_t \Delta}{\lambda_t^{\alpha-1}} \left( 3(2L)^{\frac{\alpha}{2}} (F_N(x^t) - F_N(\widehat{x}))^{\frac{\alpha}{2}} + 3\mathbb{E}_i \|\nabla f_i(\widehat{x})\|^\alpha \right). \tag{49}$$

For $\alpha < 2$ from applying Young's inequality, we gain

$$\frac{2^{\frac{3\alpha}{2}+1} 3^\alpha L^{\frac{\alpha}{2}} \Delta}{\lambda_t^{\alpha-1}} \gamma_t (F_N(x^t) - F_N(\widehat{x}))^{\frac{\alpha}{2}} = \frac{C}{\lambda_t^{\alpha-1}} \gamma_t (F_N(x^t) - F_N(\widehat{x}))^{\frac{\alpha}{2}} = \frac{C}{\lambda_t^{\alpha-1}} \alpha^{\frac{\alpha}{2}} \gamma_t^{\frac{2-\alpha}{2}} \frac{\gamma_t^{\frac{\alpha}{2}}}{\alpha^{\frac{\alpha}{2}}} (F_N(x^t) - F_N(\widehat{x}))^{\frac{\alpha}{2}}$$

$$\overset{C.1}{\le} \frac{2-\alpha}{2} \left( \frac{C\alpha^{\frac{\alpha}{2}} \gamma_t^{\frac{2-\alpha}{2}}}{\lambda_t^{\alpha-1}} \right)^{\frac{2}{2-\alpha}} + \frac{\alpha}{2} \frac{\gamma_t}{\alpha} (F_N(x^t) - F_N(\widehat{x})) = \left( 1 - \frac{\alpha}{2} \right) \frac{C^{\frac{2}{2-\alpha}} \alpha^{\frac{\alpha}{2-\alpha}}}{\lambda_t^{\frac{2\alpha-2}{2-\alpha}}} \gamma_t + \frac{\gamma_t}{2} (F_N(x^t) - F_N(\widehat{x})),$$

where $C = 2^{\frac{3\alpha}{2}+1} 3^\alpha L^{\frac{\alpha}{2}} \Delta$.

For $\alpha = 2$, the term above becomes linear in the objective gap:

$$\frac{2^4 3^2 L \Delta}{\lambda_t} \gamma_t \left( F_N(x^t) - F_N(\widehat{x}) \right).$$

By the definition we have $\lambda_t \geq 2^5 3^2 L \Delta$. Therefore,

$$\frac{2^4 3^2 L \Delta}{\lambda_t} \gamma_t \left( F_N(x^t) - F_N(\widehat{x}) \right) \leq \frac{\gamma_t}{2} \left( F_N(x^t) - F_N(\widehat{x}) \right).$$

The second term in (47) is bounded as follows:

$$2\gamma_t^2 \mathbb{E}_i \|\nabla f_i(x^t) - \nabla f_i(\widehat{x})\|^2 \leq 4\gamma_t^2 L \left( F_N(x^t) - F_N(\widehat{x}) \right). \tag{50}$$

For $\gamma_t \leq \frac{1}{4L}$ we have

$$-2\gamma_t(F_N(x^t) - F_N(\widehat{x})) + 4\gamma_t^2 L(F_N(x^t) - F_N(\widehat{x})) \leq -\gamma_t(F_N(x^t) - F_N(\widehat{x})).$$

The third term in 47 is bounded as follows:

$$2\gamma_t^2 \mathbb{E}_i \|\mathrm{clip}_{\lambda_t} \left( \nabla f_i(\widehat{x}) \right)\|^2 \overset{C.4}{\leq} 2\gamma_t^2 \lambda_t^{2-\alpha} \mathbb{E}_i \|\nabla f_i(\widehat{x})\|^\alpha. \tag{51}$$

Plugging (49), (50) and (51) in (47), we obtain:

$$\mathbb{E}_i \|x^{t+1} - \widehat{x}\|^2 \leq \|x^t - \widehat{x}\|^2 - \frac{\gamma_t}{2} \left( F_N(x^t) - F_N(\widehat{x}) \right) + \frac{C_1}{\lambda_t^{\alpha-1}} \gamma_t \mathbb{E}_i \|\nabla f_i(\widehat{x})\|^\alpha + 2\gamma_t^2 \lambda_t^{2-\alpha} \mathbb{E}_i \|\nabla f_i(\widehat{x})\|^\alpha + I(\alpha \neq 2) \frac{C_2}{\lambda_t^{\frac{2\alpha-2}{2-\alpha}}} \gamma_t.$$

Define $\delta_t = \mathbb{E}_{\mathrm{alg}} \|x^t - \widehat{x}\|^2$. Then, after summing over the iterations, taking full expectation with respect to the randomness of the method and applying the convexity, we obtain

$$\mathbb{E}_{\mathrm{alg}} \left[ F_N(\bar{x}^T) - F_N(\widehat{x}) \right] \leq \frac{2}{T} \sum_{t=1}^T \frac{1}{\gamma_t} (\delta_t - \delta_{t+1}) + \frac{2\mathbb{E}_i \|\nabla f_i(\widehat{x})\|^\alpha}{T} \sum_{t=1}^T \lambda_t^{2-\alpha} \gamma_t + \frac{C_1 \mathbb{E}_i \|\nabla f_i(\widehat{x})\|^\alpha}{T} \sum_{t=1}^T \lambda_t^{1-\alpha}$$

$$+ I(\alpha \neq 2) \frac{C_2}{T} \sum_{t=1}^T \lambda_t^{\frac{2-2\alpha}{2-\alpha}}$$

$$= \frac{2}{T} \sum_{t=2}^T \delta_t \left( \frac{1}{\gamma_t} - \frac{1}{\gamma_{t-1}} \right) + \frac{2\delta_1}{\gamma_1 T} - \frac{2\delta_{T+1}}{\gamma_T T} + \frac{2\mathbb{E}_i \|\nabla f_i(\widehat{x})\|^\alpha}{T} \sum_{t=1}^T \lambda_t^{2-\alpha} \gamma_t + \frac{C_1 \mathbb{E}_i \|\nabla f_i(\widehat{x})\|^\alpha}{T} \sum_{t=1}^T \lambda_t^{1-\alpha} + I(\alpha \neq 2) \frac{C_2}{T} \sum_{t=1}^T \lambda_t^{\frac{2-2\alpha}{2-\alpha}}$$

$$\leq \frac{2}{T} \sum_{t=2}^T \delta_t \left( \frac{1}{\gamma_t} - \frac{1}{\gamma_{t-1}} \right) + \frac{2\delta_1}{\gamma_1 T} + \frac{2\mathbb{E}_i \|\nabla f_i(\widehat{x})\|^\alpha}{T} \sum_{t=1}^T \lambda_t^{2-\alpha} \gamma_t + \frac{C_1 \mathbb{E}_i \|\nabla f_i(\widehat{x})\|^\alpha}{T} \sum_{t=1}^T \lambda_t^{1-\alpha} + I(\alpha \neq 2) \frac{C_2}{T} \sum_{t=1}^T \lambda_t^{\frac{2-2\alpha}{2-\alpha}}$$

$$\leq \frac{2}{T} \sum_{t=2}^T \delta_t \left( \frac{1}{\gamma_t} - \frac{1}{\gamma_{t-1}} \right) + \frac{2\delta_1}{\gamma_1 T} + \frac{2^\alpha \left( \mathbb{E}_i \|\nabla f_i(x_\star)\|^\alpha + \mathbb{E}_i \|\nabla f_i(\widehat{x}) - \nabla f_i(x_\star)\|^\alpha \right)}{T} \sum_{t=1}^T \lambda_t^{2-\alpha} \gamma_t$$

$$+ \frac{2^{\alpha-1} C_1 \left( \mathbb{E}_i \|\nabla f_i(x_\star)\|^\alpha + \mathbb{E}_i \|\nabla f_i(\widehat{x}) - \nabla f_i(x_\star)\|^\alpha \right)}{T} \sum_{t=1}^T \lambda_t^{1-\alpha} + I(\alpha \neq 2) \frac{C_2}{T} \sum_{t=1}^T \lambda_t^{\frac{2-2\alpha}{2-\alpha}}$$

$$\leq \frac{2}{T} \sum_{t=2}^T \delta_t \left( \frac{1}{\gamma_t} - \frac{1}{\gamma_{t-1}} \right) + \frac{2\delta_1}{\gamma_1 T} + \frac{2^\alpha \left( \mathbb{E}_i \|\nabla f_i(x_\star)\|^\alpha + L^\alpha \Delta^\alpha \right)}{T} \sum_{t=1}^T \lambda_t^{2-\alpha} \gamma_t$$

$$+ \frac{2^{\alpha-1} C_1 \left( \mathbb{E}_i \|\nabla f_i(x_\star)\|^\alpha + L^\alpha \Delta^\alpha \right)}{T} \sum_{t=1}^T \lambda_t^{1-\alpha} + I(\alpha \neq 2) \frac{C_2}{T} \sum_{t=1}^T \lambda_t^{\frac{2-2\alpha}{2-\alpha}}.$$

If $\gamma_{t+1} \leq \gamma_t$, then,

$$\mathbb{E}_{\text{alg}}\left[F_N(\bar{x}^T) - F_N(\widehat{x})\right] \leq$$

$$\leq \frac{2}{T} \sum_{t=2}^{T} \Delta^2 \left(\frac{1}{\gamma_t} - \frac{1}{\gamma_{t-1}}\right) + \frac{2\Delta^2}{\gamma_1 T} + \frac{2^{\alpha}\left(\mathbb{E}_i \|\nabla f_i(x_\star)\|^{\alpha} + L^{\alpha}\Delta^{\alpha}\right)}{T} \sum_{t=1}^{T} \lambda_t^{2-\alpha} \gamma_t$$

$$+\frac{2^{\alpha-1} C_1 \left(\mathbb{E}_i \|\nabla f_i(x_\star)\|^{\alpha} + L^{\alpha}\Delta^{\alpha}\right)}{T} \sum_{t=1}^{T} \lambda_t^{1-\alpha} + I(\alpha \neq 2)\frac{C_2}{T} \sum_{t=1}^{T} \lambda_t^{\frac{2-2\alpha}{2-\alpha}}$$

$$\leq \frac{2\Delta^2}{\gamma_T T} + \frac{2^{\alpha}\left(\mathbb{E}_i \|\nabla f_i(x_\star)\|^{\alpha} + L^{\alpha}\Delta^{\alpha}\right)}{T} \sum_{t=1}^{T} \lambda_t^{2-\alpha} \gamma_t + \frac{2^{\alpha-1} C_1 \left(\mathbb{E}_i \|\nabla f_i(x_\star)\|^{\alpha} + L^{\alpha}\Delta^{\alpha}\right)}{T} \sum_{t=1}^{T} \lambda_t^{1-\alpha} + I(\alpha \neq 2)\frac{C_2}{T} \sum_{t=1}^{T} \lambda_t^{\frac{2-2\alpha}{2-\alpha}}.$$

After inserting all the constants $C_1$ and $C_2$ we obtain the desired bound. $\qquad\square$

**Corollary C.2.** *Under the Assumptions of Theorem C.1, let the step sizes and clipping radii be chosen as* $\gamma_t = \gamma_0 t^{-1/\alpha}$, $\lambda_t = \lambda_0 t^{1/\alpha}$, *where* $\gamma_0 \leq \frac{1}{4L}$ *and* $\lambda_0 \geq 2^5 3^2 L\Delta + 2\|\nabla F_N(x^1)\|$. *Then, for* $\alpha \in (1, 2)$, *we have*

$$\mathbb{E}_{\text{alg}}\left[F_N(\bar{x}^T) - F_N(\widehat{x})\right]$$

$$= \mathcal{O}\left(\frac{\Delta^2}{\gamma_0 T^{\frac{\alpha-1}{\alpha}}} + \frac{\gamma_0 \lambda_0^{2-\alpha}\left(\mathbb{E}_i\|\nabla f_i(x_\star)\|^{\alpha} + L^{\alpha}\Delta^{\alpha}\right)}{T^{\frac{\alpha-1}{\alpha}}} + \frac{\lambda_0^{1-\alpha}\left(\Delta\mathbb{E}_i\|\nabla f_i(x_\star)\|^{\alpha} + L^{\alpha}\Delta^{\alpha+1}\right)}{T^{\frac{\alpha-1}{\alpha}}} + \Delta^{\frac{2}{2-\alpha}} L^{\frac{\alpha}{2-\alpha}} \lambda_0^{\frac{2-2\alpha}{2-\alpha}} R_\alpha(T)\right),$$

$$(52)$$

*where*

$$R_\alpha(T) := \begin{cases} T^{-\frac{2(\alpha-1)}{\alpha(2-\alpha)}}, & 1 < \alpha < \sqrt{2}, \\[2mm] \dfrac{\log(eT)}{T}, & \alpha = \sqrt{2}, \\[2mm] \dfrac{1}{T}, & \sqrt{2} < \alpha < 2. \end{cases}$$

*For* $\alpha = 2$, *the last term is absent and the bound becomes*

$$\mathbb{E}_{\text{alg}}\left[F_N(\bar{x}^T) - F_N(\widehat{x})\right] = \mathcal{O}\left(\frac{\Delta^2}{\gamma_0\sqrt{T}} + \frac{\gamma_0\left(\mathbb{E}_i\|\nabla f_i(x_\star)\|^2 + L^2\Delta^2\right)}{\sqrt{T}} + \frac{\lambda_0^{-1}\left(\Delta\mathbb{E}_i\|\nabla f_i(x_\star)\|^2 + L^2\Delta^3\right)}{\sqrt{T}}\right). \qquad (53)$$

*Proof.* We substitute $\gamma_t = \gamma_0 t^{-1/\alpha}$ and $\lambda_t = \lambda_0 t^{1/\alpha}$ into Theorem C.1. First,

$$\frac{\Delta^2}{\gamma_T T} = \frac{\Delta^2}{\gamma_0 T^{\frac{\alpha-1}{\alpha}}}.$$

Next,

$$\frac{1}{T} \sum_{t=1}^{T} \gamma_t \lambda_t^{2-\alpha} = \frac{\gamma_0 \lambda_0^{2-\alpha}}{T} \sum_{t=1}^{T} t^{-\frac{1}{\alpha}} t^{\frac{2-\alpha}{\alpha}} = \frac{\gamma_0 \lambda_0^{2-\alpha}}{T} \sum_{t=1}^{T} t^{\frac{1-\alpha}{\alpha}} = \mathcal{O}\left(\frac{\gamma_0 \lambda_0^{2-\alpha}}{T^{\frac{\alpha-1}{\alpha}}}\right),$$

and similarly,

$$\frac{1}{T} \sum_{t=1}^{T} \lambda_t^{1-\alpha} = \frac{\lambda_0^{1-\alpha}}{T} \sum_{t=1}^{T} t^{\frac{1-\alpha}{\alpha}} = \mathcal{O}\left(\frac{\lambda_0^{1-\alpha}}{T^{\frac{\alpha-1}{\alpha}}}\right).$$

It remains to handle the last term, which appears only for $\alpha < 2$:

$$\frac{1}{T} \sum_{t=1}^{T} \lambda_t^{\frac{2-2\alpha}{2-\alpha}} = \frac{\lambda_0^{\frac{2-2\alpha}{2-\alpha}}}{T} \sum_{t=1}^{T} t^{-\frac{2(\alpha-1)}{\alpha(2-\alpha)}}.$$

The last sum gives the three regimes in the definition of $R_\alpha(T)$: if $\frac{2(\alpha-1)}{\alpha(2-\alpha)} < 1$, i.e. $\alpha < \sqrt{2}$, the sum is polynomial; if $\alpha = \sqrt{2}$, it is logarithmic; and if $\alpha > \sqrt{2}$, it is bounded by a constant. This proves the claim. Note, that for any $\alpha \in (1, 2)$ the last term decays faster than the previous three. $\qquad\square$

**Corollary C.3.** *Under the Assumptions of Theorem C.1, let the step sizes and clipping radii be chosen as*

$$\gamma_t \equiv \bar{\gamma}_T := \min\left\{\frac{1}{4L}, \frac{\gamma_0}{T^{1/\alpha}}\right\}, \qquad \lambda_t \equiv \bar{\lambda}_T := \max\left\{2^5 3^2 L\Delta + 2\|\nabla F_N(x^1)\|, \lambda_0 T^{1/\alpha}\right\},$$

*where $\gamma_0, \lambda_0 > 0$. Then,*

$$\mathbb{E}_{\mathrm{alg}}\left[F_N(\bar{x}^T) - F_N(\hat{x})\right]$$

$$= \mathcal{O}\left(\frac{\Delta^2}{\bar{\gamma}_T T} + \bar{\gamma}_T \bar{\lambda}_T^{2-\alpha} \left(\mathbb{E}_i\|\nabla f_i(x_\star)\|^\alpha + L^\alpha \Delta^\alpha\right) + \bar{\lambda}_T^{1-\alpha}\left(\Delta \mathbb{E}_i\|\nabla f_i(x_\star)\|^\alpha + L^\alpha \Delta^{\alpha+1}\right)\right.$$

$$\left. + I(\alpha \neq 2)\Delta^{\frac{2}{2-\alpha}} L^{\frac{\alpha}{2-\alpha}} \bar{\lambda}_T^{\frac{2-2\alpha}{2-\alpha}}\right). \tag{54}$$

*In particular, for all $\alpha \in (1,2]$ this implies the simplified rate*

$$\mathbb{E}_{\mathrm{alg}}\left[F_N(\bar{x}^T) - F_N(\hat{x})\right]$$

$$= \mathcal{O}\left(\frac{(L + \gamma_0^{-1})\Delta^2}{T^{\frac{\alpha-1}{\alpha}}} + \frac{\gamma_0\left(L^{2-\alpha}\Delta^{2-\alpha} + \|\nabla F_N(x^1)\|^{2-\alpha} + \lambda_0^{2-\alpha}\right)\left(\mathbb{E}_i\|\nabla f_i(x_\star)\|^\alpha + L^\alpha \Delta^\alpha\right)}{T^{\frac{\alpha-1}{\alpha}}}\right.$$

$$\left. + \frac{\lambda_0^{1-\alpha}\left(\Delta \mathbb{E}_i\|\nabla f_i(x_\star)\|^\alpha + L^\alpha \Delta^{\alpha+1}\right)}{T^{\frac{\alpha-1}{\alpha}}} + I(\alpha \neq 2)\frac{\Delta^{\frac{2}{2-\alpha}} L^{\frac{\alpha}{2-\alpha}} \lambda_0^{\frac{2-2\alpha}{2-\alpha}}}{T^{\frac{\alpha-1}{\alpha}}}\right). \tag{55}$$

*Proof.* Since the schedules are constant in $t$, Theorem C.1 gives

$$\mathbb{E}_{\mathrm{alg}}\left[F_N(\bar{x}^T) - F_N(\hat{x})\right]$$

$$= \mathcal{O}\left(\frac{\Delta^2}{\bar{\gamma}_T T} + \bar{\gamma}_T \bar{\lambda}_T^{2-\alpha} \left(\mathbb{E}_i\|\nabla f_i(x_\star)\|^\alpha + L^\alpha \Delta^\alpha\right) + \bar{\lambda}_T^{1-\alpha}\left(\Delta \mathbb{E}_i\|\nabla f_i(x_\star)\|^\alpha + L^\alpha \Delta^{\alpha+1}\right)\right.$$

$$\left. + I(\alpha \neq 2)\Delta^{\frac{2}{2-\alpha}} L^{\frac{\alpha}{2-\alpha}} \bar{\lambda}_T^{\frac{2-2\alpha}{2-\alpha}}\right),$$

which proves the first part.

For the simplified bound, we use

$$\frac{1}{\bar{\gamma}_T} \leq 4L + \frac{T^{1/\alpha}}{\gamma_0}, \qquad \bar{\gamma}_T \leq \frac{\gamma_0}{T^{1/\alpha}},$$

and, since $2 - \alpha \geq 0$,

$$\bar{\lambda}_T^{2-\alpha} \leq \left(2^5 3^2 L\Delta + 2\|\nabla F_N(x^1)\|\right)^{2-\alpha} + \lambda_0^{2-\alpha} T^{\frac{2-\alpha}{\alpha}}.$$

Moreover, since $1 - \alpha < 0$,

$$\bar{\lambda}_T^{1-\alpha} \leq \lambda_0^{1-\alpha} T^{\frac{1-\alpha}{\alpha}}.$$

Finally, for $\alpha < 2$,

$$\bar{\lambda}_T^{\frac{2-2\alpha}{2-\alpha}} \leq \lambda_0^{\frac{2-2\alpha}{2-\alpha}} T^{-\frac{2(\alpha-1)}{\alpha(2-\alpha)}} \leq \lambda_0^{\frac{2-2\alpha}{2-\alpha}} T^{-\frac{\alpha-1}{\alpha}}.$$

Combining these estimates proves the second bound. $\qquad\square$

**Strongly convex case.**

Now that we have obtained the bounds for the convex scenario, we proceed with the strongly convex one. We provide two different rates, that differ in their dependence on $T$ and $\mathbb{E}_i\|\nabla f_i(x_\star)\|$. We further point out, that optimal bounds in terms of the number of iterations $T$ may be suboptimal, if we take into account influence of the sample size.

**Theorem C.4.** *Let the Assumptions 3.2, 3.3 and 3.4 be satisfied. Then, for any $\alpha \in (1,2]$ choosing $\gamma_t = \frac{4}{\mu(t+16L/\mu)}$, $\lambda_t = \lambda_0 t^{1/\alpha}$ with $\lambda_0 \geq \max\{2^5 3^2 L\Delta + 2\|\nabla F_N(x^1)\|, \left(2^{3\alpha+5} 3^{2\alpha} L^\alpha(\Delta\|\nabla F_N(x^1)\| + L\Delta^2)^{\alpha-1}/\mu\right)^{1/(2(\alpha-1))}\}$ results in the following convergence bound for ClipSGD:*

$$\mathbb{E}_{\mathrm{alg}}\left[F_N(\tilde{x}^T) - F_N(\hat{x})\right] = \mathcal{O}\left(\frac{L^2\Delta^2}{\mu T^2} + \frac{\left(\mathbb{E}_i\|\nabla f_i(\hat{x})\|^\alpha\right)^2 \lambda_0^{2(1-\alpha)}}{\mu T^{\frac{2(\alpha-1)}{\alpha}}} + \frac{\mathbb{E}_i\|\nabla f_i(\hat{x})\|^\alpha \lambda_0^{2-\alpha}}{\mu T^{\frac{2(\alpha-1)}{\alpha}}}\right), \tag{56}$$

*where $\tilde{x}^T = \frac{1}{W_T}\sum_{t=1}^{T} w_t x^t$, $W_T = \sum_{t=1}^{T} w_t$, $w_t = t + \frac{16L}{\mu}$, $\mathbb{E}_{\mathrm{alg}}$ is the expectation with respect to the random indices sampled during the algorithm, conditionally on the dataset, and $\mathbb{E}_i$ denotes the uniform average over the fixed dataset.*

*Proof.* Similarly to the convex case, projection onto the closed convex set $\mathcal{X}$ does not affect the convergence bounds. Therefore, we obtain

$$\|x^{t+1} - \widehat{x}\|^2 \leq \|x^t - \widehat{x}\|^2 - 2\gamma_t \left\langle \mathrm{clip}_{\lambda_t}\left(\nabla f_i(x^t)\right), x^t - \widehat{x}\right\rangle + \gamma_t^2 \|\mathrm{clip}_{\lambda_t}\left(\nabla f_i(x^t)\right)\|^2$$

$$= \|x^t - \widehat{x}\|^2 - 2\gamma_t \left\langle \mathrm{clip}_{\lambda_t}\left(\nabla f_i(x^t)\right), x^t - \widehat{x}\right\rangle + \gamma_t^2 \|\mathrm{clip}_{\lambda_t}\left(\nabla f_i(x^t)\right) \pm \mathrm{clip}_{\lambda_t}\left(\nabla f_i(\widehat{x})\right)\|^2$$

$$\leq \|x^t - \widehat{x}\|^2 - 2\gamma_t \left\langle \mathrm{clip}_{\lambda_t}\left(\nabla f_i(x^t)\right), x^t - \widehat{x}\right\rangle + 2\gamma_t^2 \|\mathrm{clip}_{\lambda_t}\left(\nabla f_i(x^t)\right) - \mathrm{clip}_{\lambda_t}\left(\nabla f_i(\widehat{x})\right)\|^2 + 2\gamma_t^2 \|\mathrm{clip}_{\lambda_t}\left(\nabla f_i(\widehat{x})\right)\|^2$$

$$\leq \|x^t - \widehat{x}\|^2 - 2\gamma_t \left\langle \mathrm{clip}_{\lambda_t}\left(\nabla f_i(x^t)\right), x^t - \widehat{x}\right\rangle + 2\gamma_t^2 \|\nabla f_i(x^t) - \nabla f_i(\widehat{x})\|^2 + 2\gamma_t^2 \|\mathrm{clip}_{\lambda_t}\left(\nabla f_i(\widehat{x})\right)\|^2$$

$$\leq (1 - \gamma_t \mu)\|x^t - \widehat{x}\|^2 - 2\gamma_t \left(F_N(x^t) - F_N(\widehat{x})\right) - 2\gamma_t \left\langle \mathrm{clip}_{\lambda_t}\left(\nabla f_i(x^t)\right) - \nabla F_N(x^t), x^t - \widehat{x}\right\rangle + 2\gamma_t^2 \|\nabla f_i(x^t) - \nabla f_i(\widehat{x})\|^2$$
$$+ 2\gamma_t^2 \|\mathrm{clip}_{\lambda_t}\left(\nabla f_i(\widehat{x})\right)\|^2.$$

Then we take the conditional expectation and use the Young's inequality:

$$\mathbb{E}_i \|x^{t+1} - \widehat{x}\|^2 \leq (1 - \gamma_t \mu)\|x^t - \widehat{x}\|^2 - 2\gamma_t (F_N(x^t) - F_N(\widehat{x})) - 2\gamma_t \left\langle \mathbb{E}_i \mathrm{clip}_{\lambda_t}\left(\nabla f_i(x^t)\right) - \nabla F_N(x^t), x^t - \widehat{x}\right\rangle$$
$$+ 2\gamma_t^2 \mathbb{E}_i \|\nabla f_i(x^t) - \nabla f_i(\widehat{x})\|^2 + 2\gamma_t^2 \mathbb{E}_i \|\mathrm{clip}_{\lambda_t}\left(\nabla f_i(\widehat{x})\right)\|^2$$

$$\leq (1 - \gamma_t \mu)\|x^t - \widehat{x}\|^2 - 2\gamma_t (F_N(x^t) - F_N(\widehat{x})) + 2\gamma_t \left(\frac{\mu}{4}\|x^t - \widehat{x}\|^2 + \frac{4}{\mu}\|\mathbb{E}_i \mathrm{clip}_{\lambda_t}\left(\nabla f_i(x^t)\right) - \nabla F_N(x^t)\|^2\right)$$
$$+ 2\gamma_t^2 \mathbb{E}_i \|\nabla f_i(x^t) - \nabla f_i(\widehat{x})\|^2 + 2\gamma_t^2 \mathbb{E}_i \|\mathrm{clip}_{\lambda_t}\left(\nabla f_i(\widehat{x})\right)\|^2$$

$$= \left(1 - \frac{\gamma_t \mu}{2}\right)\|x^t - \widehat{x}\|^2 - 2\gamma_t (F_N(x^t) - F_N(\widehat{x})) + \frac{8\gamma_t}{\mu}\|\mathbb{E}_i \mathrm{clip}_{\lambda_t}\left(\nabla f_i(x^t)\right) - \nabla F_N(x^t)\|^2 + 2\gamma_t^2 \mathbb{E}_i \|\nabla f_i(x^t) - \nabla f_i(\widehat{x})\|^2$$
$$+ 2\gamma_t^2 \mathbb{E}_i \|\mathrm{clip}_{\lambda_t}\left(\nabla f_i(\widehat{x})\right)\|^2.$$

As in convex case, further we use the smoothness and Lemma C.4:

$$\mathbb{E}_i \|x^{t+1} - \widehat{x}\|^2 \leq \left(1 - \frac{\gamma_t \mu}{2}\right)\|x^t - \widehat{x}\|^2 + \left(4\gamma_t^2 L - 2\gamma_t\right)(F_N(x^t) - F_N(\widehat{x})) + \frac{8\gamma_t}{\mu}\|\mathbb{E}_i \mathrm{clip}_{\lambda_t}\left(\nabla f_i(x^t)\right) - \nabla F_N(x^t)\|^2$$
$$+ 2\gamma_t^2 \mathbb{E}_i \|\mathrm{clip}_{\lambda_t}\left(\nabla f_i(\widehat{x})\right)\|^2$$

$$\leq \left(1 - \frac{\gamma_t \mu}{2}\right)\|x^t - \widehat{x}\|^2 + \left(4\gamma_t^2 L - 2\gamma_t\right)(F_N(x^t) - F_N(\widehat{x})) + \frac{8\gamma_t}{\mu}\|\mathbb{E}_i \mathrm{clip}_{\lambda_t}\left(\nabla f_i(x^t)\right) - \nabla F_N(x^t)\|^2$$
$$+ 2\gamma_t^2 \lambda_t^{2-\alpha} \mathbb{E}_i \|\nabla f_i(\widehat{x})\|^\alpha.$$

With $\gamma_t \leq \frac{1}{4L}$:

$$\mathbb{E}_i \|x^{t+1} - \widehat{x}\|^2 \leq \left(1 - \frac{\gamma_t \mu}{2}\right)\|x^t - \widehat{x}\|^2 - \gamma_t (F_N(x^t) - F_N(\widehat{x})) + \frac{8\gamma_t}{\mu}\|\mathbb{E}_i \mathrm{clip}_{\lambda_t}\left(\nabla f_i(x^t)\right) - \nabla F_N(x^t)\|^2$$
$$+ 2\gamma_t^2 \lambda_t^{2-\alpha} \mathbb{E}_i \|\nabla f_i(\widehat{x})\|^\alpha.$$

With $\lambda_t \geq 2L\Delta + 2\|\nabla F_N(x^1)\|$ we can apply Lemma C.5:

$$\|\mathbb{E}_i \mathrm{clip}_{\lambda_t}\left(\nabla f_i(x^t)\right) - \nabla F_N(x^t)\|^2 \leq \frac{2^{2\alpha}}{\lambda_t^{2(\alpha-1)}}\left(\mathbb{E}_i \|\nabla f_i(x^t) - \nabla F_N(x^t)\|^\alpha\right)^2$$

$$\leq \frac{2^{2\alpha} 3^{2(\alpha-1)}}{\lambda_t^{2(\alpha-1)}}\left(\mathbb{E}_i \|\nabla f_i(x^t) - \nabla f_i(\widehat{x})\|^\alpha + \mathbb{E}_i \|\nabla f_i(\widehat{x})\|^\alpha + \|\nabla F_N(x^t)\|^\alpha\right)^2$$

$$\leq \frac{2^{2\alpha} 3^{2(\alpha-1)}}{\lambda_t^{2(\alpha-1)}}\left(\mathbb{E}_i \|\nabla f_i(x^t) - \nabla f_i(\widehat{x})\|^\alpha + \mathbb{E}_i \|\nabla f_i(\widehat{x})\|^\alpha + 2^{\alpha-1}\|\nabla F_N(x^t) - \nabla F_N(\widehat{x})\|^\alpha + 2^{\alpha-1}\|\nabla F_N(\widehat{x})\|^\alpha\right)^2$$

$$\leq \frac{2^{2\alpha} 3^{2(\alpha-1)}}{\lambda_t^{2(\alpha-1)}}\left(\mathbb{E}_i \|\nabla f_i(x^t) - \nabla f_i(\widehat{x})\|^\alpha + 3\mathbb{E}_i \|\nabla f_i(\widehat{x})\|^\alpha + 2^{\alpha-1}\|\nabla F_N(x^t) - \nabla F_N(\widehat{x})\|^\alpha\right)^2.$$

Similar to the convex case we derive

$$\|\mathbb{E}_i \mathrm{clip}_{\lambda_t}\left(\nabla f_i(x^t)\right) - \nabla F_N(x^t)\|^2 \leq \frac{2^{2\alpha} 3^{2(\alpha-1)}}{\lambda_t^{2(\alpha-1)}}\left(3(2L)^{\frac{\alpha}{2}}\left(F_N(x^t) - F_N(\widehat{x})\right)^{\frac{\alpha}{2}} + 3\mathbb{E}_i \|\nabla f_i(\widehat{x})\|^\alpha\right)^2$$

$$\leq \frac{2^{2\alpha} 3^{2(\alpha-1)}}{\lambda_t^{2(\alpha-1)}}\left(18(2L)^\alpha\left(F_N(x^t) - F_N(\widehat{x})\right)^\alpha + 18\left(\mathbb{E}_i \|\nabla f_i(\widehat{x})\|^\alpha\right)^2\right).$$

Combining these inequalities altogether we obtain the following:

$$\mathbb{E}_i \|x^{t+1} - \widehat{x}\|^2 \leq \left(1 - \frac{\gamma_t \mu}{2}\right) \|x^t - \widehat{x}\|^2 + \left(\frac{2^{3\alpha+4}3^{2\alpha}L^\alpha \gamma_t}{\mu \lambda_t^{2(\alpha-1)}} (F_N(x^t) - F_N(\widehat{x}))^{\alpha-1} - \gamma_t\right)(F_N(x^t) - F_N(\widehat{x}))$$

$$+ \frac{2^{2\alpha+4}3^{2\alpha}\gamma_t}{\mu \lambda_t^{2(\alpha-1)}} \left(\mathbb{E}_i \|\nabla f_i(\widehat{x})\|^\alpha\right)^2 + 2\gamma_t^2 \lambda_t^{2-\alpha} \mathbb{E}_i \|\nabla f_i(\widehat{x})\|^\alpha.$$

Then we work with the second term. From smoothness and the boundedness of the set $\mathcal{X}$ we have

$$
\begin{aligned}
F_N(x^t) - F_N(\widehat{x}) &\leq \langle \nabla F_N(x^t), x^t - \widehat{x} \rangle \leq \|\nabla F_N(x^t)\| \cdot \|x^t - \widehat{x}\| \leq \Delta \|\nabla F_N(x^t)\| \\
&\leq \Delta \left(\|\nabla F_N(x^1)\| + \|\nabla F_N(x^t) - \nabla F_N(x^1)\|\right) \leq \Delta(\|\nabla F_N(x^1)\| + L\Delta).
\end{aligned}
$$

Therefore, with the choice of $\lambda_t$ as following:

$$\lambda_t^{2(\alpha-1)} \geq \frac{2^{3\alpha+5}3^{2\alpha}L^\alpha}{\mu}\left(\Delta\|\nabla F_N(x^1)\| + L\Delta^2\right)^{\alpha-1},$$

we have

$$\mathbb{E}_i \|x^{t+1} - \widehat{x}\|^2 \leq \left(1 - \frac{\gamma_t \mu}{2}\right) \|x^t - \widehat{x}\|^2 - \frac{\gamma_t}{2}(F_N(x^t) - F_N(\widehat{x})) + \frac{2^{2\alpha+4}3^{2\alpha}\gamma_t}{\mu \lambda_t^{2(\alpha-1)}} \left(\mathbb{E}_i \|\nabla f_i(\widehat{x})\|^\alpha\right)^2$$

$$+ 2\gamma_t^2 \lambda_t^{2-\alpha} \mathbb{E}_i \|\nabla f_i(\widehat{x})\|^\alpha.$$

Then we sum this inequality with weights $w_t = t + \frac{16L}{\mu}$. Choose $\gamma_t = \frac{4}{\mu\left(t + \frac{16L}{\mu}\right)}$. Define $\widetilde{x}^T =$

$\frac{2}{(T+1+\frac{32L}{\mu})T} \sum_{t=1}^{T} \left( t + \frac{16L}{\mu} \right) x^t$. Then, since the objective function is convex, we obtain

$$\mathbb{E}_{\mathrm{alg}} \left[ F_N(\widetilde{x}^T) - F_N(\widehat{x}) \right] \leq \frac{2}{\left( T + 1 + \frac{32L}{\mu} \right) T} \sum_{t=1}^{T} w_t \left( \mathbb{E}_{\mathrm{alg}} \left[ F_N(x^t) - F_N(\widehat{x}) \right] \right)$$

$$\leq \frac{4}{\left( T + 1 + \frac{32L}{\mu} \right) T} \sum_{t=1}^{T} \left[ \frac{w_t(1 - \gamma_t \mu/2)}{\gamma_t} \mathbb{E}_{\mathrm{alg}} \|x^t - \widehat{x}\|^2 - \frac{w_t}{\gamma_t} \mathbb{E}_{\mathrm{alg}} \|x^{t+1} - \widehat{x}\|^2 \right]$$

$$+ \frac{2^{2\alpha+5} 3^{2\alpha}}{\mu \left( T + 1 + \frac{32L}{\mu} \right) T} (\mathbb{E}_i \|\nabla f_i(\widehat{x})\|^\alpha)^2 \sum_{t=1}^{T} w_t \lambda_t^{2(1-\alpha)} + \frac{8}{\mu \left( T + 1 + \frac{32L}{\mu} \right) T} \mathbb{E}_i \|\nabla f_i(\widehat{x})\|^\alpha \sum_{t=1}^{T} \lambda_t^{2-\alpha}$$

$$= \frac{\mu}{\left( T + 1 + \frac{32L}{\mu} \right) T} \sum_{t=1}^{T} \left[ \left( t + \frac{16L}{\mu} \right) \left( t + \frac{16L}{\mu} - 2 \right) \mathbb{E}_{\mathrm{alg}} \|x^t - \widehat{x}\|^2 - \left( t + \frac{16L}{\mu} \right)^2 \mathbb{E}_{\mathrm{alg}} \|x^{t+1} - \widehat{x}\|^2 \right]$$

$$+ \frac{2^{2\alpha+5} 3^{2\alpha}}{\mu \left( T + 1 + \frac{32L}{\mu} \right) T} (\mathbb{E}_i \|\nabla f_i(\widehat{x})\|^\alpha)^2 \sum_{t=1}^{T} t \lambda_t^{2(1-\alpha)} + \frac{2^{2\alpha+9} 3^{2\alpha} L}{\mu^2 \left( T + 1 + \frac{32L}{\mu} \right) T} (\mathbb{E}_i \|\nabla f_i(\widehat{x})\|^\alpha)^2 \sum_{t=1}^{T} \lambda_t^{2(1-\alpha)}$$

$$+ \frac{8}{\mu \left( T + 1 + \frac{32L}{\mu} \right) T} \mathbb{E}_i \|\nabla f_i(\widehat{x})\|^\alpha \sum_{t=1}^{T} \lambda_t^{2-\alpha}$$

$$\leq \frac{\mu}{\left( T + 1 + \frac{32L}{\mu} \right) T} \sum_{t=1}^{T} \left[ \left( t + \frac{16L}{\mu} - 1 \right)^2 \mathbb{E}_{\mathrm{alg}} \|x^t - \widehat{x}\|^2 - \left( t + \frac{16L}{\mu} \right)^2 \mathbb{E}_{\mathrm{alg}} \|x^{t+1} - \widehat{x}\|^2 \right]$$

$$+ \frac{2^{2\alpha+5} 3^{2\alpha}}{\mu \left( T + 1 + \frac{32L}{\mu} \right) T} (\mathbb{E}_i \|\nabla f_i(\widehat{x})\|^\alpha)^2 \sum_{t=1}^{T} t \lambda_t^{2(1-\alpha)} + \frac{2^{2\alpha+9} 3^{2\alpha} L}{\mu^2 \left( T + 1 + \frac{32L}{\mu} \right) T} (\mathbb{E}_i \|\nabla f_i(\widehat{x})\|^\alpha)^2 \sum_{t=1}^{T} \lambda_t^{2(1-\alpha)}$$

$$+ \frac{8}{\mu \left( T + 1 + \frac{32L}{\mu} \right) T} \mathbb{E}_i \|\nabla f_i(\widehat{x})\|^\alpha \sum_{t=1}^{T} \lambda_t^{2-\alpha}$$

$$\leq \frac{\mu}{\left( T + 1 + \frac{32L}{\mu} \right) T} \left[ \left( \frac{16L}{\mu} \right)^2 \|x^1 - \widehat{x}\|^2 - \left( T + \frac{16L}{\mu} \right)^2 \mathbb{E}_{\mathrm{alg}} \|x^{T+1} - \widehat{x}\|^2 \right]$$

$$+ \frac{2^{2\alpha+5} 3^{2\alpha}}{\mu \left( T + 1 + \frac{32L}{\mu} \right) T} (\mathbb{E}_i \|\nabla f_i(\widehat{x})\|^\alpha)^2 \sum_{t=1}^{T} t \lambda_t^{2(1-\alpha)} + \frac{2^{2\alpha+9} 3^{2\alpha} L}{\mu^2 \left( T + 1 + \frac{32L}{\mu} \right) T} (\mathbb{E}_i \|\nabla f_i(\widehat{x})\|^\alpha)^2 \sum_{t=1}^{T} \lambda_t^{2(1-\alpha)}$$

$$+ \frac{8}{\mu \left( T + 1 + \frac{32L}{\mu} \right) T} \mathbb{E}_i \|\nabla f_i(\widehat{x})\|^\alpha \sum_{t=1}^{T} \lambda_t^{2-\alpha}.$$

If we take $\lambda_t = \lambda_0 t^{1/\alpha}$, then, we obtain

$$\mathbb{E}_{\mathrm{alg}} \left[ F_N(\widetilde{x}^T) - F_N(\widehat{x}) \right] = \mathcal{O} \left( \frac{L^2 \|x^1 - \widehat{x}\|^2}{\mu T^2} + \frac{(\mathbb{E}_i \|\nabla f_i(\widehat{x})\|^\alpha)^2 \lambda_0^{2(1-\alpha)}}{\mu T^{\frac{2(\alpha-1)}{\alpha}}} + \frac{(\mathbb{E}_i \|\nabla f_i(\widehat{x})\|^\alpha)^2 L \lambda_0^{2(1-\alpha)}}{\mu^2 T^{\frac{3\alpha-2}{\alpha}}} + \frac{\mathbb{E}_i \|\nabla f_i(\widehat{x})\|^\alpha \lambda_0^{2-\alpha}}{\mu T^{\frac{2(\alpha-1)}{\alpha}}} \right).$$

$\square$

**Theorem C.5.** *Let the Assumptions 3.2, 3.3, 3.4 and 3.5 be satisfied. Then, for any $\alpha \in (1, 2]$ choosing $\gamma_t = \frac{4}{\mu(t+16L/\mu)}, \lambda_t = \lambda_0 t$ with $\lambda_0 \geq 2^5 3^2 L\Delta + 2\|\nabla F_N(x^1)\|$ results in the following convergence bounds for ClipSGD:*

$$\mathbb{E}_{\mathrm{alg}} \left[ F_N(\widetilde{x}^T) - F_N(\widehat{x}) \right] = \mathcal{O} \left( \frac{L^2 \Delta^2}{\mu T^2} + \frac{\mathbb{E}_i \|\nabla f_i(\widehat{x})\|^\alpha \lambda_0^{2-\alpha}}{\mu T^{\alpha-1}} + \frac{\Delta \mathbb{E}_i \|\nabla f_i(\widehat{x})\|^\alpha \lambda_0^{1-\alpha}}{\mu T^{\alpha-1}} \right), \tag{57}$$

*where $\widetilde{x}^T = \frac{1}{W_T} \sum_{t=1}^{T} w_t x^t$, $W_T = \sum_{t=1}^{T} w_t$, $w_t = t + \frac{16L}{\mu}$, $\mathbb{E}_{\mathrm{alg}}$ is the expectation with respect to the random indices sampled during the algorithm, conditionally on the dataset, and $\mathbb{E}_i$ denotes the uniform average over the fixed dataset.*

*Proof.* As in previous theorem we obtain the following inequality:

$$\|x^{t+1} - \widehat{x}\|^2 \leq (1 - \gamma_t \mu)\|x^t - \widehat{x}\|^2 - 2\gamma_t \left(F_N(x^t) - F_N(\widehat{x})\right) - 2\gamma_t \left\langle \mathrm{clip}_{\lambda_t}\left(\nabla f_i(x^t)\right) - \nabla F_N(x^t), x^t - \widehat{x} \right\rangle$$
$$+ 2\gamma_t^2 \|\nabla f_i(x^t) - \nabla f_i(\widehat{x})\|^2 + 2\gamma_t^2 \|\mathrm{clip}_{\lambda_t}\left(\nabla f_i(\widehat{x})\right)\|^2.$$

Similar to convex case with $\gamma_0 \leq \frac{1}{4L}$ and $\lambda_t \geq 2^5 3^2 L\Delta + 2\|\nabla F_N(x^1)\|$, we obtain the following bound:

$$\mathbb{E}_i \|x^{t+1} - \widehat{x}\|^2 \leq (1 - \gamma_t \mu)\|x^t - \widehat{x}\|^2 - \frac{\gamma_t}{2}(F_N(x^t) - F_N(\widehat{x})) + \frac{C_1}{\lambda_t^{\alpha-1}}\gamma_t \mathbb{E}_i \|\nabla f_i(\widehat{x})\|^\alpha$$
$$+ 2\gamma_t^2 \lambda_t^{2-\alpha} \mathbb{E}_i \|\nabla f_i(\widehat{x})\|^\alpha + I(\alpha \neq 2)\frac{C_2}{\lambda_t^{\frac{2\alpha-2}{2-\alpha}}}\gamma_t.$$

Rearranging the terms we get

$$F_N(x^t) - F_N(\widehat{x}) \leq \frac{2(1 - \gamma_t \mu)}{\gamma_t}\|x^t - \widehat{x}\|^2 - \frac{2}{\gamma_t}\mathbb{E}_i \|x^{t+1} - \widehat{x}\|^2 + \frac{2C_1 \mathbb{E}_i \|\nabla f_i(\widehat{x})\|^\alpha}{\lambda_t^{\alpha-1}}$$
$$+ 4\gamma_t \lambda_t^{2-\alpha} \mathbb{E}_i \|\nabla f_i(\widehat{x})\|^\alpha + \frac{C_2}{\lambda_t^{\frac{2\alpha-2}{2-\alpha}}}$$
$$\leq \frac{2(1 - \gamma_t \mu/2)}{\gamma_t}\|x^t - \widehat{x}\|^2 - \frac{2}{\gamma_t}\mathbb{E}_i \|x^{t+1} - \widehat{x}\|^2 + \frac{2C_1 \mathbb{E}_i \|\nabla f_i(\widehat{x})\|^\alpha}{\lambda_t^{\alpha-1}}$$
$$+ 4\gamma_t \lambda_t^{2-\alpha} \mathbb{E}_i \|\nabla f_i(\widehat{x})\|^\alpha + \frac{C_2}{\lambda_t^{\frac{2\alpha-2}{2-\alpha}}}.$$

Then, we sum this inequality with weights $w_t = t + \frac{16L}{\mu}$. Choose $\gamma_t = \frac{4}{\mu(t + \frac{16L}{\mu})}$. Define $\widetilde{x}^T = \frac{2}{(T+1+\frac{32L}{\mu})T} \sum_{t=1}^{T}\left(t + \frac{16L}{\mu}\right)x^t$. Then, since the objective is convex we obtain

$$\mathbb{E}_{\mathrm{alg}}\left[F_N(\widetilde{x}^T) - F_N(\widehat{x})\right] \leq \frac{2}{\left(T + 1 + \frac{32L}{\mu}\right)T} \sum_{t=1}^{T} w_t \left(\mathbb{E}_{\mathrm{alg}}\left[F_N(x^t) - F_N(\widehat{x})\right]\right)$$

$$\leq \frac{4}{\left(T + 1 + \frac{32L}{\mu}\right)T} \sum_{t=1}^{T}\left[\frac{w_t(1 - \gamma_t \mu/2)}{\gamma_t}\mathbb{E}_{\mathrm{alg}}\|x^t - \widehat{x}\|^2 - \frac{w_t}{\gamma_t}\mathbb{E}_{\mathrm{alg}}\|x^{t+1} - \widehat{x}\|^2\right]$$

$$+ \frac{4C_1 \mathbb{E}_i \|\nabla f_i(\widehat{x})\|^\alpha}{\left(T + 1 + \frac{32L}{\mu}\right)T} \sum_{t=1}^{T} w_t \lambda_t^{1-\alpha} + \frac{8\mathbb{E}_i \|\nabla f_i(\widehat{x})\|^\alpha}{\left(T + 1 + \frac{32L}{\mu}\right)T} \sum_{t=1}^{T} \gamma_t w_t \lambda_t^{2-\alpha} + \frac{2C_2}{\left(T + 1 + \frac{32L}{\mu}\right)T} \sum_{t=1}^{T} w_t \lambda_t^{\frac{2-2\alpha}{2-\alpha}}$$

$$\leq \frac{\mu}{\left(T + 1 + \frac{32L}{\mu}\right)T}\left(\left(\frac{16L}{\mu}\right)^2 \|x^1 - \widehat{x}\|^2 - \left(T + \frac{16L}{\mu}\right)^2 \mathbb{E}_{\mathrm{alg}}\|x^{T+1} - \widehat{x}\|^2\right) + \frac{4C_1 \mathbb{E}_i \|\nabla f_i(\widehat{x})\|^\alpha}{\left(T + 1 + \frac{32L}{\mu}\right)T} \sum_{t=1}^{T} t\lambda_t^{1-\alpha}$$

$$+ \frac{64LC_1 \mathbb{E}_i \|\nabla f_i(\widehat{x})\|^\alpha}{\mu\left(T + 1 + \frac{32L}{\mu}\right)T} \sum_{t=1}^{T} \lambda_t^{1-\alpha} + \frac{32\mathbb{E}_i \|\nabla f_i(\widehat{x})\|^\alpha}{\mu\left(T + 1 + \frac{32L}{\mu}\right)T} \sum_{t=1}^{T} \lambda_t^{2-\alpha} + \frac{2C_2}{\left(T + 1 + \frac{32L}{\mu}\right)T} \sum_{t=1}^{T} t\lambda_t^{\frac{2-2\alpha}{2-\alpha}}$$

$$+ \frac{32LC_2}{\mu\left(T + 1 + \frac{32L}{\mu}\right)T} \sum_{t=1}^{T} \lambda_t^{\frac{2-2\alpha}{2-\alpha}}$$

If we take $\lambda_t = \lambda_0 t$, then, we obtain

$$\mathbb{E}_{\mathrm{alg}}\left[F_N(\widetilde{x}^T) - F_N(\widehat{x})\right] = \mathcal{O}\left(\frac{L^2\|x^1 - \widehat{x}\|^2}{\mu T^2} + \frac{\Delta\mathbb{E}_i\|\nabla f_i(\widehat{x})\|^\alpha}{T^{\alpha-1}} + \frac{L\Delta\mathbb{E}_i\|\nabla f_i(\widehat{x})\|^\alpha}{\mu T^{\alpha-1}} + \frac{\mathbb{E}_i\|\nabla f_i(\widehat{x})\|^\alpha}{\mu T^{\alpha-1}}\right.$$

$$\left. + I(\alpha \neq 2)\Delta^{\frac{2}{2-\alpha}}L^{\frac{\alpha}{2-\alpha}}R_1(T,\alpha) + I(\alpha \neq 2)\frac{\Delta^{\frac{2}{2-\alpha}}L^{\frac{2(\alpha-1)}{2-\alpha}}}{\mu}R_2(T,\alpha)\right),$$

where

$$R_1(T,\alpha) = \begin{cases} T^{-\frac{2(\alpha-1)}{2-\alpha}}, & 1 < \alpha < \frac{3}{2}, \\ \dfrac{\log(eT)}{T^2}, & \alpha = \frac{3}{2}, \\ \dfrac{1}{T^2}, & \frac{3}{2} < \alpha < 2, \end{cases} \qquad R_2(T,\alpha) = \begin{cases} T^{-\frac{\alpha}{2-\alpha}}, & 1 < \alpha < \frac{4}{3}, \\ \dfrac{\log(eT)}{T^2}, & \alpha = \frac{4}{3}, \\ \dfrac{1}{T^2}, & \frac{4}{3} < \alpha < 2. \end{cases} \tag{58}$$

$\square$

To combine the optimization bounds with the high-probability generalization bounds, fix any $q \in (1,\alpha)$ and any $\delta \in (0,1)$. Here $\delta$ denotes the failure probability. By Theorem 3.1, with probability at least $1 - \delta$ with respect to the draw of the dataset,

$$F(\widehat{x}) - F(x_\star) = \mathcal{O}\left(\frac{\Delta^{2-q}\mathbb{E}\|\nabla f(x_\star,\xi)\|^q}{\mu^{q-1}N^{q-1}\delta}\right). \tag{59}$$

On this event, we can control the empirical gradient moment at $\widehat{x}$ through the one at $x_\star$:

$$\mathbb{E}_i\|\nabla f_i(\widehat{x})\|^\alpha = \frac{1}{N}\sum_{i=1}^N \|\nabla f(\widehat{x},\xi_i)\|^\alpha$$

$$\leq \frac{2^{\alpha-1}}{N}\sum_{i=1}^N \|\nabla f(x_\star,\xi_i)\|^\alpha + \frac{2^{\alpha-1}}{N}\sum_{i=1}^N \|\nabla f(\widehat{x},\xi_i) - \nabla f(x_\star,\xi_i)\|^\alpha$$

$$\leq 2^{\alpha-1}\mathbb{E}_i\|\nabla f_i(x_\star)\|^\alpha + 2^{\alpha-1}L^\alpha\|\widehat{x} - x_\star\|^\alpha$$

$$\leq 2^{\alpha-1}\mathbb{E}_i\|\nabla f_i(x_\star)\|^\alpha + 2^{\alpha-1}L^\alpha\left(\frac{2}{\mu}\right)^{\frac{\alpha}{2}}(F(\widehat{x}) - F(x_\star))^{\frac{\alpha}{2}}$$

$$\leq 2^{\alpha-1}\mathbb{E}_i\|\nabla f_i(x_\star)\|^\alpha + \mathcal{O}\left(\frac{L^\alpha\Delta^{\frac{\alpha(2-q)}{2}}\left(\mathbb{E}\|\nabla f(x_\star,\xi)\|^q\right)^{\frac{\alpha}{2}}}{\mu^{\frac{\alpha q}{2}}N^{\frac{\alpha(q-1)}{2}}\delta^{\frac{\alpha}{2}}}\right). \tag{60}$$

In the last step we used equation 59. Therefore, the optimization bounds can be combined with generalization by replacing every occurrence of $\mathbb{E}_i\|\nabla f_i(\widehat{x})\|^\alpha$ by the right-hand side of equation 60.

**Corollary C.6.** *Let the assumptions of Theorem C.4 be satisfied. Fix any $q \in (1,\alpha)$ and $\delta \in (0,1)$. Then, with probability at least $1 - \delta$ with respect to the draw of the dataset, running ClipSGD with $\lambda_t = \lambda_0 t^{1/\alpha}$ gives*

$$\mathbb{E}_{\mathrm{alg}}\left[F_N(\widetilde{x}^T) - F_N(\widehat{x})\right] = \mathcal{O}\left(\frac{L^2\Delta^2}{\mu T^2} + \frac{(\mathbb{E}_i\|\nabla f_i(x_\star)\|^\alpha)^2\lambda_0^{2(1-\alpha)}}{\mu T^{\frac{2(\alpha-1)}{\alpha}}} + \frac{L^{2\alpha}\Delta^{\alpha(2-q)}\left(\mathbb{E}\|\nabla f(x_\star,\xi)\|^q\right)^\alpha\lambda_0^{2(1-\alpha)}}{\mu^{\alpha q+1}N^{\alpha(q-1)}\delta^\alpha T^{\frac{2(\alpha-1)}{\alpha}}}\right.$$

$$\left. + \frac{\mathbb{E}_i\|\nabla f_i(x_\star)\|^\alpha\lambda_0^{2-\alpha}}{\mu T^{\frac{2(\alpha-1)}{\alpha}}} + \frac{L^\alpha\Delta^{\frac{\alpha(2-q)}{2}}\left(\mathbb{E}\|\nabla f(x_\star,\xi)\|^q\right)^{\frac{\alpha}{2}}\lambda_0^{2-\alpha}}{\mu^{\frac{\alpha q}{2}+1}N^{\frac{\alpha(q-1)}{2}}\delta^{\frac{\alpha}{2}}T^{\frac{2(\alpha-1)}{\alpha}}}\right). \tag{61}$$

*Moreover, under the assumptions of Theorem C.5, running ClipSGD with $\lambda_t = \lambda_0 t$ gives*

$$\mathbb{E}_{\mathrm{alg}}\left[F_N(\widetilde{x}^T) - F_N(\widehat{x})\right] = \mathcal{O}\left(\frac{L^2\Delta^2}{\mu T^2} + \frac{\mathbb{E}_i\|\nabla f_i(x_\star)\|^\alpha\lambda_0^{2-\alpha}}{\mu T^{\alpha-1}} + \frac{L^\alpha\Delta^{\frac{\alpha(2-q)}{2}}\left(\mathbb{E}\|\nabla f(x_\star,\xi)\|^q\right)^{\frac{\alpha}{2}}\lambda_0^{2-\alpha}}{\mu^{\frac{\alpha q}{2}+1}N^{\frac{\alpha(q-1)}{2}}\delta^{\frac{\alpha}{2}}T^{\alpha-1}}\right.$$

$$+ \frac{\Delta\mathbb{E}_i\|\nabla f_i(x_\star)\|^\alpha\lambda_0^{1-\alpha}}{T^{\alpha-1}} + \frac{L^\alpha\Delta^{\frac{\alpha(2-q)}{2}+1}\left(\mathbb{E}\|\nabla f(x_\star,\xi)\|^q\right)^{\frac{\alpha}{2}}\lambda_0^{1-\alpha}}{\mu^{\frac{\alpha q}{2}}N^{\frac{\alpha(q-1)}{2}}\delta^{\frac{\alpha}{2}}T^{\alpha-1}}$$

$$\left. + I(\alpha \neq 2)\Delta^{\frac{2}{2-\alpha}}L^{\frac{\alpha}{2-\alpha}}\lambda_0^{\frac{2-2\alpha}{2-\alpha}}R_1(T,\alpha) + I(\alpha \neq 2)\frac{\Delta^{\frac{2}{2-\alpha}}L^{\frac{\alpha}{2-\alpha}+1}\lambda_0^{\frac{2-2\alpha}{2-\alpha}}}{\mu}R_2(T,\alpha)\right). \tag{62}$$

*Where $R_1$ and $R_2$ are defined as in 58.*

## D. Combining Optimization and Heavy-Tailedness bounds

In this section we analyze the optimization bounds with the obtained earlier bounds on $\frac{1}{N} \sum_{i=1}^{N} \|\nabla f(x_\star, \xi_i)\|^{\alpha_{\text{alg}}}$ and derive optimal step size and clipping radii schedules

**Lemma D.1.** *Let the Assumptions 3.1, 3.3 3.4 and 3.5 be satisfied. Then, running ClipSGD with $\gamma_t = \gamma_0 t^{-1/\alpha_{\text{alg}}}$ and $\lambda_t = \lambda_0 t^{1/\alpha_{\text{alg}}}$ with $\gamma_0$, $\lambda_0$ according to Theorem C.1 results in following bounds with high probability according to the drawn samples:*

1. *If $\alpha_{\text{alg}} < \alpha_{\text{true}}$ the overall bound is*

$$\mathbb{E}_{\text{alg}} \left[ F_N(\overline{x}^T) - F_N(\widehat{x}) \right] = \mathcal{O} \left( \frac{\mathbb{E}\|\nabla f(x_\star, \xi)\|^{\alpha_{\text{alg}}}}{T^{\frac{\alpha_{\text{alg}}-1}{\alpha_{\text{alg}}}}} \right). \tag{63}$$

2. *If $\alpha_{\text{alg}} = \alpha_{\text{true}}$ the overall bound is*

$$\mathbb{E}_{\text{alg}} \left[ F_N(\overline{x}^T) - F_N(\widehat{x}) \right] = \mathcal{O} \left( \frac{\ln N}{T^{\frac{\alpha_{\text{alg}}-1}{\alpha_{\text{alg}}}}} \right). \tag{64}$$

3. *If $\alpha_{\text{alg}} > \alpha_{\text{true}}$ the overall bound is*

$$\mathbb{E}_{\text{alg}} \left[ F_N(\overline{x}^T) - F_N(\widehat{x}) \right] = \mathcal{O} \left( \frac{N^{\frac{\alpha_{\text{alg}}}{\alpha_{\text{true}}}-1}}{T^{\frac{\alpha_{\text{alg}}-1}{\alpha_{\text{alg}}}}} \right). \tag{65}$$

*Proof.* The stochastic term in Theorem C.1 is the following:

$$\mathbb{E}_{\text{alg}} \left[ F_N(\overline{x}^T) - F_N(\widehat{x}) \right] = \mathcal{O} \left( \frac{\frac{1}{N} \sum_{i=1}^{N} \|\nabla f(x_\star, \xi_i)\|^{\alpha_{\text{alg}}}}{T^{\frac{\alpha_{\text{alg}}-1}{\alpha_{\text{alg}}}}} \right).$$

Since Assumption 3.5 is satisfied, applying the Theorem B.3 results in the following bounds with high probability according to the drawn samples. $\qquad\square$

Then we derive the optimal $\alpha_{\text{alg}}$ based on different relations between $N$ and $T$. As in the main paper we note that from the Assumption 3.5 with $\alpha_{\text{alg}} \to \alpha_{\text{true}} - 0$, we have $\mathbb{E}\|\nabla f(x_\star, \xi)\|^{\alpha_{\text{alg}}} \to \infty$. Therefore, as the number of samples and iterations is considered to be finite, in choosing the optimal $\alpha_{\text{alg}}$ we distinguish only two cases: $\alpha_{\text{alg}} = \alpha_{\text{true}}$ and $\alpha_{\text{alg}} > \alpha_{\text{true}}$.

**Lemma D.2.** *Let all the Assumptions for the D.1 be satisfied. Then, for 3 different regimes, based on the relation between $N$ and $T$, we have the following optimal choice of $\alpha_{\text{alg}}$:*

1. *If $N \gg T$, the optimal choice is $\alpha_{\text{alg}} = \alpha_{\text{true}}$, which results in overall bound*

$$\mathbb{E}_{\text{alg}} \left[ F_N(\overline{x}^T) - F_N(\widehat{x}) \right] = \mathcal{O} \left( \frac{\ln N}{T^{\frac{\alpha_{\text{true}}-1}{\alpha_{\text{true}}}}} \right). \tag{66}$$

2. *If $N \approx T$, the optimal choice is $\alpha_{\text{alg}} = \alpha_{\text{true}}$, which results in overall bound*

$$\mathbb{E}_{\text{alg}} \left[ F_N(\overline{x}^T) - F_N(\widehat{x}) \right] = \mathcal{O} \left( \frac{\ln N}{T^{\frac{\alpha_{\text{true}}-1}{\alpha_{\text{true}}}}} \right). \tag{67}$$

3. *If $N \ll T$, the optimal choice is $\alpha_{\text{alg}} = 2$, which results in overall bound*

$$\mathbb{E}_{\text{alg}} \left[ F_N(\overline{x}^T) - F_N(\widehat{x}) \right] = \mathcal{O} \left( \frac{1}{\sqrt{T}} \right). \tag{68}$$

*Proof.* If $N \gg T$, then, in order to mitigate the effect of rising $N$ one should choose $\alpha_{\text{alg}} = \alpha_{\text{true}}$.

If $N \ll T$, then, $N^{\frac{2}{\alpha_{\text{true}}}-1} \ll \sqrt{T}$ therefore, the optimal choice is the one with the best asymptotics in $T$, which is $\alpha_{\text{alg}} = 2$.

If $N \approx T$, then, for $\alpha_{\text{alg}} = \alpha_{\text{true}}$ the stochastic term becomes proportional to

$$\frac{\ln N}{T^{\frac{\alpha_{\text{true}}-1}{\alpha_{\text{true}}}}} \approx \frac{\ln T}{T^{\frac{\alpha_{\text{true}}-1}{\alpha_{\text{true}}}}}.$$

For $\alpha_{\text{alg}} > \alpha_{\text{true}}$ it is proportional to

$$\frac{N^{\frac{\alpha_{\text{alg}}}{\alpha_{\text{true}}}-1}}{T^{\frac{\alpha_{\text{alg}}-1}{\alpha_{\text{alg}}}}} \approx \frac{T^{\frac{\alpha_{\text{alg}}}{\alpha_{\text{true}}}-1}}{T^{\frac{\alpha_{\text{alg}}-1}{\alpha_{\text{alg}}}}} = \frac{1}{T^{-\frac{\alpha_{\text{alg}}}{\alpha_{\text{true}}}-\frac{1}{\alpha_{\text{alg}}}+2}}.$$

The minimum of function $\frac{\alpha}{\alpha_{\text{true}}} + \frac{1}{\alpha} - 2$ is obtained at $\alpha = \sqrt{\alpha_{\text{true}}}$, which is strictly less than $\alpha_{\text{true}} \in (1, 2]$. Therefore, taking $\alpha_{\text{alg}} = \alpha_{\text{true}} + \varepsilon$, for $\varepsilon > 0$ we compare

$$\frac{\alpha_{\text{true}} - 1}{\alpha_{\text{true}}} \quad \text{and} \quad \frac{\alpha_{\text{true}}(\alpha_{\text{true}} - 1) - \varepsilon^2}{(\alpha_{\text{true}} + \varepsilon)\alpha_{\text{true}}}.$$

Multiplying by $\alpha_{\text{true}}(\alpha_{\text{true}} + \varepsilon)$ and eliminating the common terms this results in comparing

$$\varepsilon(\alpha_{\text{true}} - 1) \quad \text{and} \quad -\varepsilon^2.$$

The left side is always greater, than, the right one, therefore, the optimal choice of $\alpha_{\text{alg}}$ is $\alpha_{\text{alg}} = \alpha_{\text{true}}$. $\qquad \square$

In case we additionally tune $\gamma_0$ and $\lambda_0$ as follows:

$$\gamma_0 = \Delta \left( \frac{N}{\sum_{i=1}^{N} \|\nabla f_i(x_\star)\|^{\alpha_{\text{alg}}}} \right)^{1/\alpha_{\text{alg}}}, \tag{69}$$

$$\lambda_0 = \left( \frac{1}{N} \sum_{i=1}^{N} \|\nabla f_i(x_\star)\|^{\alpha_{\text{alg}}} \right)^{1/\alpha_{\text{alg}}}, \tag{70}$$

we can obtain better convergence rates with different optimal choice of $\alpha_{\text{alg}}$

**Lemma D.3.** *Let all the Assumptions for the D.1 be satisfied and $\gamma_0$ and $\lambda_0$ be tuned as in equations 69. Then, for 3 different regimes, based on the relation between $N$ and $T$ we have the following optimal choice of $\alpha_{\text{alg}}$:*

1. *If $N \gg T$, the optimal choice is $\alpha_{\text{alg}} = \alpha_{\text{true}}$, which results in overall bound*

$$\mathbb{E}_{\text{alg}} \left[ F_N(\overline{x}^T) - F_N(\widehat{x}) \right] = \mathcal{O} \left( \frac{(\ln N)^{\frac{1}{\alpha_{\text{true}}}}}{T^{\frac{\alpha_{\text{true}}-1}{\alpha_{\text{true}}}}} \right). \tag{71}$$

2. *If $N \approx T$, the optimal choice is $\alpha_{\text{alg}} > \alpha_{\text{true}}$, which results in overall bound*

$$\mathbb{E}_{\text{alg}} \left[ F_N(\overline{x}^T) - F_N(\widehat{x}) \right] = \mathcal{O} \left( \frac{1}{T^{\frac{\alpha_{\text{true}}-1}{\alpha_{\text{true}}}}} \right). \tag{72}$$

3. *If $N \ll T$, the optimal choice is $\alpha_{\text{alg}} = 2$, which results in overall bound*

$$\mathbb{E}_{\text{alg}} \left[ F_N(\overline{x}^T) - F_N(\widehat{x}) \right] = \mathcal{O} \left( \frac{1}{\sqrt{T}} \right). \tag{73}$$

*Proof.* From Theorem C.1 the following convergence rates, including $\gamma_0$ and $\lambda_0$ can be derived:

$$\mathbb{E}_{\text{alg}} \left[ F_N(\overline{x}^T) - F_N(\widehat{x}) \right] = \mathcal{O} \left( \frac{\Delta^2}{\gamma_0 T^{\frac{\alpha_{\text{alg}}-1}{\alpha_{\text{alg}}}}} + \frac{\gamma_0 \lambda_0^{2-\alpha_{\text{alg}}} \left( \frac{1}{N} \sum_{i=1}^{N} \|\nabla f(x_\star, \xi_i)\|^{\alpha_{\text{alg}}} \right)}{T^{\frac{\alpha_{\text{alg}}-1}{\alpha_{\text{alg}}}}} + \frac{\Delta \lambda_0^{1-\alpha_{\text{alg}}} \left( \frac{1}{N} \sum_{i=1}^{N} \|\nabla f(x_\star, \xi_i)\|^{\alpha_{\text{alg}}} \right)}{T^{\frac{\alpha_{\text{alg}}-1}{\alpha_{\text{alg}}}}} \right).$$

This expression is minimized with $\gamma_0$ and $\lambda_0$ chosen as in equations 69, with resulting bound

$$\mathbb{E}_{\text{alg}} \left[ F_N(\overline{x}^T) - F_N(\widehat{x}) \right] = \mathcal{O} \left( \frac{\Delta \left( \frac{1}{N} \sum_{i=1}^{N} \|\nabla f(x_\star, \xi_i)\|^{\alpha_{\text{alg}}} \right)^{\frac{1}{\alpha_{\text{alg}}}}}{T^{\frac{\alpha_{\text{alg}}-1}{\alpha_{\text{alg}}}}} \right).$$

With $\alpha_{\text{alg}} < \alpha_{\text{true}}$ this scales as

$$\frac{\left(\mathbb{E}\|\nabla f(x_\star, \xi)\|^{\alpha_{\text{alg}}}\right)^{\frac{1}{\alpha_{\text{alg}}}}}{T^{\frac{\alpha_{\text{alg}}-1}{\alpha_{\text{alg}}}}}.$$

With $\alpha_{\text{alg}} = \alpha_{\text{true}}$ this scales as

$$\frac{(\ln N)^{\frac{1}{\alpha_{\text{true}}}}}{T^{\frac{\alpha_{\text{true}}-1}{\alpha_{\text{true}}}}}.$$

With $\alpha_{\text{alg}} > \alpha_{\text{true}}$ this scales as

$$\frac{N^{\left(\frac{\alpha_{\text{alg}}}{\alpha_{\text{true}}}-1\right)\frac{1}{\alpha_{\text{alg}}}}}{T^{\frac{\alpha_{\text{alg}}-1}{\alpha_{\text{alg}}}}}.$$

Cases $N \gg T$ and $N \ll T$ are derived as previously. The most interesting is the case with $N \approx T$. For $\alpha_{\text{alg}} > \alpha_{\text{true}}$ it results in

$$\frac{N^{\left(\frac{\alpha_{\text{alg}}}{\alpha_{\text{true}}}-1\right)\frac{1}{\alpha_{\text{alg}}}}}{T^{\frac{\alpha_{\text{alg}}-1}{\alpha_{\text{alg}}}}} = \left(\frac{N^{\frac{\alpha_{\text{alg}}-\alpha_{\text{true}}}{\alpha_{\text{true}}}}}{T^{\alpha_{\text{alg}}-1}}\right)^{\frac{1}{\alpha_{\text{alg}}}} \approx \left(\frac{T^{\frac{\alpha_{\text{alg}}-\alpha_{\text{true}}}{\alpha_{\text{true}}}}}{T^{\alpha_{\text{alg}}-1}}\right)^{\frac{1}{\alpha_{\text{alg}}}} = \left(T^{\frac{\alpha_{\text{alg}}}{\alpha_{\text{true}}}-1-\alpha_{\text{alg}}+1}\right)^{\frac{1}{\alpha_{\text{alg}}}} = T^{\frac{1}{\alpha_{\text{true}}}-1} = \frac{1}{T^{\frac{\alpha_{\text{true}}-1}{\alpha_{\text{true}}}}},$$

which is faster, than with choice $\alpha_{\text{alg}} = \alpha_{\text{true}}$ due to logarithmic factors. $\qquad\square$

**Corollary D.1.** *Though, all results above are obtained for horizon-free scheduling, they also hold for horizon-fixed scheduling with $\gamma_t = \gamma_0 T^{-1/\alpha_{\text{alg}}}, \lambda_t = \lambda_0 T^{1/\alpha_{\text{alg}}}$.*

As for the convex case, similar results for the strongly convex one can also be achieved.

**Lemma D.4.** *Let the Assumptions 3.2, 3.3, 3.4 and 3.5 be satisfied. Then, running ClipSGD with $\gamma_t = \frac{4}{\mu\left(t+\frac{16L}{\mu}\right)}$ and $\lambda_t = \lambda_0 t^{1/\alpha_{\text{alg}}}$ or $\lambda_t = \lambda_0 t$ with $\lambda_0$ satisfying the conditions of Theorems C.4 and C.5 respectively results in following bounds with high probability according to the drawn samples:*

1. *If $\alpha_{\text{alg}} < \alpha_{\text{true}}$ the overall bound is*

$$\mathbb{E}_{\text{alg}}\left[F_N(\tilde{x}^T) - F_N(\hat{x})\right] = \mathcal{O}\left(\min\left\{\frac{(\mathbb{E}\|\nabla f(x_\star, \xi)\|^{\alpha_{\text{alg}}})^2}{\mu T^{\frac{2(\alpha_{\text{alg}}-1)}{\alpha_{\text{alg}}}}}; \frac{\mathbb{E}\|\nabla f(x_\star, \xi)\|^{\alpha_{\text{alg}}}}{\mu T^{\alpha_{\text{alg}}-1}}\right\}\right). \tag{74}$$

2. *If $\alpha_{\text{alg}} = \alpha_{\text{true}}$ the overall bound is*

$$\mathbb{E}_{\text{alg}}\left[F_N(\tilde{x}^T) - F_N(\hat{x})\right] = \mathcal{O}\left(\min\left\{\frac{(\ln N)^2}{\mu T^{\frac{2(\alpha_{\text{alg}}-1)}{\alpha_{\text{alg}}}}}; \frac{\ln N}{\mu T^{\alpha_{\text{alg}}-1}}\right\}\right). \tag{75}$$

3. *If $\alpha_{\text{alg}} > \alpha_{\text{true}}$ the overall bound is*

$$\mathbb{E}_{\text{alg}}\left[F_N(\tilde{x}^T) - F_N(\hat{x})\right] = \mathcal{O}\left(\min\left\{\frac{N^{\frac{2\alpha_{\text{alg}}}{\alpha_{\text{true}}}-2}}{\mu T^{\frac{2(\alpha_{\text{alg}}-1)}{\alpha_{\text{alg}}}}}; \frac{N^{\frac{\alpha_{\text{alg}}}{\alpha_{\text{true}}}-1}}{\mu T^{\alpha_{\text{alg}}-1}}\right\}\right). \tag{76}$$

*where $\tilde{x}^T = \frac{1}{W_T}\sum_{t=1}^{T} w_t x^t, W_T = \sum_{t=1}^{T} w_t, w_t = t + \frac{16L}{\mu}$. Taking the better of the two clipping schedules yields the minimum of the two bounds.*

*Proof.* According to the Theorem C.4, if the clipping is taken as $\lambda_t = \lambda_0 t^{1/\alpha_{\text{alg}}}$ the main stochastic term has asymptotic

$$\mathcal{O}\left(\frac{\left(\frac{1}{N}\sum_{i=1}^{N}\|\nabla f(x_\star, \xi_i)\|^{\alpha_{\text{alg}}}\right)^2}{\mu T^{\frac{2(\alpha_{\text{alg}}-1)}{\alpha_{\text{alg}}}}}\right),$$

and to the Theorem C.5 with $\lambda_t = \lambda_0 t$:

$$\mathcal{O}\left(\frac{\frac{1}{N}\sum_{i=1}^{N}\|\nabla f(x_\star, \xi_i)\|^{\alpha_{\mathrm{alg}}}}{\mu T^{\alpha_{\mathrm{alg}}-1}}\right).$$

Since Assumption 3.5 is satisfied, applying the Theorem B.3 results in the following bounds with high probability according to the drawn samples. □

**Lemma D.5.** *Let all the Assumptions for the D.4 be satisfied. Then, for 3 different regimes based on the relation between $N$ and $T$ we have the following optimal choice of $\alpha_{\mathrm{alg}}$:*

1. *If $N \gg T$, the optimal choice is $\alpha_{\mathrm{alg}} = \alpha_{\mathrm{true}}$, which results in overall bound*

$$\mathbb{E}_{\mathrm{alg}}\left[F_N(\widetilde{x}^T) - F_N(\widehat{x})\right] = \mathcal{O}\left(\frac{\ln N}{T^{\alpha_{\mathrm{true}}-1}}\right), \tag{77}$$

   *the second bound is optimal.*

2. *If $N \approx T$, the optimal choice is $\alpha_{\mathrm{alg}} = 2$, which results in overall bound*

$$\mathbb{E}_{\mathrm{alg}}\left[F_N(\widetilde{x}^T) - F_N(\widehat{x})\right] = \mathcal{O}\left(\frac{1}{T^{\frac{2(\alpha_{\mathrm{true}}-1)}{\alpha_{\mathrm{true}}}}}\right), \tag{78}$$

   *the second bound is optimal.*

3. *If $N \ll T$, the optimal choice is $\alpha_{\mathrm{alg}} = 2$, which results in overall bound*

$$\mathbb{E}_{\mathrm{alg}}\left[F_N(\widetilde{x}^T) - F_N(\widehat{x})\right] = \mathcal{O}\left(\frac{1}{T}\right), \tag{79}$$

   *both bounds are optimal.*

*Proof.* If $N \gg T$, then, in order to mitigate the effect of rising $N$ one should choose $\alpha_{\mathrm{true}} = \alpha_{\mathrm{alg}}$, therefore, the second bound will be optimal, since it includes $\ln N$ and not $(\ln N)^2$.

If $N \ll T$, then the optimal choice is the one with the best asymptotic in $T$. For both cases this is $\alpha_{\mathrm{alg}} = 2$, which results in $\mathcal{O}(1/T)$.

If $N \approx T$, then, with $\alpha_{\mathrm{alg}} = \alpha_{\mathrm{true}}$ the bound becomes

$$\mathcal{O}\left(\min\left\{\frac{(\ln T)^2}{T^{\frac{2(\alpha_{\mathrm{true}}-1)}{\alpha_{\mathrm{true}}}}}; \frac{\ln T}{T^{\alpha_{\mathrm{true}}-1}}\right\}\right).$$

With $\alpha_{\mathrm{alg}} > \alpha_{\mathrm{true}}$ the second bound becomes proportional to the following:

$$\frac{N^{\frac{\alpha_{\mathrm{alg}}}{\alpha_{\mathrm{true}}}-1}}{T^{\alpha_{\mathrm{alg}}-1}} \approx \frac{T^{\frac{\alpha_{\mathrm{alg}}}{\alpha_{\mathrm{true}}}-1}}{T^{\alpha_{\mathrm{alg}}-1}} = T^{\frac{\alpha_{\mathrm{alg}}}{\alpha_{\mathrm{true}}}-1-\alpha_{\mathrm{alg}}+1} = \frac{1}{T^{\alpha_{\mathrm{alg}}\frac{\alpha_{\mathrm{true}}-1}{\alpha_{\mathrm{true}}}}},$$

which reaches its minimum with $\alpha_{\mathrm{alg}} = 2$. The first bound, on the other way, with $\alpha_{\mathrm{alg}} > \alpha_{\mathrm{true}}$ becomes the following:

$$\frac{N^{\frac{2\alpha_{\mathrm{alg}}}{\alpha_{\mathrm{true}}}-2}}{T^{\frac{2(\alpha_{\mathrm{alg}}-1)}{\alpha_{\mathrm{alg}}}}} \approx \frac{T^{\frac{2\alpha_{\mathrm{alg}}}{\alpha_{\mathrm{true}}}-2}}{T^{\frac{2(\alpha_{\mathrm{alg}}-1)}{\alpha_{\mathrm{alg}}}}} = T^{\frac{2\alpha_{\mathrm{alg}}^2-4\alpha_{\mathrm{alg}}\alpha_{\mathrm{true}}+2\alpha_{\mathrm{true}}}{\alpha_{\mathrm{true}}\alpha_{\mathrm{alg}}}}.$$

The minimum of this power is obtained at $\alpha_{\mathrm{alg}} = \sqrt{\alpha_{\mathrm{true}}}$, which is less than $\alpha_{\mathrm{true}}$. Substituting $\alpha_{\mathrm{alg}} = \alpha_{\mathrm{true}}$ we obtain $T^{\frac{2(1-\alpha_{\mathrm{true}})}{\alpha_{\mathrm{true}}}}$, therefore, with $\alpha_{\mathrm{alg}} > \alpha_{\mathrm{true}}$, this bound will be worse. □

Similar to the convex case, one can additionally tune the $\lambda_0$ as following:

$$\lambda_0 = \left(\frac{1}{N}\sum_{i=1}^{N}\|\nabla f_i(x_\star)\|^{\alpha_{\mathrm{alg}}}\right)^{1/\alpha_{\mathrm{alg}}}, \tag{80}$$

which improves the convergence bounds with $\lambda_t = t^{1/\alpha_{\mathrm{alg}}}$.

**Lemma D.6.** *Let all the Assumptions for the D.4 be satisfied, and $\lambda_0$ is tuned as in equation 80. Then, for 3 different regimes based on the relation between $N$ and $T$ we have the following optimal choice of $\alpha_{\mathrm{alg}}$:*

1. *If $N \gg T$, the optimal choice is $\alpha_{\mathrm{alg}} = \alpha_{\mathrm{true}}$, which results in overall bound*

$$\mathbb{E}_{\mathrm{alg}}\left[F_N(\widetilde{x}^T) - F_N(\widehat{x})\right] = \mathcal{O}\left(\frac{\ln N}{T^{\alpha_{\mathrm{true}}-1}}\right), \tag{81}$$

*the second bound is optimal.*

2. *If $N \approx T$, the optimal choice is $\alpha_{\mathrm{alg}} > \alpha_{\mathrm{true}}$, which results in overall bound*

$$\mathbb{E}_{\mathrm{alg}}\left[F_N(\widetilde{x}^T) - F_N(\widehat{x})\right] = \mathcal{O}\left(\frac{1}{T^{\frac{2(\alpha_{\mathrm{true}}-1)}{\alpha_{\mathrm{true}}}}}\right), \tag{82}$$

*the first bound is optimal. The second bound matches it with $\alpha_{\mathrm{alg}} = 2$.*

3. *If $N \ll T$, the optimal choice is $\alpha_{\mathrm{alg}} = 2$, which results in overall bound*

$$\mathbb{E}_{\mathrm{alg}}\left[F_N(\widetilde{x}^T) - F_N(\widehat{x})\right] = \mathcal{O}\left(\frac{1}{T}\right), \tag{83}$$

*both bounds are optimal.*

*Proof.* From Theorem C.4 the following convergence rates, including $\lambda_0$ can be derived:

$$\mathbb{E}_{\mathrm{alg}}\left[F_N(\widetilde{x}^T) - F_N(\widehat{x})\right] = \mathcal{O}\left(\frac{L^2\Delta^2}{\mu T^2} + \frac{\lambda_0^{2(1-\alpha_{\mathrm{alg}})}\left(\frac{1}{N}\sum_{i=1}^{N}\|\nabla f(x_\star, \xi_i)\|^{\alpha_{\mathrm{alg}}}\right)^2}{\mu T^{\frac{2(\alpha_{\mathrm{alg}}-1)}{\alpha_{\mathrm{alg}}}}} + \frac{\lambda_0^{2-\alpha_{\mathrm{alg}}}\left(\frac{1}{N}\sum_{i=1}^{N}\|\nabla f(x_\star, \xi_i)\|^{\alpha_{\mathrm{alg}}}\right)}{\mu T^{\frac{2(\alpha_{\mathrm{alg}}-1)}{\alpha_{\mathrm{alg}}}}}\right).$$

This equation is minimized with $\lambda_0$ chosen as in equation 80, with resulting bound

$$\mathbb{E}_{\mathrm{alg}}\left[F_N(\widetilde{x}^T) - F_N(\widehat{x})\right] = \mathcal{O}\left(\frac{L^2\Delta^2}{\mu T^2} + \frac{\left(\frac{1}{N}\sum_{i=1}^{N}\|\nabla f(x_\star, \xi_i)\|^{\alpha_{\mathrm{alg}}}\right)^{\frac{2}{\alpha_{\mathrm{alg}}}}}{\mu T^{\frac{2(\alpha_{\mathrm{alg}}-1)}{\alpha_{\mathrm{alg}}}}}\right).$$

With $\alpha_{\mathrm{alg}} < \alpha_{\mathrm{true}}$ this scales as

$$\frac{\left(\mathbb{E}\|\nabla f(x_\star, \xi)\|^{\alpha_{\mathrm{alg}}}\right)^{\frac{2}{\alpha_{\mathrm{alg}}}}}{T^{\frac{2(\alpha_{\mathrm{alg}}-1)}{\alpha_{\mathrm{alg}}}}}.$$

With $\alpha_{\mathrm{alg}} = \alpha_{\mathrm{true}}$ this scales as

$$\frac{(\ln N)^{\frac{2}{\alpha_{\mathrm{true}}}}}{T^{\frac{2(\alpha_{\mathrm{true}}-1)}{\alpha_{\mathrm{true}}}}}.$$

With $\alpha_{\mathrm{alg}} > \alpha_{\mathrm{true}}$ this scales as

$$\frac{N^{\left(\frac{\alpha_{\mathrm{alg}}}{\alpha_{\mathrm{true}}}-1\right)\frac{2}{\alpha_{\mathrm{alg}}}}}{T^{\frac{2(\alpha_{\mathrm{alg}}-1)}{\alpha_{\mathrm{alg}}}}}.$$

Cases $N \gg T$ and $N \ll T$ do not change. However, with $N \approx T$ for $\alpha_{\mathrm{alg}} > \alpha_{\mathrm{true}}$ the first bound now scales as

$$\frac{N^{\left(\frac{\alpha_{\mathrm{alg}}}{\alpha_{\mathrm{true}}}-1\right)\frac{2}{\alpha_{\mathrm{alg}}}}}{T^{\frac{2(\alpha_{\mathrm{alg}}-1)}{\alpha_{\mathrm{alg}}}}} \approx \frac{T^{\left(\frac{\alpha_{\mathrm{alg}}}{\alpha_{\mathrm{true}}}-1\right)\frac{2}{\alpha_{\mathrm{alg}}}}}{T^{\frac{2(\alpha_{\mathrm{alg}}-1)}{\alpha_{\mathrm{alg}}}}} = \frac{1}{T^{\frac{2(\alpha_{\mathrm{true}}-1)}{\alpha_{\mathrm{true}}}}}.$$

$\square$

