# OpenReview forum: "From Optimization to Generalization under Heavy-Tailed Data: The Role of Gradient Clipping"
_ICML.cc/2026/Conference — ICML 2026 regular_

### Official Review · Reviewer_7SNV · 2026-03-11

**Soundness:** 3
**Presentation:** 3
**Significance:** 2
**Originality:** 2
**Overall Recommendation:** 4
**Confidence:** 4

**Summary:**

The paper discusses stochastic convex optimization under heavy-tailed data. In this setting, the authors show convergence bounds for SGD with gradient clipping (with and without an additional assumption of strong convexity). In addition, they show upper and lower bounds for the generalization of ERM in the strongly convex regime.

**Compliance With Llm Reviewing Policy:**

Affirmed.

**Final Justification:**

My concerns have been resolved. However, the paper discusses the constrained case, and the authors state in their rebuttal that they will address the case of $\nabla F(x_\star) \neq 0$. I maintain my score of 4.

**Key Questions For Authors:**

1. In Theorem 3.6, the bounds can become arbitrarily bad as $q\to 1$. Do the authors have intuition for the reason for that?

2. Is it possible to achieve the improved bounds using clipping without tuning $\alpha_{alg}= \alpha_{true}$ (e.g., by using doubling or adaptive learning rate)?

3. Do the statements and proofs are transferable to the case of $\nabla F(x_\star) \neq 0$?
First,  this fact is being used at the proofs.
Second, if the norm of this gradient is very large, it does not make sense to use clipping since by the smoothness and the boundedness of the domain, all gradients are large.

4. Do the statements and proofs are transferable to unbounded domain, e.g., with $\Delta=\|x^0-\hat x\|$?

5. A typo: In some places (e.g., Theorem 3.6) there are references to Theorems instead of Assumptions.

**Limitations:**

yes

**Strengths And Weaknesses:**

Strengths

1. The paper is well written and easy to follow. The authors explain the significance of their results.
2. The authors demonstrate the effect of gradient clipping and show that when $N>>T$, it improves the rate of SGD.
3. The ERM results are interesting and extend previous work from the stochastic convex optimization literature.

Weaknesses

1. It is not clear whether the work discusses constrained optimization or unconstrained optimization. It seems that the assumptions and results are not consistent with either choice. On one hand, the authors assume Assumption 3.4 of compact domain and use the diameter $\Delta$ in their bounds. On the other hand, in line 169, they state that $\nabla F(x_\star)=0$, which generally true only in the unconstrained setting. The authors use this fact in the proofs. Please see questions 3,4 below.

2. The improvements in the rate of SGD are only in regime where the hyperparameters are tuned according to $\alpha_{true}$. For this,
the algorithm needs to know what is $\alpha_{true}$ which does not seemed to be a reasonable assumption.

3. The analysis of the ERM results is based on the analysis given in previous work.

---

> ### Author Rebuttal · Authors · 2026-03-31
>
> Thank you for the careful review and for highlighting the interesting ERM contribution and the $N\gg T$ regime.
>
> ---
>
> **W1 (constrained vs unconstrained setting).** You are right that the current wording mixes two viewpoints. The intended setting of Theorems 3.8/3.14 is **projected ClipSGD on a compact convex set** $X$ (Algorithm 1 already includes $proj_X$), so the discussion should not implicitly rely on $\nabla F(x_\star)=0$. We will revise Assumption 3.5 to be stated directly for the *centered* gradient at $x_\star$, i.e., $\nabla f(x_\star,\xi)-\nabla F(x_\star)$, which is the natural object in constrained problems as well. The role of $\Delta$ is to control clipping bias after projection.
>
> ---
>
> **W2 (need to know $\alpha_{\rm true}$?).** Exact knowledge of $\alpha_{\rm true}$ is only needed to optimize the $T$-dependence. The main qualitative benefit of clipping does **not** require exact matching. Corollary 3.11 already shows the robustness: if $\alpha_{\rm alg}<\alpha_{\rm true}$, the harmful polynomial dependence on $N$ disappears; if $\alpha_{\rm alg}=\alpha_{\rm true}$, it becomes logarithmic; only if $\alpha_{\rm alg}>\alpha_{\rm true}$ does polynomial $N$-growth reappear. So conservative tuning already protects against dataset-size deterioration, even if it is not $T$-optimal. This is consistent with prior heavy-tail clipping work, where exact tail information is mainly needed for rate-optimal tuning rather than for mere convergence [1,2,7].
>
> ---
>
> **W3 (ERM analysis uses previous ideas).** We agree that the leave-one-out / stability coupling is not new by itself [4,5]. Our claim is not novelty of each ingredient, but of the combination: heavy-tail asymptotics at the population minimizer plus finite-sum analysis yield, to our knowledge, the first picture that simultaneously gives (i) ERM generalization bounds with explicit heavy-tail dependence [6], (ii) a matching lower bound, and (iii) an explanation of why clipping improves the optimization/generalization interplay by removing the harmful growth of the classical finite-sum noise proxy.
>
> ---
>
> **Q1 (why the bound worsens as $q\to1$).** Intuitively, $q\to1$ means tails so heavy that only moments barely above first order exist. Then rare extremes dominate sample averages, so one cannot expect near-$1/N$ generalization. The theorem reflects this through the rate $N^{-(q-1)}$, consistent with known lower-bound phenomena in heavy-tailed stochastic convex optimization [3]. When $\alpha$ is closer to 2, we can choose $q$ closer to 2 and recover rates close to the classical $1/N$ behavior.
>
> ---
>
> **Q2 (improvement without exact tuning / adaptivity).** Exact optimal tuning is interesting and largely open. What our current results already show is that exact matching is not necessary for the *qualitative* gain over SGD: choosing $\alpha_{\rm alg}\le\alpha_{\rm true}$ already removes or greatly weakens the bad $N$-dependence. Data-driven/adaptive tuning is a natural future direction.
>
> ---
>
> **Q3 (transfer to $\nabla F(x_\star)\neq0$).** The natural statement is to center the tail assumption at $\nabla F(x_\star)$, not to assume it vanishes. This would introduce additional dependency on $\|\nabla F(x_\star)\|$, which is another reason the constrained and unconstrained cases should be separated more carefully. We have proven the theorems without this assumption and will include proofs in the next version.
>
> ---
>
> **Q4 (unbounded domains).** Our current proofs use bounded $X$ to control clipping bias through $\Delta$. We believe similar ideas can be extended to unbounded domains under additional control of the iterates (e.g., regularization or distance-to-optimum bounds), in the spirit of prior heavy-tailed analyses [7,8], but that extension is outside the present scope.
>
> ---
>
> **Q5 (typos).** Thank you -- we will fix the Theorem/Assumption cross-references throughout.
>
> ---
>
> **References**
>
> [1] E. Gorbunov, M. Danilova, and A. Gasnikov, “Stochastic Optimization with Heavy-Tailed Noise via Accelerated Gradient Clipping,” NeurIPS, 2020.
>
> [2] A. Koloskova, H. Hendrikx, and S. U. Stich, “Revisiting Gradient Clipping: Stochastic Bias and Tight Convergence Guarantees,” ICML, 2023.
>
> [3] N. M. Vural et al., “Mirror Descent Strikes Again: Optimal Stochastic Convex Optimization under Infinite Noise Variance,” COLT, 2022.
>
> [4] O. Bousquet and A. Elisseeff, “Stability and Generalization,” JMLR, 2002.
>
> [5] S. Shalev-Shwartz, O. Shamir, N. Srebro, and K. Sridharan, “Learnability, Stability and Uniform Convergence,” JMLR, 2010.
>
> [6] H. Liu and J. Tong, “New Sample Complexity Bounds for Sample Average Approximation in Heavy-Tailed Stochastic Programming,” ICML, 2024.
>
> [7] T. D. Nguyen, T. H. Nguyen, A. Ene, and H. Nguyen, “Improved Convergence in High Probability of Clipped Gradient Methods with Heavy Tailed Noise,” NeurIPS, 2023.
>
> [8] A. Sadiev et al., “High-Probability Bounds for Stochastic Optimization and Variational Inequalities: the Case of Unbounded Variance,” ICML, 2023.

---

> > ### Author Rebuttal · Reviewer_7SNV · 2026-04-02
> >
> > Thank you for the detailed response. I maintain my positive score.

---

> > > ### Author Response · Authors · 2026-04-06
> > >
> > > Thank you for the thoughtful feedback on our work and acknowledging our responses. We are glad that we have addressed all your concerns, and would be grateful, if you champion our paper in the AC-Reviewer discussion.

---

### Official Review · Reviewer_h7cp · 2026-03-11

**Soundness:** 3
**Presentation:** 3
**Significance:** 3
**Originality:** 2
**Overall Recommendation:** 4
**Confidence:** 3

**Summary:**

This paper considers convergence error bounds for clipped SGD. In particular, the paper, provides a strong rates for the difference between the function value and the mean parameter (mean over the trajectory) and the function value at the converged point. The paper shows that the rates depend on the heavy tail index of the distribution at the optimal.

**Compliance With Llm Reviewing Policy:**

Affirmed.

**Key Questions For Authors:**

1. What is the formal meaning of $\sim$ in Assumption 3.5

2. In Theorem 3.8 what does $\mathbb{E}_i$ mean? It says it is the average across the sample size. But in my understanding $N$ is fixed and we always use one sample per iteration.

3. In the same theorem you use the notation $f_i(x_{star})$ what is this?

Typos:

Lien 206 Theorem 3.6 it says assume Theorem 3.2-3.5 hold. You mean assumptions.

**Strengths And Weaknesses:**

**Strengths**

The paper is reasonably well written and provides some strong results on convergence and generalization. In general, I quite like the results and think they are good contributions to the machine learning literature. However, I have a couple of concerns listed below as weaknesses.

The regimes based on the amount of data versus the number of epochs is quite interesting.

**Weakness**

1. Since clipped SGD is primarily used to train Neural Networks (which seems to be the motivation for this paper as well), Assumptions 3.1 and 3.2 are quite unrealistic. Neural networks are functions of the parameters are highly non-convex.

2. The results focus on the difference between $F_N(\bar{x})$ and $F_N(\hat{x})$. However, I have two concerns with this quantity. First is the use of $\bar{x}$ which is the mean of all of the iterates seen during the training. Is it not more natural to use $x_T$. Can the authors comment on bounding $\mathbb{E}_{alg}[F_N(x_T) - F_N(\hat{x})]$?

Second, this is convergence rate of the expected training error. However, can we say something with high probability or about the generalization error in this setting.

---

> ### Author Rebuttal · Authors · 2026-03-31
>
> Thank you for the positive assessment of the paper’s contributions on convergence/generalization and for highlighting the interesting $N$-vs-epochs regimes. We appreciate the constructive questions below.
>
> ---
>
> **W1 (convex / strongly convex assumptions vs neural networks).** We agree that these assumptions do not model full deep-network training. Our goal is more modest: to give a first clean optimization-to-generalization theory for finite-sum minimization under heavy-tailed sampling, in a regime where the ERM/population-minimizer relation can be analyzed sharply. Convex and strongly convex models are still important in their own right (e.g., linear/logistic models and regularized generalized linear models), and they are the standard starting point in heavy-tailed optimization theory [1–3]. We will revise the motivation to make this scope clearer and avoid suggesting that the theorems directly cover general deep-network training.
>
>
> ---
>
> **W2 (average iterate vs last iterate).** This is an excellent point. We use the averaged iterate because it gives the cleanest finite-sum guarantee under clipping and heavy-tailed noise, and because our main phenomenon—the dataset-size dependence of the noise-at-the-optimum term—does not rely on averaging per se. Classical convex SGD analyses are also typically stated for averaged iterates [4]. We agree that the final iterate is often more natural in practice, but last-iterate guarantees are technically more delicate, so we chose to focus the paper on the heavy-tail / finite-sum effect rather than on the iterate-selection issue. We will add a brief discussion clarifying this point.
>
>
> ---
>
> **W3 (high probability / generalization).** For generalization, yes: Theorem 3.6 already provides both an in-expectation and a high-probability ERM bound. For the optimization part, the current paper gives in-expectation guarantees; extending the finite-sum clipping analysis to high probability is very interesting, and existing high-probability clipped/robust analyses suggest this is feasible [2,5,6], but we did not include that extension in the current draft in order to keep the paper focused. We will make this distinction clearer.
>
>
> ---
>
> **Q1 (meaning of $\sim$ in Assumption 3.5).** We mean regular variation in the standard sense:
> $$
> \lim_{r\to\infty} \frac{\Pr(\|\nabla f(x_\star,\xi)\|\ge r)}{c r^{-\alpha}} = 1.
> $$
> We will state this explicitly.
>
>
> ---
>
> **Q2 (meaning of $\mathbb E_i$ in Theorem 3.8).** Here $\mathbb E_i$ denotes the empirical average over the fixed sampled dataset, i.e.,
> $$
> \mathbb E_i\,\|\nabla f_i(x_\star)\|^\alpha = \frac1N\sum_{i=1}^N \|\nabla f(x_\star,\xi_i)\|^\alpha,
> $$
> with $f_i(\cdot)=f(\cdot,\xi_i)$. The randomness in $\mathbb E_{\rm alg}$ is then only from the index sampling during optimization. We agree this notation should be defined more explicitly.
>
>
> ---
>
> **Q3 (notation $f_i(x_\star)$).** Yes—this is shorthand for $f(x_\star,\xi_i)$. We will add this notation immediately before the theorem to avoid ambiguity.
>
> ---
>
> **References**
>
> [1] Y. Nesterov, *Lectures on Convex Optimization*, Springer, 2018.
>
> [2] E. Gorbunov, M. Danilova, and A. Gasnikov, “Stochastic Optimization with Heavy-Tailed Noise via Accelerated Gradient Clipping,” NeurIPS, 2020.
>
> [3] H. Liu and J. Tong, “New Sample Complexity Bounds for Sample Average Approximation in Heavy-Tailed Stochastic Programming,” ICML, 2024.
>
> [4] E. Moulines and F. Bach, “Non-Asymptotic Analysis of Stochastic Approximation Algorithms for Machine Learning,” NeurIPS, 2011.
>
> [5] T. D. Nguyen, T. H. Nguyen, A. Ene, and H. Nguyen, “Improved Convergence in High Probability of Clipped Gradient Methods with Heavy Tailed Noise,” NeurIPS, 2023.
>
> [6] A. Sadiev et al., “High-Probability Bounds for Stochastic Optimization and Variational Inequalities: the Case of Unbounded Variance,” ICML, 2023.

---

> > ### Author Rebuttal · Reviewer_h7cp · 2026-04-03
> >
> > Thank you for clarifying the questions. I maintain my positive score of weak accept. I would be willing to go to accept if the non-convex setting or the last iterate setting is analyzed.

---

> > > ### Author Response · Authors · 2026-04-06
> > >
> > > We thank the reviewer for the follow-up and are glad that our previous response clarified the main questions. We also appreciate the suggestion that analyzing either the last-iterate setting or the non-convex setting would further strengthen the paper. We briefly comment on both directions below.
> > >
> > > ---
> > >
> > > **On the last-iterate analysis.** The last-iterate setting is indeed both practically relevant and theoretically appealing. However, in the heavy-tailed regime, existing last-iterate analyses differ substantially across problem classes, and extending our results is not straightforward. In particular, the techniques used in our convex and strongly convex analysis are somewhat non-standard for the heavy-tailed literature, since they rely on isolating the dataset-dependent term $\frac{1}{N}\sum_{i=1}^N ||\nabla f_i(x_\star)||^{\alpha}$, rather than directly analyzing central moments. Developing a last-iterate analysis within this framework is therefore a challenging direction, and we view it as an interesting topic for future work.
> > >
> > > ---
> > >
> > > **On the non-convex setting.** Regarding the non-convex case, the literature is broad but fragmented. Some works study the stability of SGD in non-convex optimization [1, 2], but these results are not algorithm-agnostic, as they depend on the specific update rule and typically require stronger assumptions, such as uniformly bounded gradients. Other works on stochastic average approximation in the non-convex setting establish asymptotic convergence rates [3-5], which are qualitatively weaker than the non-asymptotic guarantees proved in our paper.
> > >
> > > We therefore view the present contribution - a non-asymptotic analysis of both stochastic average approximation and optimization under heavy-tailed data in the convex and strongly convex settings - as self-contained. Extending this framework to the non-convex case would likely require substantially different ideas and would, in our view, warrant a separate paper.
> > >
> > > ---
> > >
> > > [1] Hardt, Moritz, Ben Recht, and Yoram Singer. "Train faster, generalize better: Stability of stochastic gradient descent." International conference on machine learning. PMLR, 2016.
> > >
> > > [2] Zhou, Yi, Yingbin Liang, and Huishuai Zhang. "Understanding generalization error of SGD in nonconvex optimization." Machine Learning 111.1 (2022): 345-375.
> > >
> > > [3] Bastin, Fabian, Cinzia Cirillo, and Philippe L. Toint. "Convergence theory for nonconvex stochastic programming with an application to mixed logit." Mathematical Programming 108.2 (2006): 207-234.
> > >
> > > [4] Xu, Huifu, and Dali Zhang. "Smooth sample average approximation of stationary points in nonsmooth stochastic optimization and applications." Mathematical programming 119.2 (2009): 371-401.
> > >
> > > [5] Banholzer, Dirk, Jörg Fliege, and Ralf Werner. "On rates of convergence for sample average approximations in the almost sure sense and in mean." Mathematical Programming 191.1 (2022): 307-345.

---

### Official Review · Reviewer_f8XY · 2026-03-13

**Soundness:** 3
**Presentation:** 4
**Significance:** 2
**Originality:** 2
**Overall Recommendation:** 4
**Confidence:** 2

**Summary:**

This paper studies gradient clipping in finite-sum empirical risk minimization under heavy-tailed data. Its main goal is to resolve the apparent mismatch between the standard heavy-tailed-noise view of stochastic optimization and the fact that, for any fixed dataset, all moments are finite. The paper argues that the key issue is the separation between data sampling and optimization randomness: although the variance proxy is finite conditional on a dataset, under heavy-tailed sampling it can still grow with dataset size \(N\), which hurts standard SGD. The authors show that gradient clipping can alleviate this harmful dataset-size dependence, derive convergence guarantees for convex and strongly convex settings under broad clipping and step-size schedules, and establish generalization bounds linked to the tail behavior of gradients at the population minimizer. The paper also highlights different optimization regimes depending on the relationship between \(N\) and \(T\), and supports its claims with synthetic experiments showing that clipping improves estimation behavior especially when the dataset is large relative to the optimization budget.

**Compliance With Llm Reviewing Policy:**

Affirmed.

**Key Questions For Authors:**

- In Theorem 3.6, the relationship between \(q\) and \(N\) seems somewhat coupled. I would appreciate it if the authors could discuss this point more clearly.
- In practical large-scale training, gradient clipping often seems to be used not primarily as a variance-reduction method, but rather as a tool for stabilizing training. How do the authors view the apparent tension between this practical motivation and the conclusions of the paper?
- Is there practical evidence supporting the dataset-scale effect claimed by the authors? For example, under similar model scales, do multi-epoch training tasks and near-streaming training tasks tend to use different standard gradient clipping scales?

**Limitations:**

Beyond the issues mentioned above, I do not have additional concerns regarding limitations.

**Strengths And Weaknesses:**

### Strengths

- Most of the assumptions made in the paper are fairly standard.
- The paper is written in a concise and accessible way, and the overall presentation is very clear.
- The paper discusses many interesting properties of clipping, including how tail gradient noise can affect generalization, and how clipping can improve convergence even when the tail distribution of the gradient is unknown. These are all very interesting observations.
- The paper provides theoretical guarantees regarding the stability of the clipping parameter.
- The paper emphasizes that its main novelty lies in the finite-dataset setting. In this respect, it does provide meaningful theoretical distinctions under different \(N/T\) ratios, and the experiments also support this point.


### Weaknesses

- The assumption of a polynomially decaying tail is not very clearly justified or empirically verified. In real large-scale training, are the gradients that get clipped truly coming from the tail of a stable polynomial-tailed distribution, or are they simply a few extreme outliers?
- The experiments are still conducted in relatively basic settings. Since part of the core contribution of the paper concerns hyperparameter selection, I would still like to see experimental results on more realistic datasets to better validate the practical relevance of the theory.

### Minor Comments

- Line 202: “theorem” should be changed to “assumptions.”

---

> ### Author Rebuttal · Authors · 2026-03-31
>
> Thank you for the thoughtful review and for highlighting the clarity of the presentation, the meaningful finite-dataset novelty, and the interesting observations about clipping and generalization. We appreciate these comments.
>
> ---
>
> **W1 (justification of polynomial tails / outliers vs true tails).** We agree that the paper should be clearer here. Our theory does **not** assume that practical training gradients follow an exact stable law. The assumption is only *regular variation* of the tail, which is strictly weaker and is used because it yields sharp finite-sum scaling. Empirically, several works report heavy-tailed behavior of SGD-related quantities [1–3], but we agree that a universal “exact polynomial tail law” for all models/tasks would be too strong a claim. The right interpretation of our assumption is therefore: it is a canonical model of regimes where rare extreme samples dominate empirical second moments, and it is precisely in that regime that our dataset-size effect becomes sharp. We will state this caveat explicitly.
>
>
> ---
>
> **W2 (experiments are basic).** This is fair. Our experiments were designed to isolate the *specific mechanism* that is novel in the paper—how the dataset-size dependence differs for SGD and ClipSGD at fixed optimization budget—without confounding factors from architecture or optimizer heuristics. That said, we agree that a more realistic experiment would strengthen the paper. In the revised version / camera-ready, we plan to add a larger-scale experiment that varies $N$ and the number of passes for the same model while comparing tuned vs agnostic clipping schedules. We will also clarify that the current experiments are meant as validation of the theory’s qualitative prediction, not as a comprehensive empirical benchmark.
>
>
> ---
>
> **Q1 (coupling between $q$ and $N$ in Theorem 3.6).** The parameter $q\in(1,\alpha)$ is an analysis knob: larger $q$ gives a faster rate $N^{-(q-1)}$, but it also requires a higher finite moment. The lower bound on $N$ in Theorem 3.6 is used only to simplify the displayed bound by absorbing the smoothness-generated higher-order term that appears in the proof. This is standard in the literature [4]. In fact, this is not essential: Appendix A (Cor. A.4) gives a threshold-free version with one additional faster-decaying term. We will make this explicit in the statement and discussion so the role of $q$ is easier to interpret.
>
>
> ---
>
> **Q2 (clipping for stabilization vs variance reduction).** We do not see a tension here. In practice, clipping is indeed often used as a stabilization device; in our view, heavy-tailed theory explains one important mechanism behind that stabilization. Rare but very large gradients are exactly what make second-moment-based SGD guarantees deteriorate, while clipping suppresses their influence. So “stabilizing training” and “mitigating heavy-tailed noise” are largely complementary views of the same phenomenon [5–7]. We also agree that clipping can help for other reasons (e.g., exploding gradients) [5], and we will revise the discussion so the paper is framed as isolating **one** mechanism, not claiming exclusivity.
>
>
> ---
>
> **Q3 (practical evidence for the dataset-scale effect).** At present, we are not aware of a systematic empirical study showing a universal law for the optimal clipping threshold as a function of the $N/T$ regime. We therefore do not want to overclaim. Our result should be interpreted as a *theoretical prediction*: for fixed compute budget, larger datasets can make the classical SGD noise proxy worse, while clipping remains robust. This prediction is directionally consistent with empirical heavy-tail studies [1–3] and with the widespread use of clipping in unstable large-scale training [5,6], but the exact dependence of tuned clipping thresholds on $N/T$ remains an open experimental question. We will phrase this more carefully.
>
> ---
>
> **References**
>
> [1] M. Gürbüzbalaban, U. Şimşekli, and L. Zhu, “The Heavy-Tail Phenomenon in SGD,” ICML, 2021.
>
> [2] M. Barsbey et al., “Heavy Tails in SGD and Compressibility of Overparametrized Neural Networks,” NeurIPS, 2021.
>
> [3] Z. Jiao and M. Keller-Ressel, “Emergence of Heavy Tails in Homogenized Stochastic Gradient Descent,” NeurIPS, 2024.
>
> [4] H. Liu and J. Tong, “New Sample Complexity Bounds for Sample Average Approximation in Heavy-Tailed Stochastic Programming,” ICML, 2024.
>
> [5] R. Pascanu, T. Mikolov, and Y. Bengio, “On the Difficulty of Training Recurrent Neural Networks,” ICML, 2013.
>
> [6] J. Zhang et al., “Why Are Adaptive Methods Good for Attention Models?,” NeurIPS, 2020.
>
> [7] E. Gorbunov, M. Danilova, and A. Gasnikov, “Stochastic Optimization with Heavy-Tailed Noise via Accelerated Gradient Clipping,” NeurIPS, 2020.

---

### Official Review · Reviewer_RugF · 2026-03-18

**Soundness:** 3
**Presentation:** 3
**Significance:** 3
**Originality:** 3
**Overall Recommendation:** 3
**Confidence:** 4

**Summary:**

This paper studies finite-sum minimization under heavy-tailed gradient noise at the global distributional minizer. Generalization bounds, and convergence guarantees under ClippedSGD for convex, and strongly convex smooth functions are presented. Numerical experiments validate that clippedSGD can reduce generalization errors compared to vanilla SGD.

**Compliance With Llm Reviewing Policy:**

Affirmed.

**Key Questions For Authors:**

1. $\sigma_{2, N}$ in equation (5) is dependent on $\hat{x}$, even it is finite for a fixed $\hat{x}$, it can be unbounded for $\hat{x} \in \mathbb{R}^d$.

**Strengths And Weaknesses:**

Strengths:
1. This paper studies sampling noise for heavy-tailed data and shed light on how heavy-tailed noise is incurred.
2. The gradient noise assumption is made only on the distributional global minimum.

Weaknesses:
1. What is heavy-tailed data is undefined in the introduction, and there lacks motivations on this part.
2. Maybe give some examples why this particular noise model is considered.
3. Bounded domain is assumed, which seems to me make the results restrictive. Is this work the first one to study the generalization/optimization bounds for finite-sum minimization under heavy-tailed noise? If not, a more careful contrastion is needed.
4. The considered noise model is limited to the polynomial tail case.

---

> ### Author Rebuttal · Authors · 2026-03-31
>
> Thank you for the careful review and for highlighting the paper’s soundness, presentation, significance, and originality. We are encouraged by these assessments, and we hope the clarifications below address the main concerns.
>
> ---
>
> **W1 (definition / motivation).** In our paper, “heavy-tailed data” means that sampling $\xi$ induces a heavy-tailed distribution of gradients at the population minimizer, formalized in Assumption 3.5 as $\Pr(\|\nabla f(x_\star,\xi)\|\ge r)\sim c r^{-\alpha}$, $\alpha\in(1,2]$. This is closely related to the standard heavy-tailed-noise condition in Eq. (4) and is the regime studied in prior optimization work [1,4,6]. We agree that this should be stated much earlier in the Introduction. The motivation is exactly the paradox our paper addresses: conditional on a fixed dataset all moments are finite, yet heavy-tailed sampling makes the dataset-dependent noise proxy relevant for finite-sum SGD grow with $N$ [1–3].
>
>
> ---
>
> **W2 (why this noise model).** We use regularly varying / polynomial tails for two reasons. First, this is a canonical and broad heavy-tail model: it is weaker than assuming an exact stable distribution and covers domains of attraction of $\alpha$-stable laws [1,4]. Second, it is precisely what allows sharp asymptotics for empirical quantities such as $\frac1N\sum_i \|\nabla f(x_\star,\xi_i)\|^2$, which is the key step behind Theorem 3.10. If one assumes only the existence of a low-order moment, one can still obtain moment-based guarantees, but not the sharp $N$-dependence that is central in our paper. We will make this justification clearer and add concrete examples (e.g., Pareto / regularly varying tails).
>
>
> ---
>
> **W3 (bounded domain / contrast to prior work).** The bounded-domain assumption is mainly technical: it lets us control the clipping bias uniformly through $\Delta$ and keep the optimization/generalization statements clean. We agree that this limitation should be stated more prominently. At the same time, the paper already contains a regularized convex extension in Appendix A, which partially relaxes this issue. On novelty: to the best of our knowledge, prior heavy-tailed clipping papers analyze the streaming setting or assume noise conditions along the whole trajectory [4,6], whereas recent heavy-tailed SAA/ERM results focus on the streaming oracle case rather than finite-sum optimization [5]. Our contribution is to explicitly separate data sampling and optimization randomness in the finite-sum regime, derive the resulting $N$-dependent growth of the classical noise-at-the-optimum term, and connect this optimization effect to generalization.
>
>
> ---
>
> **W4 (polynomial tails only).** We agree that the current theory is tailored to regularly varying tails. This is a deliberate modeling choice, because the stable-law machinery is what yields the sharp scaling in Theorem 3.10. We do not mean to suggest that other weak-tail models are unimportant; rather, regularly varying tails are the canonical setting in which extreme samples dominate averages, and hence the cleanest setting for the finite-sum effect we study [1–4]. We will state this scope more explicitly.
>
>
> ---
>
> **Q1 (role of $\sigma_{2,N}$ and dependence on $\widehat x$).** We completely agree that $\sigma_{2,N}$ should not be interpreted as uniformly bounded over all $\widehat x\in\mathbb R^d$. Our point is different: for the ERM minimizer of a fixed sampled dataset, the quantity is finite dataset-wise, but random across datasets, and under heavy-tailed sampling it typically *grows* with $N$. This is exactly what Theorem 3.10 formalizes. We will revise the text after Eq. (5) to make clear that we are not claiming boundedness in $\widehat x$; rather, we are showing that the standard finite-sum SGD proxy is finite conditionally on the dataset yet statistically non-benign as the dataset size increases.
>
> ---
>
> **References**
>
> [1] M. Gürbüzbalaban, U. Şimşekli, and L. Zhu, “The Heavy-Tail Phenomenon in SGD,” ICML, 2021.
>
> [2] M. Barsbey et al., “Heavy Tails in SGD and Compressibility of Overparametrized Neural Networks,” NeurIPS, 2021.
>
> [3] Z. Jiao and M. Keller-Ressel, “Emergence of Heavy Tails in Homogenized Stochastic Gradient Descent,” NeurIPS, 2024.
>
> [4] E. Gorbunov, M. Danilova, and A. Gasnikov, “Stochastic Optimization with Heavy-Tailed Noise via Accelerated Gradient Clipping,” NeurIPS, 2020.
>
> [5] H. Liu and J. Tong, “New Sample Complexity Bounds for Sample Average Approximation in Heavy-Tailed Stochastic Programming,” ICML, 2024.
>
> [6] A. Sadiev et al., “High-Probability Bounds for Stochastic Optimization and Variational Inequalities: the Case of Unbounded Variance,” ICML, 2023.

---

### Decision · Program_Chairs · 2026-04-30

**Decision:**

Accept (regular)

**Comment:**

The paper separates data sampling from optimization randomness in finite-sum ERM under heavy-tailed data, showing that the classical variance proxy grows with dataset size N under heavy tails and that gradient clipping avoids this growth. Convergence guarantees are given for both convex and strongly convex smooth objectives, and matching upper and lower generalization bounds are established in the strongly convex setting.

All four reviewers rate technical soundness as "good." Three give positive recommendations (4/4/4), one gives weak reject (3). The regime-dependent analysis based on the relationship between dataset size N and optimization budget T is recognized as an interesting contribution by Reviewers f8XY, h7cp, and 7SNV. Presentation is rated good to excellent.

The main weakness is limited originality: three reviewers rate it "fair". Reviewer 7SNV notes the ERM analysis relies on existing stability and leave-one-out tools. The compact domain assumption is a restriction (by Reviewer RugF), though Reviewer 7SNV states this is standard for an initial constrained-setting analysis. Experiments are basic.

Post-rebuttal, Reviewers h7cp and 7SNV formally marked all concerns as fully resolved. Reviewer 7SNV actively supports acceptance, considering the theoretical results sufficient for ICML. Reviewer RugF maintains the score of 3 based on the compactness assumption, but rated all four sub-dimensions as "good" (3/4), and acknowledges the novelty of the problem formulation and that the results are solid.

Overall, the paper is a technically sound and clearly written theoretical contribution with matching bounds in a well-motivated setting. The novelty is incremental in tools but meaningful in perspective. I recommend accept.